# Don't be so Monotone: Relaxing Stochastic Line Search in Over-Parameterized Models

**Leonardo Galli, Holger Rauhut**
RWTH Aachen University
Aachen
{galli, rauhut}@mathc.rwth-aachen.de

**Mark Schmidt**
University of British Columbia
Canada CIFAR AI Chair (Amii)
schmidtm@cs.ubc.ca

## Abstract

Recent works have shown that line search methods can speed up Stochastic Gradient Descent (SGD) and Adam in modern over-parameterized settings. However, existing line searches may take steps that are smaller than necessary since they require a monotone decrease of the (mini-)batch objective function. We explore nonmonotone line search methods to relax this condition and possibly accept larger step sizes. Despite the lack of a monotonic decrease, we prove the same fast rates of convergence as in the monotone case. Our experiments show that nonmonotone methods improve the speed of convergence and generalization properties of SGD/Adam even beyond the previous monotone line searches. We propose a POlyak NOnmonotone Stochastic (PoNoS) method, obtained by combining a nonmonotone line search with a Polyak initial step size. Furthermore, we develop a new resetting technique that in the majority of the iterations reduces the amount of backtracks to zero while still maintaining a large initial step size. To the best of our knowledge, a first runtime comparison shows that the epoch-wise advantage of line-search-based methods gets reflected in the overall computational time.

## 1 Introduction

Stochastic Gradient Descent (SGD) [Robbins and Monro, 1951] is the workhorse for the whole Deep Learning (DL) activity today. Even though its simplicity and low memory requirements seem crucial for dealing with these huge models, the success of SGD is strongly connected to the choice of the learning rate. In the field of deterministic optimization [Nocedal and Wright, 2006], this problem is addressed [Armijo, 1966] by employing a line search technique to select the step size. Following this approach, a recent branch of research has started to re-introduce the use of line search techniques for training DL models [Mahsereci and Hennig, 2017, Paquette and Scheinberg, 2018, Bollapragada et al., 2018, Truong and Nguyen, 2018, Vaswani et al., 2019]. These methods have been shown to speed up SGD [Vaswani et al., 2019] and Adam [Duchi et al., 2011, Vaswani et al., 2020] in the over-parameterized regime. However, the existing stochastic line searches may take step sizes that are smaller than necessary since they require a monotone decrease in the (mini-)batch objective function. In particular, the highly non-linear non-convex landscapes of DL training losses suggest that it may be too restrictive to impose such a condition [Grippo et al., 1986]. In addition, the stochastic nature of SGD discourages imposing a monotone requirement on a function that is not directly the one we are minimizing [Ferris and Lucidi, 1994] (mini- vs full-batch function). Furthermore, the recent research on the edge of stability phenomenon [Cohen et al., 2021, Nacson et al., 2022] shows that the best performance for training DL models with Gradient Descent (GD) is obtained when large step sizes yield a nonmonotone decrease of the loss function. In this paper, we propose the use of nonmonotone line search methods [Grippo et al., 1986, Zhang and Hager, 2004] to possibly accept increases in the objective function and thus larger step sizes.

In parallel with line search techniques, many other methods [Baydin et al., 2017, Luo et al., 2019, Mutschler and Zell, 2020, Truong and Nguyen, 2021] have been recently developed to automatically

37th Conference on Neural Information Processing Systems (NeurIPS 2023).

select the step size. In Asi and Duchi [2019], Berrada et al. [2020], Loizou et al. [2021], Gower et al. [2021, 2022], Li et al. [2022], the Polyak step size from Polyak [1969] has been adapted to SGD and successfully employed to train DL models. However, in this setting, the Polyak step size may become very large and it may lead to divergence or, in more favorable cases, induce the (mini-batch) function to decrease nonmonotonically. Starting from Berrada et al. [2020], the Polyak step size has been upper-bounded to avoid divergence and "smoothed" to prevent large fluctuations [Loizou et al., 2021]. However, this "smoothing" technique is a heuristic and, as we will clarify below, may reduce the Polyak step more than necessary. In this paper, we instead combine the Polyak initial step from Loizou et al. [2021] with a nonmonotone line search. In fact, nonmonotone methods are well suited for accepting as often as possible a promising (nonmontone) step size, while still controlling its growth. In the deterministic setting, the same optimization recipe has been used in the seminal papers Raydan [1997] (for the Barzilai and Borwein [1988] (BB) step size) and Grippo et al. [1986] (for the unitary step in the case of Newton).

In the over-parameterized regime, the number of free parameters dominates the number of training samples. This provides modern DL models with the ability of exactly fitting the training data. In practice, over-parameterized models are able to reduce to zero the full-batch loss and consequently also all the individual losses. This is mathematically captured by the interpolation assumption, which allows SGD with non-diminishing step sizes to achieve GD-like convergence rates [Vaswani et al., 2019]. We develop the first rates of convergence for stochastic nonmonotone line search methods under interpolation. In the non-convex case, we prove a linear rate of convergence under the Polyak-Lojasiewicz (PL) condition. In fact in Liu et al. [2022], it has been shown that wide neural networks satisfy a local version of the PL condition, making this assumption more realistic for DL models than the Strong Growth Condition (SGC) [Schmidt and Le Roux, 2013].

When considering line search methods for training DL models one has to take into account that they require one additional forward step for each backtrack (internal iteration) and each mini-batch iteration requires an unknown amount (usually within 5) of backtracks. A single forward step may require up to one-third the runtime of SGD (see Section E.6 of the supplementary materials), thus, too many of them may become a computational burden. Le Roux et al. [2012] proposed a "resetting" technique that is able to (almost) always reduce the number of backtracking steps to zero by shrinking the initial step size [Vaswani et al., 2019]. In this paper, we will show that this technique may have a negative impact on the speed of convergence, even when only employed as a safeguard (as in the "smoothing" technique [Loizou et al., 2021]). As a third contribution, we develop a new resetting technique for the Polyak step size that reduces (almost) always the number of backtracks to zero, while still maintaining a large initial step size. To conclude, we will compare the runtime of different algorithms to show that line search methods equipped with our new technique outperform non-line-search-based algorithms.

## 2  Related Works

The first nonmonotone technique was proposed by Grippo et al. [1986] to globalize the Newton method without enforcing a monotonic decrease of the objective function. After that, nonmonotone techniques have been employed to speed up various optimization methods by relaxing this monotone requirement. A few notable examples are the spectral gradient in Raydan [1997], the spectral projected gradient in Birgin et al. [2000], sequential quadratic programming methods in Zhou and Tits [1993], and the Polak-Ribière-Polyak conjugate gradient in Zhou [2013].

In the pre-deep-learning era [Plagianakos et al., 2002], shallow neural networks have been trained by combining the nonmonotone method [Grippo et al., 1986] with GD. In the context of speech recognition [Keskar and Saon, 2015], one fully-connected neural network has been trained with a combination of a single-pass full-batch nonmonotone line search [Grippo et al., 1986] and SGD. The Fast Inertial Relaxation Engine (FIRE) algorithm is combined with the nonmonotone method [Zhang and Hager, 2004] in Wang et al. [2019, 2021] for solving sparse optimization methods and training logistic regression models, while a manually adapted learning rate is employed to train DL models. In Krejić and Krklec Jerinkić [2015, 2019], Bellavia et al. [2021], the nonmonotone line search by Li and Fukushima [1999] is used together with SGD to solve classical optimization problems. In concurrent work [Hafshejani et al., 2023], the nonmonotone line search by Grippo et al. [1986] has been adapted to SGD for training small-scale kernel models. This line search maintains a nonmonotone window $W$ (usually 10) of weights in memory and it computes the current mini-batch

function value on all of them at every iteration. The method proposed in Hafshejani et al. [2023] is computationally very expensive and impractical to train modern deep learning models due to the need to store previous weights. In this paper, we instead propose the first stochastic adaptation of the nonmonotone line search method by Zhang and Hager [2004]. Thanks to this line search, no computational overhead is introduced in computing the nonmonotone term which is in fact a linear combination of the previously computed mini-batch function values. To the best of our knowledge, we propose the first stochastic nonmonotone line search method to train modern convolutional neural networks and transformers [Vaswani et al., 2017].

Our paper is the first to combine a line search technique with the Stochastic Polyak Step size (SPS) [Loizou et al., 2021]. In fact, the existing line search methods [Vaswani et al., 2019, Paquette and Scheinberg, 2018, Mahsereci and Hennig, 2017] do not address the problem of the selection of a suitable initial step size. In Vaswani et al. [2019], the authors focus on reducing the amount of backtracks and propose a "resetting" technique (3) that ends up also selecting the initial step size. In this paper, we consider these two problems separately and tackle them with two different solutions: a Polyak initial step size and a new resetting technique (5) to reduce the amount of backtracks. Regarding the latter, a similar scheme was presented in Grapiglia and Sachs [2021], however we modify it to be combined with an independent initial step size (e.g., Polyak). This modification allows the original initial step size not to be altered by the resetting technique.

In this paper, we extend Vaswani et al. [2019] by modifying the monotone line search proposed there with a nonmonotone line search. Our theoretical results extend the theorems presented in Vaswani et al. [2019] to the case of stochastic nonmonotone line search methods. Previous results in a similar context were given by Krejić and Krklec Jerinkić [2015, 2019], Bellavia et al. [2021] that assumed 1) the difference between the nonmonotone and monotone terms to be geometrically converging to 0 and 2) the batch-size to be geometrically increasing. In this paper, we replace both hypotheses with a weaker assumption (i.e., interpolation) [Meng et al., 2020] and we actually prove that the nonmonotone and monotone terms converge to the same value. In Hafshejani et al. [2023], related convergence theorems are proved under the interpolation assumption for the use of a different nonmonotone line search (i.e., Grippo et al. [1986] instead of Zhang and Hager [2004]). However in Hafshejani et al. [2023], no explanation is given on how to use an asymptotic result (Lemma 4) for achieving their non-asymptotic convergence rates (Theorem 1 and 2). Addressing this problem represents one of the main challenges of adapting the existing nonmonotone theory [Grippo et al., 1986, Dai, 2002, Zhang and Hager, 2004] to the stochastic case.

In Vaswani et al. [2019], the rates provided in the non-convex case are developed under an SGC, while we here assume the more realistic PL condition [Liu et al., 2022]. In fact in Liu et al. [2022], not only wide networks are proven to satisfy PL but also networks with skip connections (e.g., ResNet [He et al., 2016], DenseNet [Huang et al., 2017]). For this result, we extend Theorem 3.6 of Loizou et al. [2021] for stochastic nonmonotone line search methods by exploiting our proof technique. In Loizou et al. [2021], various rates are developed for SGD with a Polyak step size and it is possible to prove that they hold also for PoNoS. In fact, the step size yielded by PoNoS is by construction less or equal than Polyak's. However, the rates presented in this paper are more general since they are developed for nonmonotone line search methods and do not assume the use of any specific initial step size (e.g., they hold in combination with a BB step size).

## 3 Methods

Training machine learning models (e.g., neural networks) entails solving the finite sum problem $\min_{w \in \mathbb{R}^n} f(w) = \frac{1}{M} \sum_{i=1}^{M} f_i(w)$, where $w$ is the parameter vector and $f_i$ corresponds to a single instance of the $M$ points in the training set. We assume that $f$ is lower-bounded by some value $f^*$, that $f$ is $L$-smooth and that it either satisfies the PL condition, convexity or strong-convexity. Vanilla SGD can be described by the step $w_{k+1} = w_k - \eta \nabla f_{i_k}(w_k)$, where $i_k \in \{1, \ldots, M\}$ is one instance randomly sampled at iteration $k$, $\nabla f_{i_k}(w_k)$ is the gradient computed only w.r.t. the $i_k$-th instance and $\eta > 0$ is the step size. The mini-batch version of this method modifies $i_k$ to be a randomly selected subset of instances, i.e., $i_k \subset \{1, \ldots, M\}$, with $\nabla f_{i_k}(w_k)$ being the averaged gradient on this subset and $|i_k| = b$ the mini-batch size. Through the whole paper we assume that each stochastic function $f_{i_k}$ and gradient $\nabla f_{i_k}(w)$ evaluations are unbiased, i.e., $\mathbb{E}_{i_k}[f_{i_k}(w)] = f(w)$ and $\mathbb{E}_{i_k}[\nabla f_{i_k}(w)] = \nabla f(w)$, $\forall w \in \mathbb{R}^n$. Note that $\mathbb{E}_{i_k}$ represents the conditional expectation w.r.t. $w_k$, i.e., $\mathbb{E}_{i_k}[\cdot] = \mathbb{E}[\cdot | w_k]$. In other words, $\mathbb{E}_{i_k}$ is the expectation computed w.r.t. $\nu_k$, that is the

random variable associated with the selection of the sample (or subset) at iteration $k$ (see Bottou et al. [2018] for more details).

We interpret the over-parameterized setting in which we operate to imply the interpolation property, i.e., let $w^* \in \underset{w \in \mathbb{R}^n}{\mathrm{argmin}} f(w)$, then $w^* \in \underset{w \in \mathbb{R}^n}{\mathrm{argmin}} f_i(w) \; \forall 1 \leq i \leq M$. This property is crucial for the convergence results of SGD-based methods because it can be combined either with the Lipschitz smoothness of $f_{i_k}$ or with a line search condition to achieve the bound $\mathbb{E}_{i_k} \| \nabla f_{i_k}(w_k) \|^2 \leq a \left( f(w_k) - f(w^*) \right)$ with $a > 0$ (see Lemma 4 of the supplementary materials for the proof). This bound on the variance of the gradient results in a r.h.s. which is independent of $i_k$ and, in the convex and strongly convex cases, it can be used to replace the SGC (see Fan et al. [2023] for more details).

As previously stated, our algorithm does not employ a constant learning rate $\eta$, but the step size $\eta_k$ is instead determined at each iteration $k$ by a line search method. Given an initial step size $\eta_{k,0}$ and $\delta \in (0,1)$, the Stochastic Line Search (SLS) condition [Vaswani et al., 2019] select the smallest $l_k \in \mathbb{N}$ such that $\eta_k = \eta_{k,0}\delta^{l_k}$ satisfies the following condition

$$f_{i_k}(w_k - \eta_k \nabla f_{i_k}(w_k)) \leq f_{i_k}(w_k) - c\eta_k \| \nabla f_{i_k}(w_k) \|^2, \tag{1}$$

where $c \in (0,1)$ and $\| \cdot \|$ is the Euclidean norm. Each internal step of the line search is called backtrack since the procedure starts with the largest value $\eta_{k,0}$ and reduces (cuts) it until the condition is fulfilled. Note that (1) requires a monotone decrease of $f_{i_k}$. In this paper, we follow Zhang and Hager [2004] and propose to replace $f_{i_k}$ with a nonmonotone term $C_k$,

$$f_{i_k}(w_k - \eta_k \nabla f_{i_k}(w_k)) \leq C_k - c\eta_k \| \nabla f_{i_k}(w_k) \|^2,$$
$$C_k = \max \left\{ \tilde{C}_k; f_{i_k}(w_k) \right\}, \; \tilde{C}_k = \frac{\xi Q_k C_{k-1} + f_{i_k}(w_k)}{Q_{k+1}}, \; Q_{k+1} = \xi Q_k + 1, \tag{2}$$

where $\xi \in [0,1]$, $C_0 = Q_0 = 0$ and $C_{-1} = f_{i_0}(w_0)$. The value $\tilde{C}_k$ is a linear combination of the previously computed function values $f_{i_0}(w_0), \ldots, f_{i_k}(w_k)$ and it ranges between the strongly nonmonotone term $\frac{1}{k} \sum_{j=0}^{k} f_{i_j}(w_j)$ (with $\xi = 1$) and the monotone value $f_{i_k}(w_k)$ (with $\xi = 0$). The maximum in $C_k$ is needed to ensure the use of a value that is always greater or equal than the monotone term $f_{i_k}(w_k)$. The activation of $f_{i_k}(w_k)$ instead of $\tilde{C}_k$ happens only in the initial iterations and it is rare in deterministic problems, but becomes more common in the stochastic case. The computation of $C_k$ does not introduce overhead since the linear combination is accumulated in $C_k$.

Given $\gamma > 1$, the initial step size employed in Vaswani et al. [2019] is the following

$$\eta_{k,0} = \min\{\eta_{k-1}\gamma^{b/M}, \eta^{\max}\}. \tag{3}$$

In this paper, we propose to use the following modified version of SPS [Loizou et al., 2021]

$$\eta_{k,0} = \min \left\{ \tilde{\eta}_{k,0}, \eta^{\max} \right\} \quad \text{with } \tilde{\eta}_{k,0} := \frac{f_{i_k}(w_k) - f_{i_k}^*}{c_p \| \nabla f_{i_k}(w_k) \|^2} \quad \text{and } c_p \in (0,1). \tag{4}$$

In Loizou et al. [2021], (4) is not directly employed, but equipped with the same resetting technique (3) to control the growth of $\tilde{\eta}_{k,0}$, i.e. $\eta_{k,0} = \min \left\{ \tilde{\eta}_{k,0}, \eta_{k-1}\gamma^{b/M}, \eta^{\max} \right\}$.

Given $l_{k-1}$ the amount of backtracks at iteration $k-1$, our newly introduced resetting technique stores this value for using it in the following iteration. In particular, to relieve the line search at iteration $k$ from the burden of cutting the new step size $l_{k-1}$ times, the new initial step size is pre-scaled by $\delta^{l_{k-1}}$. In fact, $l_{k-1}$ is a good estimate for $l_k$ (see Section E.3 of the supplementary materials). However, to allow the original $\eta_{k,0}$ to be eventually used non-scaled, we reduce $l_{k-1}$ by one. We thus redefine $\eta_k$ as

$$\eta_k = \eta_{k,0}\delta^{\bar{l}_k}\delta^{l_k}, \quad \text{with } \bar{l}_k := \max\{\bar{l}_{k-1} + l_{k-1} - 1, 0\}. \tag{5}$$

In Section 5, our experiments show that thanks to (5) we can save many backtracks and reduce them to zero in the majority of the iterations. At the same time, the original $\eta_{k,0}$ is not modified and the resulting step size $\eta_k$ is always a scaled version of $\eta_{k,0}$. Moreover, the presence of the $-1$ in (5) keeps pushing $\delta^{\bar{l}_k}$ towards zero, so that the initial step size is not cut more than necessary. Note that (5) can be combined with any initial step size and it is not limited to the case of (4).

# 4 Rates of Convergence

We present the first rates of convergence for nonmonotone stochastic line search methods under interpolation (see Section B of the supplementary materials for the proofs). The three main theorems are given under strong-convexity, convexity and the PL condition. Our results do not prove convergence only for PoNoS, but more generally for methods employing (2) and any bounded initial step size, i.e.

$$\eta_{k,0} \in [\bar{\eta}^{\min}, \eta^{\max}], \qquad \text{with } \eta^{\max} > \bar{\eta}^{\min} > 0. \tag{6}$$

Many step size rules in the literature fulfill the above requirement [Berrada et al., 2020, Loizou et al., 2021, Liang et al., 2019] since when applied to non-convex DL models the step size needs to be upper bounded to achieve convergence [Bottou et al., 2018, Berrada et al., 2020].

In Theorem 1 below, we show a linear rate of convergence in the case of a strongly convex function $f$, with each $f_{i_k}$ being convex. We are able to recover the same speed of convergence of Theorem 1 in Vaswani et al. [2019], despite the presence of the nonmonotone term $C_k$ instead of $f_{i_k}(w_k)$. The challenges of the new theorem originate from studying the speed of convergence of $C_k$ to $f(w^*)$ and from combining the two interconnected sequences $\{w_k\}$ and $\{C_k\}$. It turns out that if $\xi$ is small enough (i.e., such that $b < (1 - \eta^{\min}\mu)$), then the presence of the nonmonotone term does not alter the speed of convergence of $\{w_k\}$ to $w^*$, not even in terms of constants.

**Theorem 1.** *Let $C_k$ and $\eta_k$ be defined in (2), with $\eta_{k,0}$ defined in (6). We assume interpolation, $f_{i_k}$ convex, $f$ $\mu$-strongly convex and $f_{i_k}$ $L_{i_k}$-Lipschitz smooth. Assuming $c > \frac{1}{2}$ and $\xi < \frac{1}{\left(1 + \frac{\eta^{max}}{\eta^{min}(2c-1)}\right)}$, we have*

$$\mathbb{E}\left[\|w_{k+1} - w^*\|^2 + a(C_k - f(w^*))\right] \leq d^k \left(\|w_0 - w^*\|^2 + a\left(f(w_0) - f(w^*)\right)\right),$$

*where $d := \max\left\{(1 - \eta^{min}\mu), b\right\} \in (0,1)$, $b := \left(1 + \frac{\eta^{max}}{ac}\right)\xi \in (0,1)$, $a := \eta^{min}\left(2 - \frac{1}{c}\right) > 0$ with $\eta^{min} := \min\{\frac{2\delta(1-c)}{L_{max}}, \bar{\eta}^{min}\}$.*

Comparing the above result with the corresponding deterministic rate (Theorem 3.1 from Zhang and Hager [2004]), we notice that the constants of the two rates are different. In particular, the proof technique employed in Theorem 3.1 from Zhang and Hager [2004] cannot be reused in Theorem 1 since a few bounds are not valid in the stochastic case (e.g., $\|\nabla f(w_{k+1})\| \leq \|\nabla f(w_k)\|$ or $C_{k+1} \leq C_k - \|\nabla f_{i_k}(w_k)\|$). The only common aspect in these proofs is the distinction between the two different cases defining $C_k$ (i.e., being $\tilde{C}_k$ or $f_{i_k}(w_k)$). Moreover, in both theorems $\xi$ needs to be small enough to allow the final constant to be less than 1.

In Theorem 2 we show a $O(\frac{1}{k})$ rate of convergence in the case of a convex function $f$. Interestingly, in both Theorems 1 and 2 (and also in the corresponding monotone versions from Vaswani et al. [2019]), $c$ is required to be larger than $\frac{1}{2}$. The same lower bound is also required for achieving Q-super-linear convergence for the Newton method in the deterministic case, but it is often considered too large in practice and the default value (also in the case of first-order methods) is $0.1$ or smaller [Nocedal and Wright, 2006]. In this paper, we numerically tried both values and found out that $c = 0.1$ is too small since it may indeed lead to divergence (see Section E.1 of the supplementary materials).

**Theorem 2.** *Let $C_k$ and $\eta_k$ be defined in (2), with $\eta_{k,0}$ defined in (6). We assume interpolation, $f$ convex and $f_{i_k}$ $L_{i_k}$-Lipschitz smooth. Given a constant $a_1$ such that $0 < a_1 < \left(2 - \frac{1}{c}\right)$ and assuming $c > \frac{1}{2}$ and $\xi < \frac{a_1}{2}$, we have*

$$\mathbb{E}\left[f(\bar{w}_k) - f(w^*)\right] \leq \frac{d_1}{k}\left(\frac{1}{\eta^{min}}\|w_0 - w^*\|^2 + a_1\left(f(w_0) - f(w^*)\right)\right),$$

*where $\bar{w}_k = \frac{1}{k}\sum_{j=0}^{k} w_j$, $d_1 := \frac{c}{c(2-a_1)-1} > 0$, $b_1 := \left(1 + \frac{1}{a_1 c}\right)\xi \in (0,1]$, $\eta^{min} := \min\{\frac{2\delta(1-c)}{L_{max}}, \bar{\eta}^{min}\}$.*

In Theorem 3 below we prove a linear convergence rate in the case of $f$ satisfying a PL condition. We say that a function $f : \mathbb{R}^n \to \mathbb{R}$ satisfies the PL condition if there exists $\mu > 0$ such that, $\forall w \in \mathbb{R}^n : \|\nabla f(w)\|^2 \geq 2\mu(f(w) - f(w^*))$. The proof of this theorem extends Theorem 3.6 of Loizou et al. [2021] to the use of a stochastic nonmonotone line search. Again here, the presence of a nonmonotone term does not modify the constants controlling the speed of convergence ($a_2$ below can be chosen arbitrarily small) as long as $\xi$ is small enough. The conditions on $c$ and on $\eta^{\max}$ are the same as those of Theorem 3.6 of Loizou et al. [2021] (see the proof).

**Theorem 3.** *Let $C_k$ and $\eta_k$ be defined in (2), with $\eta_{k,0}$ defined in (6). We assume interpolation, the PL condition on $f$ and that $f_{i_k}$ are $L_{i_k}$-Lipschitz smooth. Given $0 < a_2 := \frac{4\mu c(1-c) - L_{max}}{4\delta c(1-c)} + \frac{1}{2\eta^{max}}$ and assuming $\frac{2\delta(1-c)}{L_{max}} < \bar{\eta}^{min}, \eta^{max} < \frac{2\delta c(1-c)}{L_{max} - 4\mu c(1-c)}, \frac{L_{max}}{4\mu} < c < 1$ and $\xi < \frac{a_2 c}{a_2 c + L_{max}}$, we have*

$$\mathbb{E}\left[ f(w_{k+1}) - f(w^*) + a_2 \eta^{max}(C_k - f(w^*)) \right] \leq d_2^k \left(1 + a_2 \eta^{max}\right)\left(f(w_0) - f(w^*)\right)$$

*where $d_2 := \min\{\nu, b_2\} \in (0, 1)$, $\nu := \eta^{max}\left(\frac{L_{max} - 4\mu c(1-c)}{2\delta c(1-c)} + a_2\right) \in (0, 1)$, $b_2 := \left(1 + \frac{L_{max}}{a_2 c}\right)\xi \in (0, 1)$.*

Theorem 3 is particularly meaningful for modern neural networks because of the recent work of Liu et al. [2022]. More precisely, their Theorem 8 and Theorem 13 prove that, respectively given a wide-enough fully-connected or convolutional/skip-connected neural network, the corresponding function satisfies a local version of the PL condition. As a consequence, nonmonotone line search methods achieve linear convergence for training of non-convex neural networks if the assumptions of both Theorem 3 above and Theorem 8 from Liu et al. [2022] hold and if the initial point $w_0$ is close enough to $w^*$.

## 5 Experiments

In this section, we benchmark PoNoS against state-of-the-art methods for training different over-parametrized models. We focus on various deep learning architectures for multi-class image classification problems, while in Section 5.3 we consider transformers for language modelling and kernel models for binary classification. In the absence of a better estimate, we assume $f_{i_k}(w^*) = 0$ for all the experiments. Indeed the final loss is in many problems very close to 0.

Concerning datasets and architectures for the image classification task, we follow the setup by Luo et al. [2019], later employed also in Vaswani et al. [2019], Loizou et al. [2021]. In particular, we focus on the datasets MNIST, FashionMNIST, CIFAR10, CIFAR100 and SVHN, addressed with the architectures MLP [Luo et al., 2019], EfficientNet-b1 [Tan and Le, 2019], ResNet-34 [He et al., 2016], DenseNet-121 [Huang et al., 2017] and WideResNet [Zagoruyko and Komodakis, 2016]. Because of space limitation, we will not show all the experiments here, but report on the first three datasets combined respectively with the first three architectures. We refer to the supplementary materials for the complete benchmark (see Figures I-XII) and the implementation details (Section C). In Section C of the supplementary materials, we also discuss the sensitivity of the results to the selection of hyper-parameters.

In Figure 1 we compare SGD [Robbins and Monro, 1951], Adam [Kingma and Ba, 2015], SLS [Vaswani et al., 2019], SPS [Loizou et al., 2021] and PoNoS. We present train loss (log scale), test accuracy and average step size within the epoch (log scale). Note that the learning rate of SGD and Adam has been chosen through a grid-search as the one achieving the best performance on each problem. Based on Figure 1, we can make the following observations:

- PoNoS achieves the best performances both in terms of train loss and test accuracy. In particular, it often terminates by reaching a loss of $10^{-6}$ and it always achieves the highest test accuracy.
- SLS does not always achieve the same best accuracy as PoNoS. In terms of training loss, SLS is particularly slow on `mnist|mlp` and `fashion|effb1`, while competitive on the others. On these two problems, both SLS and SPS employ a step size that is remarkably smaller than that of PoNoS. On the same problems, SPS behaves similarly as SLS because the original Polyak step size is larger than the "smoothing/resetting" value (3), i.e., $\tilde{\eta}_{k,0} > \eta_{k-1}\gamma^{b/M}$ and thus the step size is controlled by (3). On the other hand, the proposed nonmonotone line search selects a larger step size, achieving faster convergence and better generalization [Nacson et al., 2022].
- SLS encounters some cancellation errors (e.g., around epoch 120 in `cifar10|resnet34`) that lead the step size to be reduced drastically, sometimes reaching values below $10^{-7}$. These events are caused by the fact that $f_{i_k}(w_k)$ and $f_{i_k}(w_{k+1})$ became numerically identical. Thanks to the nonmonotone term $C_k$ replacing $f_{i_k}(w_k)$, PoNoS avoids cancellations errors by design.
- SGD, Adam and SPS never achieve the best test accuracy nor the best training loss. In accordance with the results in Vaswani et al. [2019], the line-search-based algorithms outperform the others.

Because of space limitation, we defer the comparison between PoNoS and its monotone counterpart to the supplementary materials (Section E.2). There we show that PoNoS outperforms the other line search techniques, including a tractable stochastic adaptation of the method by Grippo et al. [1986].

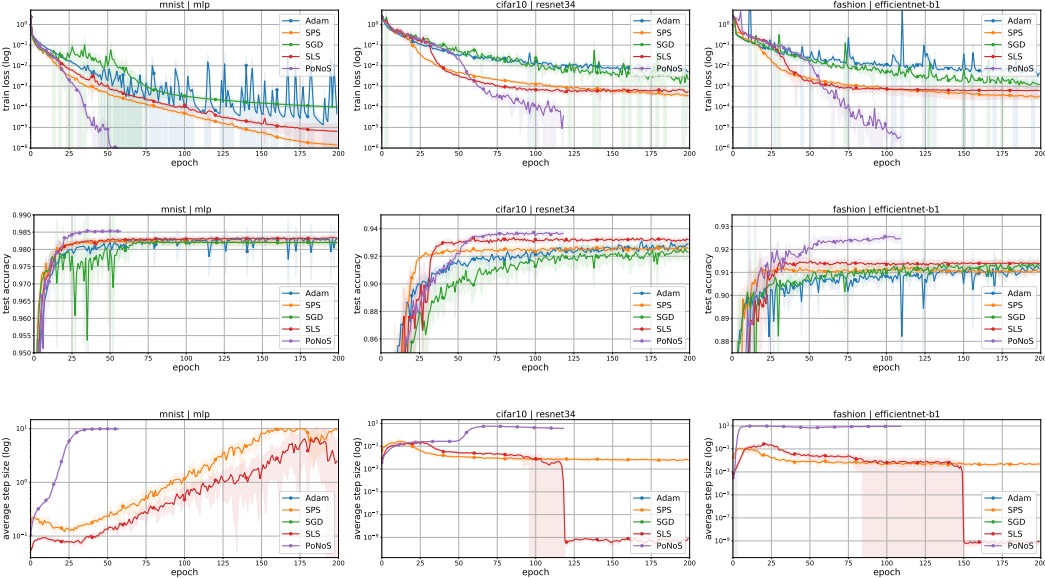

Figure 1: Comparison between the proposed method (PoNoS) and the-state-of-the-art. Each column focus on a dataset. First row: train loss. Second row: test accuracy. Third row: step size.

## 5.1 A New Resetting Technique

In this subsection, we compare different initial step sizes and resetting techniques. We fix the line search to be (2), but similar comments can be made on (1) (see the Section D.2 of the supplementary materials). In Figure 2 we compare PoNoS and PoNoS_reset0 (without (5)) with zhang_reset2 (initial step size (3)) and:

- zhang_reset3: $\eta_{k,0} = \eta_{k-1} \frac{||\nabla f_{i_{k-1}}(w_{k-1})||^2}{||\nabla f_{i_k}(w_k)||^2}$, adapted to SGD from Nocedal and Wright [2006],

- zhang_reset4: $\eta_{k,0} = \frac{2\left(f_{i_{k-1}}(w_{k-1}) - f_{i_{k-1}}(w_k)\right)}{||\nabla f_{i_{k-1}}(w_{k-1})||}$. adapted to SGD from Nocedal and Wright [2006],

- zhang_every2: same as PoNoS_reset0, but a new step is computed only every 2 iterations.

In Figure 2, we report train loss (log scale), the total amount of backtracks per epoch and the average step size within the epoch (log scale). From Figure 2, we can make the following observations:

- PoNoS and PoNoS_reset0 achieve very similar performances. In fact, the two algorithms yield step sizes that are almost always overlapping. An important difference between PoNoS_reset0 and PoNoS can be noticed in the amount of backtracks that the two algorithms require. The plots show a sum of $\frac{M}{b}$ elements, with $\frac{M}{b} = 391$ or $469$ depending on the problem and PoNoS's line hits exactly this value. This means that PoNoS employs a median of 1 backtrack per iteration for the first 5-25 epochs, while PoNoS_reset0 needs more backtracks in this stage (around 1500-3000 per-epoch, see Section D.2 of the supplementary materials). After this initial phase, both PoNoS_reset0 and PoNoS reduce the amount of backtracks until it reaches a (almost) constant value of 0.

- zhang_every2 does not achieve the same good performance as PoNoS or PoNoS_reset0. The common belief that step sizes can be used in many subsequent iterations does not find confirmation here. In fact, zhang_every2 shows that we cannot skip the application of a line search if we want to maintain the same good performances.

- All the other initial step sizes achieve poor performances on both train loss and test accuracy. In many cases, the algorithms yield step sizes (also before the line search) that are too small w.r.t. (4).

## 5.2 Time Comparison

In this subsection, we show runtime comparisons corresponding to Figure 1. In the first row of Figure 3, we plot the train loss as in the first row of Figure 1. However, the $x$-axis of Figure 3 measures the cumulative epoch time of the average of 5 different runs of the same algorithm with different seeds. In the second row of Figure 3, we report the runtime per-epoch (with shaded error bars) on

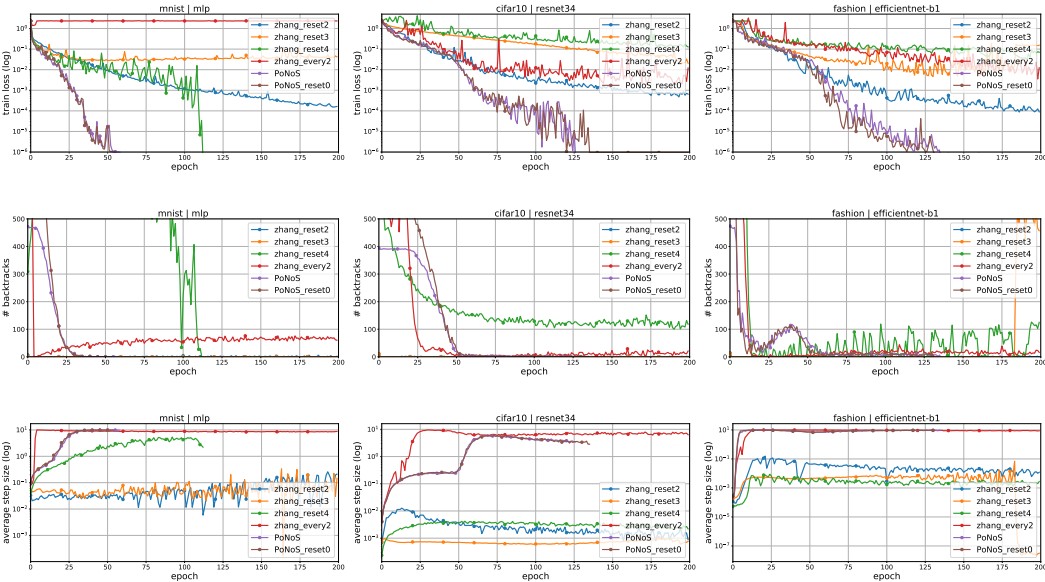

Figure 2: Comparison between different initial step sizes and resetting techniques. Each column focus on a dataset. First row: train loss. Second row: # backtracks. Third row: step size.

the $y$-axis and epochs on the $x$-axis. From Figures 1 and 3, it is clear that PoNoS is not only faster than the other methods in terms of epochs but also in terms of total computational time. In particular, PoNoS is faster than SGD, despite the fact the second achieves the lowest per-epoch time. Again from the second row of Figure 3, we can observe that PoNoS's per-epoch time makes a transition from the phase of a median of 1 backtrack (first 5-25 epochs) to the phase of a median of 0 backtracks where its time is actually overlapping with that of SLS (always a median of 0 backtracks). In the first case, PoNoS requires less than twice the time of SGD, and in the second, this time is lower than $1.5$ that of SGD. Given these measures, PoNoS becomes faster than SGD/Adam in terms of per-epoch time as soon as a grid-search (or any other hyper-parameter optimization) is employed to select the best-performing learning rate.

To conclude, let us recall that any algorithm based on a stochastic line search always requires one additional forward pass if compared with SGD. In fact, both $f_{i_k}(w_k)$ and $f_{i_k}(w_{k+1})$ are computed at each $k$ and each backtrack requires one additional forward pass. On the other hand, if we consider that one backward pass costs roughly two forward passes and that SGD needs one forward and one backward pass, any additional forward pass costs roughly one-third of SGD. These rough calculations have been verified in Section E.6 of the supplementary materials, where we profiled the single iteration of PoNoS. Following these calculations and referring to the two phases of Figure 3, one iteration of PoNoS only costs $\frac{5}{3}$ that of SGD in the first phase and $\frac{4}{3}$ in the second.

## 5.3 Experiments for Convex Losses and for Transformers

As a last benchmark, we take into account a set of convex problems from Vaswani et al. [2019], Loizou et al. [2021] and the transformers [Vaswani et al., 2017] trained from scratch in Kunstner et al. [2023]. In Figure 4 we show one convex problem (first column) and two transformers (last two column). We leave the fine-tuning of transformers to future works. Our experiments take into account binary classification problems addressed with a RBF kernel model without regularization. We show the results achieved on the dataset mushrooms, while leaving those on ijcnn, rcv1 and w8a to Section D.4 of the supplementary materials. Given the high amount of iterations, the smoothed version of the train loss will be reported. The test accuracy is also only reported in Section D.4 since (almost) all the methods achieve the best test accuracy in all the problems within the first few iterations. From the left subplot of Figure 4, we can observe that PoNoS obtains very good performances also in this setting (see supplementary materials). In Figure 4, PoNoS achieves a very low train loss ($10^{-4}$) within the first 200 iterations. Only SLS is able to catch up, but this takes 6 times the iterations of PoNoS. On this problem, SLS is the only method reaching the value ($10^{-6}$). The methods SLS and SPS behave

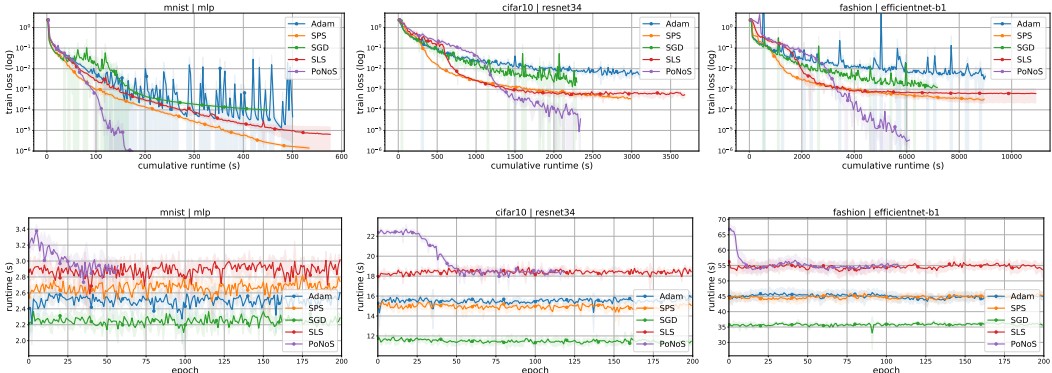

Figure 3: Time comparison (s) between the proposed method (PoNoS) and the-state-of-the-art. Each column focus on a dataset. First row: train loss vs runtime. Second row: epoch runtime.

very similarly on all the datasets (see the supplementary materials) since in both cases (3) controls the step size. As clearly shown by the comparison with PoNoS, this choice is suboptimal and the Polyak step size is faster. Because of (3), both SLS and SPS encounter slow convergence issues in many of problems of this setting. As in Figure 1, SGD and Adam are always slower than PoNoS.

We consider training transformers on language modeling datasets. In particular, we train a Transformer Encoder [Vaswani et al., 2017] on PTB [Marcus et al., 1993] and a Transformer-XL [Dai et al., 2019] on Wikitext2 [Merity et al., 2017]. In contrast to the case of convolutional neural networks, the most popular method for training transformers is Adam and not SGD [Pan and Li, 2022, Kunstner et al., 2023]. For this reason, we use a preconditioned version of PoNoS, SLS, and SPS (respectively PoNoS_prec, SLS_prec, and SPS_prec, see Section D.5 of the supplementary materials for details). In fact, Adam can be considered a preconditioned version of SGD with momentum [Vaswani et al., 2020]. As in Kunstner et al. [2023], we focus on the training procedure and defer the generalization properties to the supplementary materials. From Figure 4, we can observe that the best algorithms in this setting are preconditioned-based and not SGD-based, in accordance with the literature. Moreover, PoNoS_prec achieves similar performances as Adam's. In particular from the central subplot of Figure 4, we observe that PoNoS_prec is the only algorithm as fast as Adam, while all the others have difficulties achieving loss below 1. Taking the right subplot of Figure 4 into account, we can notice that PoNoS_prec is slower than Adam on this problem. On the other hand, there is not one order of difference between the two final losses (i.e., $\sim 1.5$ points). Moreover, we should keep in mind that Adam's learning rate has been fine-tuned separately for each problem, while PoNoS_prec has been used off-the-shelf.

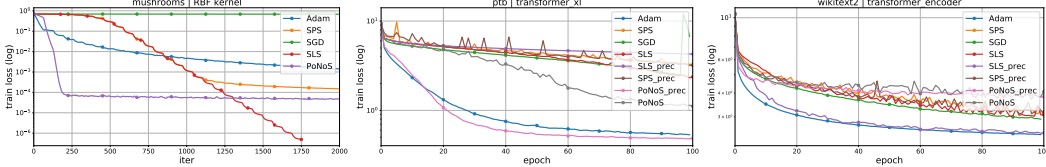

Figure 4: Train loss comparison between the new method (PoNoS) and the state-of-the-art on convex kernel models (first column) and transformers (last two columns).

## 6 Conclusion

In this work, we showed that modern DL models can be efficiently trained by nonmonotone line search methods. More precisely, nonmonotone techniques have been shown to outperform the monotone line searches existing in the literature. A stochastic Polyak step size with resetting has been employed as the initial step size for the nonmonotone line search, showing that the combined method is faster than the version without line search. Moreover, we presented the first runtime comparison

between line-search-based methods and SGD/Adam. The results show that the new line search is overall computationally faster than the state-of-the-art. A new resetting technique is developed to reduce the amount of backtracks to almost zero on average, while still maintaining a large initial step size. To conclude, the similar behavior of SLS_prec and Adam on the rightmost subplot of Figure 4 suggests that other initial step sizes might also be suited for training transformers. We leave such exploration (e.g., a stochastic BB like in Tan et al. [2016], Liang et al. [2019]) to future works.

We proved three convergence rate results for stochastic nonmonotone line search methods under interpolation and under either strong-convexity, convexity, or the PL condition. Our theory matches its monotone counterpart despite the use of a nonmonotone term. In the future, we plan to explore the conditions of the theorems in Liu et al. [2022] and to study a bridge between their local PL results and our global PL assumption. To conclude, it is worth mentioning that nonmonotone line search methods might be connected to the edge of stability phenomenon described in Cohen et al. [2021], because of the very similar behavior they induce in the decrease of (deterministic) objective functions. However, a rigorous study remains for future investigations.

## Acknowledgments

The work was conducted within the KI-Starter project "Robustness and Generalization in Training Deep Neural Networks" funded by the Ministry of Culture and Science Nordrhein-Westfalen, Germany and partially supported by the Canada CIFAR AI Chair Program.

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
