# Supplementary Materials for
# "Don't be so Monotone: Relaxing Stochastic Line Search in Over-Parameterized Models"

**Leonardo Galli, Holger Rauhut**
RWTH Aachen University
Aachen
{galli, rauhut}@mathc.rwth-aachen.de

**Mark Schmidt**
University of British Columbia
Canada CIFAR AI Chair (Amii)
schmidtm@cs.ubc.ca

## Content

37th Conference on Neural Information Processing Systems (NeurIPS 2023).

# A  The Algorithm

In this section, we give the details of our proposed algorithm PoNoS.

Training machine learning models (e.g., neural networks) entails solving the following **finite sum problem:**

$$\min_{w \in \mathbb{R}^n} f(w) = \frac{1}{M} \sum_{i=1}^{M} f_i(w), \tag{1}$$

where $w$ is the parameter vector and $f_i$ corresponds to a single instance of the $M$ points in the training set.

Given an initial step size $\eta_{k,0}$ and $\delta \in (0,1)$, the **Stochastic (Amijo) Line Search (SLS)** [Vaswani et al., 2019] select the smallest $l_k \in \mathbb{N}$ such that $\eta_k = \eta_{k,0}\delta^{l_k}$ satisfies the following condition:

$$f_{i_k}(w_k - \eta_k \nabla f_{i_k}(w_k)) \le f_{i_k}(w_k) - c\eta_k \| \nabla f_{i_k}(w_k)\|^2, \tag{2}$$

where $c \in (0,1)$ and $\| \cdot \|$ is the Euclidean norm.

The newly proposed **Stochastic Zhang & Hager line search** adapted from Zhang and Hager [2004] is

$$f_{i_k}(w_k - \eta_k \nabla f_{i_k}(w_k)) \le C_k - c\eta_k \| \nabla f_{i_k}(w_k)\|^2,$$
$$C_k = \max \left\{ \tilde{C}_k; f_{i_k}(w_k) \right\}, \ \tilde{C}_k = \frac{\xi Q_k C_{k-1} + f_{i_k}(w_k)}{Q_{k+1}}, \ Q_{k+1} = \xi Q_k + 1, \tag{3}$$

where $\xi \in [0,1]$, $C_0 = Q_0 = 0$ and $C_{-1} = f_{i_0}(w_0)$.

Given $\gamma > 1$, the **resetting/smoothing technique** employed in Vaswani et al. [2019] is

$$\eta_{k,0} = \eta_{k-1}\gamma^{b/M}, \tag{4}$$

where $\gamma > 1$ and $b$ is the mini-batch size.

Given the step size

$$\tilde{\eta}_{k,0} := \frac{f_{i_k}(w_k) - f_{i_k}^*}{c_p \| \nabla f_{i_k}(w_k)\|^2} \quad \text{and } c_p \in (0,1), \tag{5}$$

we recall **Stochastic Polyak Step (SPS) size** from Loizou et al. [2021]

$$\eta_{k,0} = \min \left\{ \tilde{\eta}_{k,0}, \eta_{k-1}\gamma^{b/M}, \eta^{\max} \right\} \quad \text{with } \eta^{\max} > 0 \text{ and } \tilde{\eta}_{k,0} \text{ defined in (5)}, \tag{6}$$

and its **non-smoothed version** (used in Algorithm 1)

$$\eta_{k,0} = \min \left\{ \tilde{\eta}_{k,0}, \eta^{\max} \right\} \quad \text{with } \tilde{\eta}_{k,0} \text{ defined in (5)}. \tag{7}$$

To employ our new **resetting technique**, we redefine $\eta_k$ as

$$\eta_k = \eta_{k,0}\delta^{\bar{l}_k}\delta^{l_k}, \quad \text{with } \bar{l}_k := \max\{\bar{l}_{k-1} + l_{k-1} - 1, 0\}. \tag{8}$$

---

**Algorithm 1:** The POlyak NOnmonotone Stochastic (PoNoS) line search method

---

**Input:** $D = \{(x_i, y_i)\}_{i=1}^{M}, w_0 \in \mathbb{R}^n, \eta^{\max} > 0, c \in (0,1), c_p \in (0,1), \delta \in (0,1), \xi \in [0,1], b$ mini-batch size, $Q_0 = 0, k = 0$

1   **for** $epoch = 0, 1, 2, \ldots, max\_epoch$ **do**
2     **for** $i = 0, 1, 2, \ldots, \frac{M}{b}$ **do**
3       sample $i_k \subset \{1, \ldots, M\} : |i_k| = b$
4       $\eta_{k,0} = (7)$
5       $l_k = 0$
6       **if** *k=0* **then**
7         $C_{-1} = f_{i_0}(w_0)$
8       $\tilde{C}_k = \frac{\xi Q_k C_{k-1} + f_{i_k}(w_k)}{\xi Q_k + 1}$
9       $C_k = \max \left\{ \tilde{C}_k; f_{i_k}(w_k) \right\}$
10      **while** $f_{i_k}(w_k - \eta_k \nabla f_{i_k}(w_k)) > C_k - c\eta_k \| \nabla f_{i_k}(w_k)\|^2$ **do**
11        $l_k = l_k + 1$
12        $\eta_k = \eta_{k,0}\delta^{\bar{l}_k}\delta^{l_k}$
13      $w_{k+1} = w_k - \eta_k \nabla f_{i_k}(w_k)$
14      $\bar{l}_{k+1} := \max\{\bar{l}_k + l_k - 1, 0\}$
15      $Q_{k+1} = \xi Q_k + 1$
16      $k = k + 1$

---

# B  Convergence Rates

Our results do not prove convergence only for PoNoS, but more in general for methods employing (3) as a line search and a bounded initial step size, i.e.,

$$\eta_{k,0} \in [\bar{\eta}^{\min}, \eta^{\max}], \quad \text{with } \eta^{\max} > \bar{\eta}^{\min} > 0. \tag{9}$$

The next lemma provides the possible range of $\eta_k$ as a consequence of the line search technique. Thanks to the fact that we always have $C_k \geq f_{i_k}(w_k)$, Lemma 1 recovers the monotone range (see Lemma 1 in Vaswani et al. [2019]). We say that $f$ is $L$-Lipschitz smooth when $f$ is continuously differentiable with Lipschitz continuous gradient, i.e.

$$\| \nabla f(x) - \nabla f(y)\| \leq L\|x - y\| \quad \forall x, y \in \mathbb{R}^n, \quad \text{with } L > 0.$$

**Lemma 1.** *Let $f_{i_k}$ be $L_{i_k}$-Liptschitz smooth. The range of the step size $\eta_k$ returned by (3) and with $\eta_{k,0}$ defined in (9) is either*

$$\eta_k \in \begin{cases} [\bar{\eta}^{min}, \eta^{max}] & \text{if } l_k = 0, \\ [\eta^{min}, \eta^{max}] & \text{if } l_k > 0, \end{cases} \tag{10}$$

*where $\eta^{min} := \min\left\{ \frac{2\delta(1-c)}{L_{max}}, \bar{\eta}^{min} \right\}$ and $L_{max} = \max_i L_i$.*

*Proof.* Let us denote $g_k := \nabla f_{i_k}(w_k)$. Applying Lemma 5 below on $f_{i_k}$, with $y = w_k - \eta_k g_k$ and $x = w_k$ we have

$$f_{i_k}(w_k - \eta_k g_k) \leq f_{i_k}(w_k) + g_k^T(w_k - \eta_k g_k - w_k) + \frac{\eta_k^2 L_{i_k}}{2}\|g_k\|^2$$

$$= f_{i_k}(w_k) - \left(\eta_k - \frac{\eta_k^2 L_{i_k}}{2}\right)\|g_k\|^2,$$

which can be rewritten as

$$f_{i_k}(w_k - \eta_k g_k) \leq p_k(\eta_k), \quad \text{with } p_k(\eta) := f_{i_k}(w_k) - \left(\eta - \frac{\eta^2 L_{i_k}}{2}\right)\|g_k\|^2. \tag{11}$$

Note that (11) is valid for any $\eta$. Let us rewrite (3) as

$$f_{i_k}(w_k - \eta_k g_k) \leq q_k(\eta_k), \quad \text{with } q_k(\eta) := C_k - c\eta\|g_k\|^2.$$

Now, the backtracking procedure in (3) admits two possible output:
Case 1: $l_k = 0$. In this case, we have $\eta_k = \eta_{k,0}$ and thus directly $\eta_k \in [\bar{\eta}^{\min}, \eta^{\max}]$.
Case 2: $l_k > 0$. In this case, we have $\eta_k < \eta_{k,0}$ with $f_{i_k}(w_k - \frac{\eta_k}{\delta}g_k) > q_k(\frac{\eta_k}{\delta})$. Then, we have that $q_k(\frac{\eta_k}{\delta}) \leq p_k(\frac{\eta_k}{\delta})$ because $q_k(\frac{\eta_k}{\delta}) > p_k(\frac{\eta_k}{\delta})$ would lead to a contradiction. In fact

$$f_{i_k}\left(w_k - \frac{\eta_k}{\delta}g_k\right) > q_k\left(\frac{\eta_k}{\delta}\right) > p_k\left(\frac{\eta_k}{\delta}\right) \geq f_{i_k}\left(w_k - \frac{\eta_k}{\delta}g_k\right)$$

is false. Thus, it has to be $q_k(\frac{\eta_k}{\delta}) \leq p_k(\frac{\eta_k}{\delta})$, from which we get that

$$f_{i_k}(w_k) - c\frac{\eta_k}{\delta}\|g_k\|^2 \leq C_k - c\frac{\eta_k}{\delta}\|g_k\|^2 \leq f_{i_k}(w_k) - \left(\frac{\eta_k}{\delta} - \frac{\eta_k^2 L_{i_k}}{2\delta^2}\right)\|g_k\|^2$$

and consequently

$$-c \leq -\left(1 - \frac{\eta_k L_{i_k}}{2\delta}\right) \Leftrightarrow \eta_k \geq \frac{2\delta(1-c)}{L_{i_k}},$$

which leads to (10).  □

One of the challenges of convergence theorems for nonmonotone line search methods is to prove that the sequence of the nonmonotone terms $\{C_k\}$ converges to $f(w^*)$. In order to achieve this, in Lemma 3 below we prove that $C_k$ and $C_{k-1}$ are lower-bounded by $f_{i_k}(w^*)$. Before that, we establish the following auxiliary result.

**Lemma 2.** *From the definition of $Q_k$ in (3), it follows*

$$1 \leq \xi Q_k + 1 \leq \frac{1}{1 - \xi}. \tag{12}$$

*and*

$$\frac{\xi Q_k}{\xi Q_k + 1} = \leq \xi \tag{13}$$

*Proof.* From the definition of $Q_{k+1}$ we have

$$1 \leq \xi Q_k + 1 =: Q_{k+1} = 1 + \sum_{j=0}^{k} \xi^{j+1} \leq \sum_{j=0}^{\infty} \xi^j = \frac{1}{1-\xi}.$$

which implies (12). Thus we have

$$\frac{\xi Q_k}{\xi Q_k + 1} = \frac{\xi Q_k + 1 - 1}{\xi Q_k + 1} = 1 - \frac{1}{\xi Q_k + 1} \leq 1 - \frac{1}{\frac{1}{1-\xi}} = \xi.$$

which implies (13) and concludes the proof. $\qquad\square$

We say that $f$ satisfies interpolations if given $w^* \in \underset{w \in \mathbb{R}^n}{\operatorname{argmin}} f(w)$, then $w^* \in \underset{w \in \mathbb{R}^n}{\operatorname{argmin}} f_i(w) \ \forall 1 \leq i \leq M$.

**Lemma 3.** *Let $C_k$ be defined in* (3). *Assuming interpolation, the following bounds hold for all $k \in \mathbb{N}$,*

$$C_k - f_{i_k}(w^*) \geq 0 \quad \forall k \tag{14}$$

*and*

$$C_{k-1} - f_{i_k}(w^*) \geq 0 \quad \forall k. \tag{15}$$

*Proof.* We will prove both statements by induction, starting with (14). For $k = 0$, interpolation yields $f_{i_0}(w_0) \geq f_{i_0}(w^*)$. Assuming now that the statement is valid for $k - 1 \in \mathbb{N}_0$ (i.e., $C_{k-1} - f_{k-1}(w^*) \geq 0$), let us prove that it is valid also for $k$. If $\tilde{C}_k > f_{i_k}(w_k)$, we have

$$C_k - f_{i_k}(w^*) = \frac{\xi Q_k}{\xi Q_k + 1} C_{k-1} + \frac{1}{\xi Q_k + 1} f_{i_k}(w_k) - f_{i_k}(w^*) \geq \frac{\xi Q_k}{\xi Q_k + 1} f_{i_{k-1}}(w^*) + \frac{1}{\xi Q_k + 1} f_{i_k}(w^*) - f_{i_k}(w^*) = 0,$$

where we used the induction hypothesis and the fact that $f_{i_{k-1}}(w^*) = f_{i_k}(w^*)$, thanks to interpolation. If $\tilde{C}_k \leq f_{i_k}(w_k)$, from interpolation we have

$$C_k - f_{i_k}(w^*) = f_{i_k}(w_k) - f_{i_k}(w^*) \geq 0.$$

The last two inequalities prove (14). We can follow a similar path for (15). For $k = 0$, from the definition of $C_{-1}$, we have $C_{-1} - f_{i_0}(w^*) = f_{i_0}(w_0) - f_{i_0}(w^*) \geq 0$. Assume now that the statement is valid for $k - 1 \in \mathbb{N}_0$ (i.e., $C_{k-2} - f_{k-1}(w^*) \geq 0$). Then, if $\tilde{C}_{k-1} > f_{i_{k-1}}(w_{k-1})$, we have

$$\begin{aligned} C_{k-1} - f_{i_k}(w^*) &= \frac{\xi Q_{k-1}}{\xi Q_{k-1} + 1} C_{k-2} + \frac{1}{\xi Q_{k-1} + 1} f_{i_{k-1}}(w_k) - f_{i_k}(w^*) \\ &\geq \frac{\xi Q_{k-1}}{\xi Q_{k-1} + 1} f_{i_{k-1}}(w^*) + \frac{1}{\xi Q_{k-1} + 1} f_{i_{k-1}}(w^*) - f_{i_k}(w^*) = 0, \end{aligned}$$

where we used again the induction hypothesis and the fact that $f_{i_{k-1}}(w^*) = f_{i_k}(w^*)$, thanks to interpolation. If $\tilde{C}_{k-1} \leq f_{i_{k-1}}(w_{k-1})$, from interpolation we have

$$C_{k-1} - f_{i_k}(w^*) = f_{i_{k-1}}(w_k) - f_{i_{k-1}}(w^*) \geq 0,$$

which concludes the proof. $\qquad\square$

The following Lemma shows the importance of the interpolation property. A similar result can be obtained by replacing the $L_{i_k}$-smoothness assumption with the line search condition (2) or (3) (see the proof of Theorem 1 below).

**Lemma 4.** *We assume interpolation and that $f_{i_k}$ are $L_{i_k}$-smooth. Then, we obtain*

$$\mathbb{E}_{i_k} \| \nabla f_{i_k}(w_k) \|^2 \leq L_{max} \left( f(w_k) - f(w^*) \right),$$

*where $L_{max} = \max_i L_i$.*

*Proof.* Let $w^* \in \underset{w \in \mathbb{R}^n}{\operatorname{argmin}} f(w)$ and $w_{i_k}^* \in \underset{w \in \mathbb{R}^n}{\operatorname{argmin}} f_{i_k}(w)$. From interpolation we have that $w^* \in \underset{w \in \mathbb{R}^n}{\operatorname{argmin}} f_{i_k}(w)$, which means that $f_{i_k}(w^*) = f_{i_k}(w_{i_k}^*)$. Thus, interpolation and $L_{i_k}$-smoothness of $f_{i_k}$ brings to

$$\| \nabla f_{i_k}(w_k) \|^2 \leq L_{i_k} \left( f_{i_k}(w_k) - f_{i_k}(w_{i_k}^*) \right) = L_{i_k} \left( f_{i_k}(w_k) - f_{i_k}(w^*) \right).$$

Now, by applying the conditional expectation $\mathbb{E}_{i_k}$ on the above inequality, we obtain

$$\mathbb{E}_{i_k} \| \nabla f_{i_k}(w_k) \|^2 \leq L_{max} \left( f(w_k) - f(w^*) \right).$$

$\qquad\square$

## B.1 Rate of Convergence for Strongly Convex Functions

In this subsection, we prove a linear rate of convergence in the case of a strongly convex function f.

**Theorem 1.** *Let $C_k$ and $\eta_k$ be defined in (3), with $\eta_{k,0}$ defined in (9). We assume interpolation, $f_{i_k}$ convex, $f$ $\mu$-strongly convex and $f_{i_k}$ $L_{i_k}$-Lipschitz smooth. If $c > \frac{1}{2}$ and $\xi < \frac{1}{\left(1 + \frac{\eta^{max}}{\eta^{min}(2c-1)}\right)}$, we have*

$$\mathbb{E}\left[\|w_{k+1} - w^*\|^2 + a(C_k - f(w^*))\right] \le d^k \left(\|w_0 - w^*\|^2 + a\left(f(w_0) - f(w^*)\right)\right),$$

*where $d := \max\left\{(1 - \eta^{min}\mu), b\right\} \in (0,1)$, $b := \left(1 + \frac{\eta^{max}}{ac}\right)\xi \in (0,1)$, $a := \eta^{min}\left(2 - \frac{1}{c}\right) > 0$ with $\eta^{min}$ as defined in Lemma 1.*

*Proof.* From (3) and interpolation we obtain

$$
\begin{aligned}
\|w_{k+1} - w^*\|^2 &= \|w_k - \eta_k \nabla f_{i_k}(w_k) - w^*\|^2 \\
&= \|w_k - w^*\|^2 - 2\eta_k\langle\nabla f_{i_k}(w_k), w_k - w^*\rangle + \eta_k^2\|\nabla f_{i_k}(w_k)\|^2 \\
&\le \|w_k - w^*\|^2 - 2\eta_k\langle\nabla f_{i_k}(w_k), w_k - w^*\rangle + \frac{\eta_k}{c}\left(C_k - f_{i_k}(w_{k+1})\right) \\
&\le \|w_k - w^*\|^2 - 2\eta_k\langle\nabla f_{i_k}(w_k), w_k - w^*\rangle + \frac{\eta_k}{c}\left(C_k - f_{i_k}(w^*)\right).
\end{aligned}
\tag{16}
$$

Let us distinguish 2 cases: either 1) $f_{i_k}(w_k) \ge \tilde{C}_k$ and then $C_k = f_{i_k}(w_k)$, or 2) $f_{i_k}(w_k) < \tilde{C}_k$ and then $C_k = \tilde{C}_k$. Let us first analyze case 1). With $c > \frac{1}{2}$ and $a = \eta^{min}\left(2 - \frac{1}{c}\right)$, from (16) we have

$$
\begin{aligned}
\|w_{k+1} - w^*\|^2 + a(C_k - f_{i_k}(w^*)) &\le \|w_k - w^*\|^2 - 2\eta_k\langle\nabla f_{i_k}(w_k), w_k - w^*\rangle + \left(a + \frac{\eta_k}{c}\right)(C_k - f_{i_k}(w^*)) \\
&= \|w_k - w^*\|^2 - 2\eta_k\langle\nabla f_{i_k}(w_k), w_k - w^*\rangle + \frac{\eta_k}{c}\left(f_{i_k}(w_k) - f_{i_k}(w^*)\right) \\
&\quad + a\left(f_{i_k}(w_k) - f_{i_k}(w^*)\right) \\
&\le \|w_k - w^*\|^2 + \eta^{min}\left[-2\langle\nabla f_{i_k}(w_k), w_k - w^*\rangle + \frac{1}{c}\left(f_{i_k}(w_k) - f_{i_k}(w^*)\right)\right] \\
&\quad + a\left(f_{i_k}(w_k) - f_{i_k}(w^*)\right) \\
&= \|w_k - w^*\|^2 + \eta^{min}\left[-2\langle\nabla f_{i_k}(w_k), w_k - w^*\rangle + \left(\frac{a}{\eta^{min}} + \frac{1}{c}\right)\left(f_{i_k}(w_k) - f_{i_k}(w^*)\right)\right] \\
&\le \|w_k - w^*\|^2 + 2\eta^{min}\left[-\langle\nabla f_{i_k}(w_k), w_k - w^*\rangle + \left(f_{i_k}(w_k) - f_{i_k}(w^*)\right)\right] \\
&\le \|w_k - w^*\|^2 + 2\eta^{min}\left[-\langle\nabla f_{i_k}(w_k), w_k - w^*\rangle + \left(f_{i_k}(w_k) - f_{i_k}(w^*)\right)\right] \\
&\quad + ab(C_{k-1} - f_{i_k}(w^*)),
\end{aligned}
$$

where the second inequality follows from $c > \frac{1}{2}$ and from the fact that the term between square brackets is negative, since $f_{i_k}$ is a convex function. The third inequality follows from the definition of $a$ and the last inequality follows from (15). In the following we are going to show that the same bound can also be achieved in case 2).

Let us now analyze case 2). From (16), (12) and (13), and again from $c > \frac{1}{2}$, convexity of $f_{i_k}$ and the definition of $a$, we have

$$
\begin{aligned}
\|w_{k+1} - w^*\|^2 + a(C_k - f_{i_k}(w^*)) &\le \|w_k - w^*\|^2 - 2\eta_k\langle\nabla f_{i_k}(w_k), w_k - w^*\rangle + \left(a + \frac{\eta_k}{c}\right)\left(\tilde{C}_k - f_{i_k}(w^*)\right) \\
&= \|w_k - w^*\|^2 - 2\eta_k\langle\nabla f_{i_k}(w_k), w_k - w^*\rangle + \left(a + \frac{\eta_k}{c}\right)\frac{1}{\xi Q_k + 1}\left(f_{i_k}(w_k) - f_{i_k}(w^*)\right) \\
&\quad + \left(a + \frac{\eta_k}{c}\right)\frac{\xi Q_k}{\xi Q_k + 1}\left(C_{k-1} - f_{i_k}(w^*)\right) \\
&\le \|w_k - w^*\|^2 - 2\eta_k\langle\nabla f_{i_k}(w_k), w_k - w^*\rangle + 2\eta^{min}\left(f_{i_k}(w_k) - f_{i_k}(w^*)\right) \\
&\quad + \left(a + \frac{\eta_k}{c}\right)\xi\left(C_{k-1} - f_{i_k}(w^*)\right) \\
&\le \|w_k - w^*\|^2 + 2\eta^{min}\left[-\langle\nabla f_{i_k}(w_k), w_k - w^*\rangle + \left(f_{i_k}(w_k) - f_{i_k}(w^*)\right)\right] \\
&\quad + \left(a + \frac{\eta_k}{c}\right)\xi\left(C_{k-1} - f_{i_k}(w^*)\right) \\
&\le \|w_k - w^*\|^2 + 2\eta^{min}\left[-\langle\nabla f_{i_k}(w_k), w_k - w^*\rangle + \left(f_{i_k}(w_k) - f_{i_k}(w^*)\right)\right] \\
&\quad + a\left(1 + \frac{\eta^{max}}{ac}\right)\xi\left(C_{k-1} - f_{i_k}(w^*)\right),
\end{aligned}
$$

where the fourth inequality follows from (10). By defining $b = \left(1 + \frac{\eta^{\max}}{ac}\right)\xi$ we can conclude that the same bound holds in both cases 1) and 2). Let us now show that $b < 1$. Under the assumption that $c > \frac{1}{2}$ and $\xi < \frac{1}{\left(1 + \frac{\eta^{\max}}{\eta^{\min}(2c-1)}\right)}$ we have

$$b = \left(1 + \frac{\eta^{\max}}{ac}\right)\xi = \left(1 + \frac{\eta^{\max}}{\eta^{\min}\left(2 - \frac{1}{c}\right)c}\right)\xi = \left(1 + \frac{\eta^{\max}}{\eta^{\min}(2c-1)}\right)\xi < 1.$$

Now, by taking expectation w.r.t. $i_k$ on the common bound achieved in both cases 1) and 2), using $\mathbb{E}_{i_k}[\nabla f_{i_k}(w_k)] = \nabla f(w_k)$, and applying strong convexity of $f$ we obtain

$$\begin{aligned}
\mathbb{E}_{i_k}\left[\|w_{k+1} - w^*\|^2\right] + a(\mathbb{E}_{i_k}[C_k] - f(w^*)) &\leq \|w_k - w^*\|^2 + 2\eta^{\min}\left[-\langle\nabla f(w_k), w_k - w^*\rangle + f(w_k) - f(w^*)\right] \\
&\quad + ab\left(C_{k-1} - f(w^*)\right) \\
&\leq \|w_k - w^*\|^2 - 2\eta^{\min}\frac{\mu}{2}\|w_k - w^*\|^2 + ab\left(C_{k-1} - f(w^*)\right) \\
&= (1 - \eta^{\min}\mu)\|w_k - w^*\|^2 + ab\left(C_{k-1} - f(w^*)\right) \\
&\leq d\left(\|w_k - w^*\|^2 + a\left(C_{k-1} - f(w^*)\right)\right),
\end{aligned}$$

where $d := \max\left\{(1 - \eta^{\min}\mu), b\right\}$ and in the first inequality we used the fact that $C_{k-1}$ does not depend on $i_k$. Taking the total expectation gives

$$\mathbb{E}\left[\|w_{k+1} - w^*\|^2 + a(C_k - f(w^*))\right] \leq d\,\mathbb{E}\left[\left(\|w_k - w^*\|^2 + a\left(C_{k-1} - f(w^*)\right)\right)\right].$$

At this point we can use the above inequality recursively, resulting in

$$\begin{aligned}
\mathbb{E}\left[\|w_{k+1} - w^*\|^2 + a(C_k - f(w^*))\right] &\leq d^k\,\mathbb{E}\left[\|w_0 - w^*\|^2 + a\left(C_{-1} - f(w^*)\right)\right] \\
&= d^k\left(\|w_0 - w^*\|^2 + a\left(f(w_0) - f(w^*)\right)\right),
\end{aligned}$$

where in the last equality we have used that $C_{-1} = f_{i_0}(w_0)$. $\qquad\square$

## B.2 Rate of Convergence for Convex Functions

In this subsection, we prove a $O(\frac{1}{k})$ rate of convergence in the case of a convex function.

**Theorem 2.** *Let $C_k$ and $\eta_k$ be defined in* (3), *with $\eta_{k,0}$ defined in* (9). *We assume interpolation, $f$ convex and $f_{i_k}$ $L_{i_k}$-Lipschitz smooth. Given a constant $a_1$ such that $0 < a_1 < \left(2 - \frac{1}{c}\right)$, if $c > \frac{1}{2}$ and $\xi < \frac{a_1}{2}$, we have*

$$\mathbb{E}\left[f(\bar{w}_k) - f(w^*)\right] \leq \frac{d_1}{k}\left(\frac{1}{\eta^{\min}}\|w_0 - w^*\|^2 + a_1\left(f(w_0) - f(w^*)\right)\right),$$

*where $\bar{w}_k = \frac{1}{k}\sum_{j=0}^k w_j$ and $d_1 := \frac{c}{c(2-a_1)-1} > 0$.*

*Proof.* From (3) and interpolation we obtain

$$\begin{aligned}
\|w_{k+1} - w^*\|^2 &= \|w_k - \eta_k\nabla f_{i_k}(w_k) - w^*\|^2 \\
&= \|w_k - w^*\|^2 - 2\eta_k\langle\nabla f_{i_k}(w_k), w_k - w^*\rangle + \eta_k^2\|\nabla f_{i_k}(w_k)\|^2 \\
&\leq \|w_k - w^*\|^2 - 2\eta_k\langle\nabla f_{i_k}(w_k), w_k - w^*\rangle + \frac{\eta_k}{c}\left(C_k - f_{i_k}(w_{k+1})\right) \\
&\leq \|w_k - w^*\|^2 - 2\eta_k\langle\nabla f_{i_k}(w_k), w_k - w^*\rangle + \frac{\eta_k}{c}\left(C_k - f_{i_k}(w^*)\right).
\end{aligned} \qquad (17)$$

Let us distinguish 2 cases: either 1) $f_{i_k}(w_k) \geq \tilde{C}_k$ and then $C_k = f_{i_k}(w_k)$, or 2) $f_{i_k}(w_k) < \tilde{C}_k$ and then $C_k = \tilde{C}_k$. Let us first analyze case 1). From (17) we have

$$\begin{aligned}
\|w_{k+1} - w^*\|^2 + \eta_k a_1(C_k - f_{i_k}(w^*)) &\leq \|w_k - w^*\|^2 - 2\eta_k\langle\nabla f_{i_k}(w_k), w_k - w^*\rangle + \left(\eta_k a_1 + \frac{\eta_k}{c}\right)(C_k - f_{i_k}(w^*)) \\
&= \|w_k - w^*\|^2 + \eta_k\left[-2\langle\nabla f_{i_k}(w_k), w_k - w^*\rangle + \left(a_1 + \frac{1}{c}\right)(f_{i_k}(w_k) - f_{i_k}(w^*))\right] \\
&\leq \|w_k - w^*\|^2 + \eta_k\left[-2\langle\nabla f_{i_k}(w_k), w_k - w^*\rangle + \left(a_1 + \frac{1}{c}\right)(f_{i_k}(w_k) - f_{i_k}(w^*))\right] \\
&\quad + \eta_k a_1 b_1(C_{k-1} - f_{i_k}(w^*)),
\end{aligned}$$

where the second inequality follows from (15) and $b_1 := \left(1 + \frac{1}{a_1 c}\right) \xi > 0$. The above bound will now be proven also for case 2). From (17) we have

$$\|w_{k+1} - w^*\|^2 + \eta_k a_1 (C_k - f_{i_k}(w^*)) \le \|w_k - w^*\|^2 - 2\eta_k \langle \nabla f_{i_k}(w_k), w_k - w^* \rangle + \left(\eta_k a_1 + \frac{\eta_k}{c}\right) \left(\tilde{C}_k - f_{i_k}(w^*)\right)$$

$$= \|w_k - w^*\|^2 - 2\eta_k \langle \nabla f_{i_k}(w_k), w_k - w^* \rangle + \left(\eta_k a_1 + \frac{\eta_k}{c}\right) \frac{1}{\xi Q_k + 1} (f_{i_k}(w_k) - f_{i_k}(w^*))$$

$$+ \left(\eta_k a_1 + \frac{\eta_k}{c}\right) \frac{\xi Q_k}{\xi Q_k + 1} (C_{k-1} - f_{i_k}(w^*))$$

$$\le \|w_k - w^*\|^2 + \eta_k \left[-2\langle \nabla f_{i_k}(w_k), w_k - w^* \rangle + \left(a_1 + \frac{1}{c}\right)(f_{i_k}(w_k) - f_{i_k}(w^*))\right]$$

$$+ \eta_k a_1 \left(1 + \frac{1}{a_1 c}\right) \xi (C_{k-1} - f_{i_k}(w^*)),$$

where the second inequality follows from (12) and (13). By defining $b_1 := \left(1 + \frac{1}{a_1 c}\right) \xi$ we can conclude that the same bound can be achieved in both cases 1) and 2). Let us now show that $b_1 < 1$. From the definition of $a_1$, we have $\frac{1}{c} < 2 - a_1$, thus, under the assumption $\xi < \frac{a_1}{2}$ it holds that

$$b_1 = \left(1 + \frac{1}{a_1 c}\right) \xi < \left(1 + \frac{2 - a_1}{a_1}\right) \xi = \frac{2}{a_1} \xi < 1.$$

From $b_1 < 1$, rearranging and dividing the bound found for both cases 1) and 2) by $2\eta_k$, we have the following

$$\langle \nabla f_{i_k}(w_k), w_k - w^* \rangle \le \frac{1}{2\eta_k} \left[\|w_k - w^*\|^2 - \|w_{k+1} - w^*\|^2\right] + \left(\frac{a_1}{2} + \frac{1}{2c}\right)(f_{i_k}(w_k) - f_{i_k}(w^*))$$

$$+ \frac{a_1}{2} \left[(C_{k-1} - f_{i_k}(w^*)) - (C_k - f_{i_k}(w^*))\right].$$

Now, by taking expectation w.r.t. $i_k$, we obtain

$$\langle \nabla f(w_k), w_k - w^* \rangle \le \mathbb{E}_{i_k} \left[\frac{1}{2\eta_k} \left(\|w_k - w^*\|^2 - \|w_{k+1} - w^*\|^2\right)\right] + \left(\frac{a_1}{2} + \frac{1}{2c}\right)(f(w_k) - f(w^*))$$

$$+ \frac{a_1}{2} \left((C_{k-1} - f(w^*)) - (\mathbb{E}_{i_k}[C_k] - f(w^*))\right).$$

By convexity of $f$ we have $(f(w_k) - f(w^*)) \le \langle \nabla f(w_k), w_k - w^* \rangle$, thus,

$$f(w_k) - f(w^*) \le \mathbb{E}_{i_k} \left[\frac{1}{2\eta_k} \left(\|w_k - w^*\|^2 - \|w_{k+1} - w^*\|^2\right)\right] + \left(\frac{a_1}{2} + \frac{1}{2c}\right)(f(w_k) - f(w^*))$$

$$+ \frac{a_1}{2} \left((C_{k-1} - f(w^*)) - (\mathbb{E}_{i_k}[C_k] - f(w^*))\right).$$

Moreover, noticing that $1 - \left(\frac{a_1}{2} + \frac{1}{2c}\right) > 0$ by the definition of $a_1$, we obtain

$$f(w_k) - f(w^*) \le \mathbb{E}_{i_k} \left[\frac{d_1}{\eta_k} \left(\|w_k - w^*\|^2 - \|w_{k+1} - w^*\|^2\right)\right]$$

$$+ d_1 a_1 \left((C_{k-1} - f(w^*)) - (\mathbb{E}_{i_k}[C_k] - f(w^*))\right),$$

where $d_1 := \frac{1}{2} \frac{1}{1 - \left(\frac{a_1}{2} + \frac{1}{2c}\right)} = \frac{c}{c(2 - a_1) - 1}$. Taking the total expectation and summing from $k$ to $0$

$$\mathbb{E}\left[\sum_{j=0}^{k} f(w_j) - f(w^*)\right] \le \mathbb{E}\left[\sum_{j=0}^{k} \frac{d_1}{\eta_j} \left(\|w_j - w^*\|^2 - \|w_{j+1} - w^*\|^2\right)\right]$$

$$+ d_1 a_1 \mathbb{E}\left[\sum_{j=0}^{k} (C_{j-1} - f(w^*)) - (C_j - f(w^*))\right].$$

Recalling that $\bar{w}_k = \frac{1}{k} \sum_{j=0}^{k} w_j$, it follows from Jensen's inequality that

$$\mathbb{E}\left[f(\bar{w}_k) - f(w^*)\right] \le \mathbb{E}\left[\frac{1}{k} \sum_{j=0}^{k} f(w_j) - f(w^*)\right] = \frac{1}{k} \mathbb{E}\left[\sum_{j=0}^{k} (f(w_j) - f(w^*))\right].$$

Now, putting together the last two inequalities and defining $\Delta_j := \|w_j - w^*\|^2$ and $\Gamma_j := C_j - f(w^*)$, we have

$$\mathbb{E}\left[f(\bar{w}_k) - f(w^*)\right] \leq \frac{1}{k}d_1\,\mathbb{E}\left[\sum_{j=0}^{k}\frac{1}{\eta_j}\left(\Delta_j - \Delta_{j+1}\right)\right] + \frac{1}{k}d_1 a_1\,\mathbb{E}\left[\sum_{j=0}^{k}\Gamma_{j-1} - \Gamma_j\right]$$

$$\leq \frac{1}{k}\frac{d_1}{\eta^{\min}}\,\mathbb{E}\left[\Delta_0 - \Delta_k\right] + \frac{1}{k}d_1 a_1\,\mathbb{E}\left[\Gamma_{-1} - \Gamma_k\right]$$

$$\leq \frac{1}{k}\frac{d_1}{\eta^{\min}}\Delta_0 + \frac{1}{k}d_1 a_1\Gamma_{-1} = \frac{d_1}{k}\left(\frac{1}{\eta^{\min}}\|w_0 - w^*\|^2 + a_1\left(f(w_0) - f(w^*)\right)\right),$$

where the second inequality follows from the fact that $\eta_k \geq \eta^{\min}$ and from simplification of the telescopic sum and the last inequality follows from $\Delta_k, \Gamma_k > 0$ (because of (14) in case of $\Gamma_k$). $\qquad\square$

## B.3 Rate of Convergence for Functions Satisfying the PL Condition

In this subsection, we prove a linear convergence rate in the case of $f$ satisfying a PL condition. We say that a function $f : \mathbb{R}^n \to \mathbb{R}$ satisfies the PL condition if there exists $\mu > 0$ such that, $\forall w \in \mathbb{R}^n : \|\nabla f(w)\|^2 \geq 2\mu(f(w) - f(w^*))$. From $f_{i_k}$ being $L_{i_k}$-Lipschitz smooth $\forall i_k$, it follows that $f$ is also Lipschitz smooth. Let us call $L$ the Lipschitz smoothness constant of $f$ and let us note that $L \leq \frac{1}{M}\sum_{i=1}^{M}L_i \leq L_{max}$,

**Theorem 3.** *Let $C_k$ and $\eta_k$ be defined in* (3), *with $\eta_{k,0}$ defined in* (9). *We assume interpolation, the PL condition on $f$ and that $f_{i_k}$ are $L_{i_k}$-Lipschitz smooth. Given $0 < a_2 := \frac{4\mu c(1-c) - L_{max}}{4\delta c(1-c)} + \frac{1}{2\eta^{max}}$ and assuming $\frac{2\delta(1-c)}{L_{max}} < \bar{\eta}^{min}, \eta^{max} < \frac{2\delta c(1-c)}{L_{max} - 4\mu c(1-c)}, \frac{L_{max}}{4\mu} < c < 1$ and $\xi < \frac{a_2 c}{a_2 c + L_{max}}$, we have*

$$\mathbb{E}\left[f(w_{k+1}) - f(w^*) + a_2\eta^{max}(C_k - f(w^*))\right] \leq d_2^k\left(1 + a_2\eta^{max}\right)\left(f(w_0) - f(w^*)\right)$$

*where $d_2 := \min\{\nu, b_2\} \in (0,1)$, $\nu := \eta^{max}\left(\frac{L_{max} - 4\mu c(1-c)}{2\delta c(1-c)} + a_2\right) \in (0,1)$, $b_2 := \left(1 + \frac{L_{max}}{a_2 c}\right)\xi \in (0,1)$.*

*Proof.* From the smoothness of $f$ we obtain

$$f(w_{k+1}) \leq f(w_k) + \langle\nabla f(w_k), w_{k+1} - w_k\rangle + \frac{L}{2}\|w_{k+1} - w_k\|^2$$

$$= f(w_k) + \eta_k\langle\nabla f(w_k), \nabla f_{i_k}(w_k)\rangle + \frac{L\eta_k^2}{2}\|\nabla f_{i_k}(w_k)\|^2$$

We then rearrange, sum $a_2(C_k - f_{i_k}(w^*))$ on both sides and use (3), to obtain

$$\frac{f(w_{k+1}) - f(w_k)}{\eta_k} + a_2(C_k - f_{i_k}(w^*)) \leq -\langle\nabla f(w_k), \nabla f_{i_k}(w_k)\rangle + \frac{L\eta_k}{2}\|\nabla f_{i_k}(w_k)\|^2$$

$$+ a_2(C_k - f_{i_k}(w^*)) \qquad (18)$$

$$\leq -\langle\nabla f(w_k), \nabla f_{i_k}(w_k)\rangle + \left(\frac{L}{2c} + a_2\right)(C_k - f_{i_k}(w^*)),$$

Let us now distinguish 2 cases: either 1) $f_{i_k}(w_k) \geq \tilde{C}_k$ and then $C_k = f_{i_k}(w_k)$, or 2) $f_{i_k}(w_k) < \tilde{C}_k$ and then $C_k = \tilde{C}_k$. Let us first analyze case 1). Assuming $a_2, b_2 > 0$, from (18) we have

$$\frac{f(w_{k+1}) - f(w_k)}{\eta_k} + a_2(C_k - f_{i_k}(w^*)) \leq -\langle\nabla f(w_k), \nabla f_{i_k}(w_k)\rangle + \left(\frac{L}{2c} + a_2\right)(f_{i_k}(w_k) - f_{i_k}(w^*))$$

$$\leq -\langle\nabla f(w_k), \nabla f_{i_k}(w_k)\rangle + \left(\frac{L}{2c} + a_2\right)(f_{i_k}(w_k) - f_{i_k}(w^*))$$

$$+ a_2 b_2\left(C_{k-1} - f_{i_k}(w^*)\right)$$

where the last inequality follows from (15). The above bound will be now proven also for case 2). From (18) we have

$$\frac{f(w_{k+1}) - f(w_k)}{\eta_k} + a_2(C_k - f_{i_k}(w^*)) \leq -\langle\nabla f(w_k), \nabla f_{i_k}(w_k)\rangle + \left(\frac{L}{2c} + a_2\right)\left(\tilde{C}_k - f_{i_k}(w^*)\right)$$

$$= -\langle\nabla f(w_k), \nabla f_{i_k}(w_k)\rangle + \left(\frac{L}{2c} + a_2\right)\frac{1}{\xi Q_k + 1}\left(f_{i_k}(w_k) - f_{i_k}(w^*)\right)$$

$$+ \left(\frac{L}{2c} + a_2\right)\frac{\xi Q_k}{\xi Q_k + 1}\left(C_{k-1} - f_{i_k}(w^*)\right)$$

$$\leq -\langle\nabla f(w_k), \nabla f_{i_k}(w_k)\rangle + \left(\frac{L}{2c} + a_2\right)\left(f_{i_k}(w_k) - f_{i_k}(w^*)\right)$$

$$+ \left(\frac{L_{max}}{2c} + a_2\right)\xi\left(C_{k-1} - f_{i_k}(w^*)\right)$$

where the second inequality follows from (12) and (13) and $L \leq L_{max}$. By defining $b_2 := \left(1 + \frac{L_{max}}{a_2 c}\right) \xi$ we can conclude that the same bound can be achieved in both cases 1) and 2). Now, by taking expectation w.r.t. $i_k$ on the common bound achieved in both cases 1) and 2), and applying PL condition, we have

$$\mathbb{E}_{i_k} \left[ \frac{f(w_{k+1}) - f(w_k)}{\eta_k} + a_2(C_k - f(w^*)) \right] \leq -\langle \nabla f(w_k), \nabla f(w_k) \rangle + \left(\frac{L}{2c} + a_2\right)(f(w_k) - f(w^*))$$
$$+ a_2 b_2 \left(C_{k-1} - f(w^*)\right)$$
$$\leq \left(\frac{L}{2c} + a_2 - 2\mu\right)(f(w_k) - f(w^*))$$
$$+ a_2 b_2 \left(C_{k-1} - f(w^*)\right),$$

where in the first inequality we used the fact that $C_{k-1}$ does not depend on $i_k$ and $\mathbb{E}_{i_k} f_{i_k}(w^*) = f(w^*) = \mathbb{E}_{i_k} f(w^*)$. Thus,

$$\mathbb{E}_{i_k} \left[ \frac{f(w_{k+1}) - f(w^*)}{\eta_k} + a_2(C_k - f(w^*)) \right] \leq \mathbb{E}_{i_k} \left[ \frac{f(w_k) - f(w^*)}{\eta_k} \right] + \left(\frac{L}{2c} + a_2 - 2\mu\right)(f(w_k) - f(w^*))$$
$$+ a_2 b_2 \left(C_{k-1} - f(w^*)\right)$$
$$\leq \left(\frac{1}{\eta^{\min}} + \frac{L}{2c} + a_2 - 2\mu\right)(f(w_k) - f(w^*))$$
$$+ a_2 b_2 \left(C_{k-1} - f(w^*)\right)$$

Using $\eta_k \leq \eta^{\max}$, $L \leq L_{max}$ and taking the total expectation we obtain

$$\mathbb{E}\left[f(w_{k+1}) - f(w^*) + a_2\eta^{\max}(C_k - f(w^*))\right] \leq \eta^{\max}\left(\frac{1}{\eta^{\min}} + \frac{L_{max}}{2c} + a_2 - 2\mu\right) \mathbb{E}\left[f(w_k) - f(w^*)\right] \tag{19}$$
$$+ a_2\eta^{\max} b_2 \, \mathbb{E}\left[C_{k-1} - f(w^*)\right]$$

Defining $\nu := \eta^{\max}\left(\frac{1}{\eta^{\min}} + \frac{L_{max}}{2c} + a_2 - 2\mu\right)$, let us now show that $0 < \nu < 1$. From the assumption $\bar{\eta}^{\min} > \frac{2\delta(1-c)}{L_{max}}$, we have $\eta^{\min} = \min\left\{\frac{2\delta(1-c)}{L_{max}}, \bar{\eta}^{\min}\right\} = \frac{2\delta(1-c)}{L_{max}}$, and then

$$\nu = \eta^{\max}\left(\frac{L_{max}}{2\delta(1-c)} + \frac{L_{max}}{2c} - 2\mu + a_2\right) = \eta^{\max}\left(\frac{L_{max}}{2\delta c(1-c)} - 2\mu + a_2\right) = \eta^{\max}\left(\frac{L_{max} - 4\mu c(1-c)}{2\delta c(1-c)} + a_2\right).$$

Let $a_2 := \frac{4\mu c(1-c) - L_{max}}{4\delta c(1-c)} + \frac{1}{2\eta^{\max}}$, under the assumption $\eta^{\max} < \frac{2\delta c(1-c)}{L_{max} - 4\mu c(1-c)}$, we have

$$a_2 = \frac{4\mu c(1-c) - L_{max}}{4\delta c(1-c)} + \frac{1}{2\eta^{\max}} > \frac{4\mu c(1-c) - L_{max}}{4\delta c(1-c)} + \frac{L_{max} - 4\mu c(1-c)}{4\delta c(1-c)} = 0,$$

and thus $a_2 > 0$. Let us now use substitute $a_2$ in the definition of $\nu$, to obtain

$$\nu = \eta^{\max}\left(\frac{L_{max} - 4\mu c(1-c)}{2\delta c(1-c)} + a_2\right) = \eta^{\max}\left(\frac{L_{max} - 4\mu c(1-c)}{4\delta c(1-c)}\right) + \frac{1}{2} \tag{20}$$

Again from $\eta^{\max} < \frac{2\delta c(1-c)}{L_{max} - 4\mu c(1-c)}$ and (20), we obtain $\nu < \frac{1}{2} + \frac{1}{2} = 1$. Moreover, $L_{max} - 4\mu c(1-c) > 0$ because it is a quadratic polynomial in $c$, whose $\Delta < 0$ since of $\mu < L_{max}$. Thus, together with $\eta^{\max} > 0$ and $4\delta c(1-c) > 0$, from (20) we achieve $\nu > 0$. Regarding $b_2$, we have $b_2 > 0$ because $a_2 > 0$. Moreover, by assuming $\xi < \frac{a_2 c}{a_2 c + L_{max}}$, it also follows that

$$b_2 := \left(1 + \frac{L_{max}}{a_2 c}\right) \xi = \frac{a_2 c + L_{max}}{a_2 c} \xi < 1.$$

At this point, we need to ensure that $\frac{2\delta(1-c)}{L_{max}} < \bar{\eta}^{\min} < \eta^{\max} < \frac{2\delta c(1-c)}{L_{max} - 4\mu c(1-c)}$ or equivalently

$$0 < \frac{1}{L_{max}}\left(-1 + \frac{c\, L_{max}}{L_{max} - 4\mu c(1-c)}\right)$$
$$= \frac{1}{L_{max}}\left(\frac{c\, L_{max} - L_{max} + 4\mu c(1-c)}{L_{max} - 4\mu c(1-c)}\right)$$
$$= \frac{1}{L_{max}}\left(\frac{-4\mu c^2 + (4\mu + L_{max})c - L_{max}}{4\mu c^2 - 4\mu c + L_{max}}\right).$$

In particular, we can solve the inequality for $c$ and find that the numerator is positive for $\frac{L_{max}}{4\mu} < c < 1$ and that the denominator is always positive since $\mu < L_{max}$ (as above).

To conclude, we can use (19) recursively from $k$ to 0 and obtain

$$\mathbb{E}\left[f(w_{k+1}) - f(w^*) + a_2\eta^{\max}(C_k - f(w^*))\right] \leq d_2^k \, \mathbb{E}\left[f(w_0) - f(w^*) + a_2\eta^{\max}(C_{-1} - f(w^*))\right]$$
$$= d_2^k \left(1 + a_2\eta^{\max}\right)(f(w_0) - f(w^*))$$

where $d_2 := \min\{\nu, b_2\} < 1$ and in the last equality we have used that $C_{-1} = f_{i_0}(w_0)$. $\qquad \square$

## B.4 Common Lemmas

**Lemma 5.** *Let $f$ be $L$-Lipschitz smooth. Then*

$$f(y) \le f(x) + \nabla f(x)^T (y - x) + \frac{L}{2} \|y - x\|^2. \tag{21}$$

*Proof.* From the mean value theorem and differentiability of $f$ we have

$$
\begin{aligned}
f(y) &= f(x) + \int_0^1 \nabla f((1-t)x + ty)^T (y - x)\, dt \\
&= f(x) + \int_0^1 \nabla f((1-t)x + ty)^T (y - x) - \nabla f(x)^T (y - x)\, dt + \nabla f(x)^T (y - x) \\
&\le f(x) + \int_0^1 \|\nabla f((1-t)x + ty) - \nabla f(x)\| \cdot \|y - x\|\, dt + \nabla f(x)^T (y - x) \\
&\le f_{i_k}(x) + \int_0^1 L\|t(y-x)\| \cdot \|y - x\|\, dt + \nabla f(x)^T (y - x) \\
&= f(x) + L\|y - x\|^2 \cdot \frac{t^2}{2}\Big|_0^1 + \nabla f(x)^T (y - x) \\
&= f(x) + \nabla f(x)^T (y - x) + \frac{L}{2} \|y - x\|^2,
\end{aligned}
$$

where the second inequality follows from the Lipschitz continuity of $\nabla f$. □

**Lemma 6.** *Let $f$ be $L$-Lipschitz smooth and strongly convex. Then*

$$f(x) - f(x^*) \le \frac{1}{2\mu} \|\nabla f(x)\|^2. \tag{22}$$

*Proof.* From the strong convexity of $f$ we have

$$f(y) \ge r(x,y) := f(x) + \nabla f(x)^T (y - x) + \frac{\mu}{2} \|y - x\|^2.$$

We now minimize both sides of the above inequality w.r.t. $y$. In particular, we differentiate $r(x,y)$ w.r.t. $y$ and obtain

$$\frac{\partial\, r(x,y)}{\partial y} = \nabla f(x) + \mu(y - x) = 0 \Leftrightarrow y^* = x - \frac{1}{\mu} \nabla f(x)$$

which means that

$$r(x, y^*) = f(x) - \frac{1}{\mu} \|\nabla f(x)\|^2 + \frac{1}{2\mu} \|\nabla f(x)\|" = f(x) - \frac{1}{2\mu} \|\nabla f(x)\|^2.$$

Thus, since $\min_y f(y) \ge \min_y r(x,y)$, we can conclude that

$$f(x^*) \ge f(x) - \frac{1}{2\mu} \|\nabla f(x)\|^2. \tag{23}$$

□

**Lemma 7.** *Let $f$ be $L$-Lipschitz smooth. Then*

$$\frac{1}{2L} \|\nabla f(x)\|^2 \le f(x) - f(x^*). \tag{24}$$

*Proof.* We can repeat the same argument of Lemma 6 in the following inequality (obtained applying Lemma 5)

$$f(y) \le f(x) + \nabla f(x)^T (y - x) + \frac{L}{2} \|y - x\|^2,$$

and get that

$$f(x^*) \le f(x) - \frac{1}{2L} \|\nabla f(x)\|^2,$$

which concludes the proof. □

### B.5 The Polyak Step Size is Bounded

In this subsection, we show that the Polyak step size (7) is bounded. In particular, this step is capped at $\eta^{\max} > 0$, so it is bounded from above by $\eta^{\max}$. By definition of (7), interpolation and Lemma 7 applied on $f_{i_k}$ we get

$$\eta_{k,0} \geq \frac{f_{i_k}(w_k) - f_{i_k}^*}{c_p || \nabla f_{i_k}(w_k)||^2} \geq \frac{\frac{1}{2L_{i_k}} || \nabla f_{i_k}(w_k)||^2}{c_p || \nabla f_{i_k}(w_k)||^2} \geq \frac{1}{2c_p \, L_{max}}.$$

## C  Experimental Details

The `PyTorch` [Paszke et al., 2019] code to reproduce our results can be found at `https://github.com/leonardogalli91/PoNoS`. Also, PoNoS is there available as a `torch.optim.Opimizer`. Experiments are conducted on a machine with an NVIDIA A100 PCIe GPU with 40 GB of memory.
Problems (dataset, model):

1. MNIST, MLP (1 hidden-layer multi-layer perceptron of width 1000) [LeCun et al., 1998, Luo et al., 2019];
2. CIFAR10, ResNet-34 [Krizhevsky and Hinton, 2009, He et al., 2016];
3. CIFAR10, DenseNet-121 [Krizhevsky and Hinton, 2009, Huang et al., 2017];
4. CIFAR100, ResNet-34 [Krizhevsky and Hinton, 2009, He et al., 2016];
5. CIFAR100, DenseNet-121 [Krizhevsky and Hinton, 2009, Huang et al., 2017];
6. Fashion MNIST, EFFicientnet-B1 [Xiao et al., 2017, Tan and Le, 2019];
7. SVHN, WideResNet [Netzer et al., 2011, Zagoruyko and Komodakis, 2016];
8. mushrooms, RBF-kernel model [Chang and Lin, 2011];
9. rcv1, RBF-kernel model [Chang and Lin, 2011];
10. ijcnn, RBF-kernel model [Chang and Lin, 2011];
11. w8a, RBF-kernel model [Chang and Lin, 2011];
12. PTB, Transformer Encoder [Marcus et al., 1993, Vaswani et al., 2017];
13. Wikitext2, Transformer-XL [Dai et al., 2019, Merity et al., 2017].

All the datasets can be freely obtained respectively through the `pytorch` package (image classification), or from the LIBSVM repository [Chang and Lin, 2011] (binary classification), or from the websites `http://www.fit.vutbr.cz/~imikolov/rnnlm/simple-examples.tgz` (PTB) and `https://s3.amazonaws.com/research.metamind.io/wikitext/wikitext-2-v1.zip` (Wikitext2). In Table 1, we report a few information concerning the problems above and the hyper-parameters employed by the algorithms on that problem. In order by column, we report the number of parameters of the model $n$, the number of instances of the train set $M$, the batch size employed by the algorithms $b$, the number of iterations (# mini-batches) in each epoch $\frac{M}{b}$, the number of instances in the test set, the max amount of epochs and in brackets the corresponding max number of iterations. The combinations of dataset/model have been replaced by the corresponding number in the above listing.

The numerical results report various measures:

- train loss: the full-batch loss on the training set;
- test accuracy: accuracy on the test set;
- average step size: average of all the mini-batch step sizes within the epoch;
- initial step size: average of all the mini-batch initial step sizes within the epoch;
- gradient norm: average of all the mini-batch gradient norms within the epoch;
- # backtracks: the total number of backtracking steps required within the epoch;
- runtime: wall clock time of each epoch.

Many of the plots in this paper have been created by averaging 5 runs that differ from each other only on the random seed randomizing the algorithms. The shaded error bars in the plots correspond to the standard deviation from the mean of the 5 runs.

If not specified differently, the following is the setting of hyper-parameters used for Algorithm 1

$$\delta = 0.5, \quad \xi = 1, \quad \eta^{\max} = 10, \quad c = 0.5, \quad c_p = 0.1.$$

The performance of PoNoS is not sensitive to hyper-parameters. In fact, the same values work across experiments and there is no need to fine-tune them. Most of PoNoS's hyperparameters are set to standard values, while others are either inherited by recent papers or fixed by the theory:

| problem \ data | $n$ | $M$ | $b$ | $\frac{M}{b}$ | test size | max epochs (corr. iter.) |
|---|---|---|---|---|---|---|
| 1 | 535818 | 60000 | 128 | 469 | 10000 | 200 (93800) |
| 2 | 21282122 | 50000 | 128 | 391 | 10000 | 200 (78200) |
| 3 | 6956298 | 50000 | 128 | 391 | 10000 | 200 (78200) |
| 4 | 21328292 | 50000 | 128 | 391 | 10000 | 200 (78200) |
| 5 | 7048548 | 50000 | 128 | 391 | 10000 | 200 (78200) |
| 6 | 6525418 | 60000 | 128 | 469 | 10000 | 200 (93800) |
| 7 | 369498 | 73257 | 128 | 573 | 26032 | 200 (114600) |
| 8 | 112 | 6499 | 100 | 65 | 1625 | 35 (2275) |
| 9 | 22 | 39992 | 100 | 400 | 9998 | 35 (14000) |
| 10 | 47236 | 16194 | 100 | 162 | 4048 | 35 (5670) |
| 11 | 300 | 39799 | 100 | 398 | 9950 | 35 (13930) |
| 12 | 13828478 | 59712 | 64 | 933 | / | 100 (93300) |
| 13 | 30725904 | 7296 | 64 | 114 | / | 100 (11400) |

Table 1: Information on the problems and on some of the hyper-parameters of the algorithms. In order by column: number of parameters of the model $n$, number of instances of the train set $M$, batch size $b$, number of iterations (# mini-batches) in each epoch $\frac{M}{b}$, number of instances in the test set, max amount of epochs (corresponding max amount of iterations).

| method \ problem | 1 | 2 | 3 | 4 | 5 | 6 | 7 | 8 | 9 | 10 | 11 | 12 | 13 |
|---|---|---|---|---|---|---|---|---|---|---|---|---|---|
| SGD | 0.1 | 0.1 | 0.1 | 0.1 | 0.1 | 0.1 | 0.1 | 0.1 | 0.1 | 0.1 | 0.1 | 0.5 | 0.25 |
| Adam | $10^{-4}$ | $10^{-3}$ | $10^{-3}$ | $10^{-3}$ | $10^{-3}$ | $10^{-3}$ | $10^{-3}$ | 0.1 | 0.1 | 0.1 | 0.1 | $2.5 \cdot 10^{-4}$ | $2.5 \cdot 10^{-4}$ |

Table 2: Learning rates for SGD and Adam obtained through a grid-search procedure.

- $\delta = 0.5$, classical cut of the step [Nocedal and Wright, 2006]. SLS employs an unusual value of $0.9$ and this choice is connected to the use of their resetting technique (4). We checked the results of SLS with $\delta = 0.5$ and they indeed turned out to not be as good as with $\delta = 0.9$.

- $\xi = 1$, the fully nonmonotone version of Zhang and Hager [2004].

- $\eta^{\max} = 10$, very classical value [Vaswani et al., 2019, Loizou et al., 2021]. We conducted an ablation study in Section E.4, which shows that larger values of $\eta^{\max}$ do not have a remarkable impact on the results. These results show that PoNoS is more robust than SLS and SPS to these changes.

- $c = 0.5$, suggested by the theory. Both our theory and that of Vaswani et al. [2019] suggest employing $0.5$ for $c$, rather than the classical $0.1$ or lower [Nocedal and Wright, 2006]. The numerical results in Section E.1 support this choice. In particular, they show in Figure VII that $c = 0.1$ might bring PoNoS and its monotone counterpart to diverge.

- $c_p = 0.1$, half of the inherited value [Loizou et al., 2021]. In Loizou et al. [2021], the value $0.2$ was suggested for SPS, however in our case, the initial step size is not the final step since the backtracking procedure might reduce this value to its half (or less). For this reason, we decided to employ a step that is initially double that of SPS. The results show that PoNoS|0.1 is consistently better than PoNoS|0.2 (see Section E.5). Thus, we also checked whether SPS|0.1 would be consistently better than the original SPS, but this is not the case.

All the other optimizers were used with their default hyper-parameters and without any weight decay. The implementation of SGD and Adam is provided by `pytorch` [Paszke et al., 2019] and their learning rates were selected through a separate grid search on each of the problems. These values are collected in Table 2, where again the names are replaced by the above numbers. For the classification problems (1-11) the grid search was $\{10^{-1}, 10^{-2}, 10^{-3}, 10^{-4}\}$, while for training the language models we employed the larger grid $\{5, 2.5, 1\} \times \{10^{-1}, 10^{-2}, 10^{-3}, 10^{-4}\}$.

# D Plots Completing the Figures in the Main Paper

## D.1 Comparison between PoNoS and the state-of-the-art

See Figure I for the complete results relative to Figure 1 of the main paper. From Figure I we can additionally observe that on some problems (e.g., `cifar100|resnet34` and `cifar100|densenet121`) the step size yielded by the Polyak formula (6) is very big and grows very fast. The algorithm SPS controls this step by reducing it to (4). This step does not depend on the Polyak rule and it does not employ the local information provided by $f_{i_k}(w_k)$ and $\nabla f_{i_k}(w_k)$. Instead, it grows exponentially controlled by (4). On the other hand, the step size of PoNoS is always a scaled version of (6).

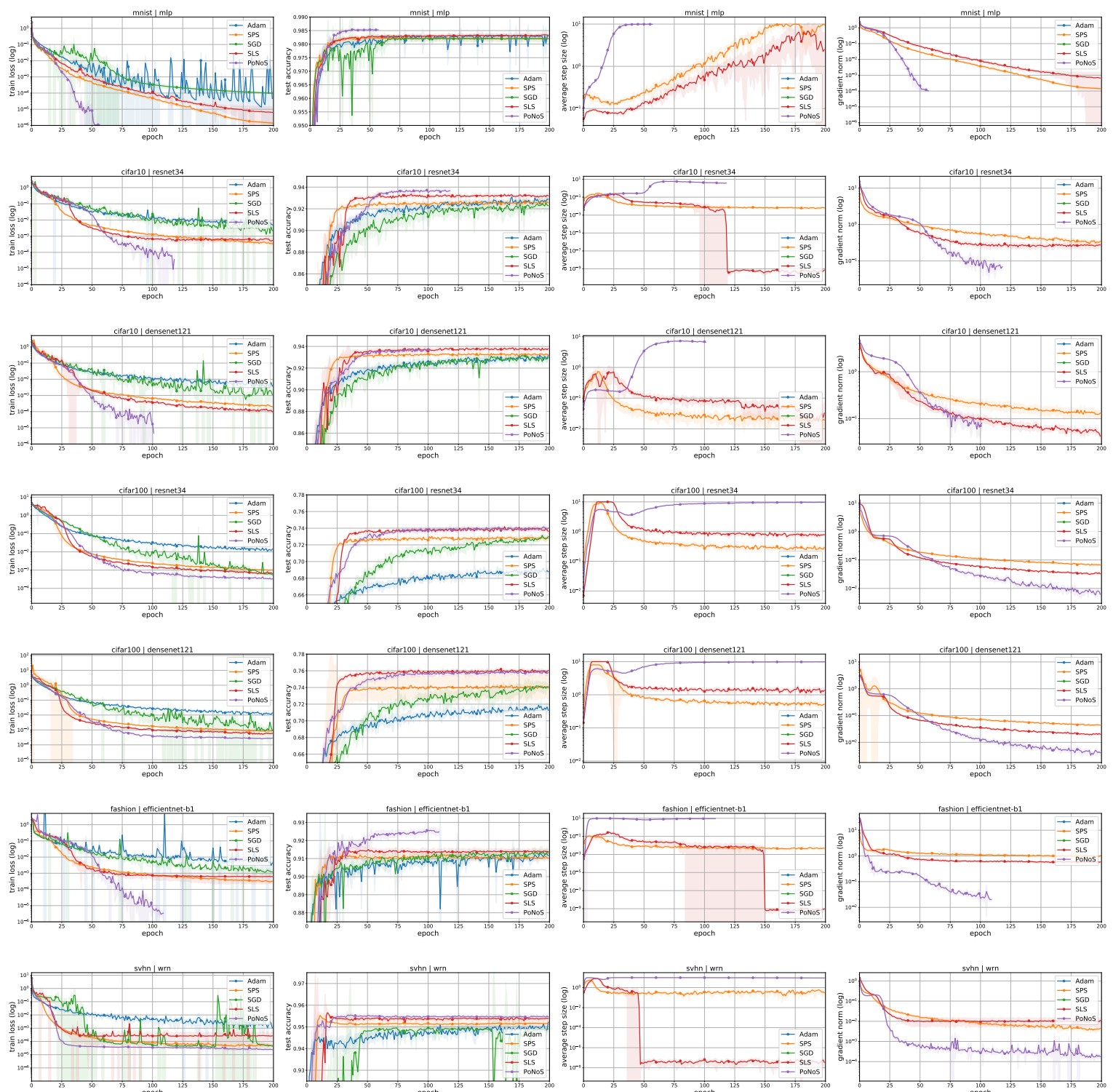

Figure I: Comparison between the proposed method (PoNoS) and the-state-of-the-art. Each row focus on a dataset/model combination. First column: train loss. Second column: test accuracy. Third column: average step size of the epoch. Fourth column: average gradient norm of the epoch.

## D.2 A New Resetting Technique

See Figure II for the complete results relative to Figure 2 of the main paper. From Figure II, we can additionally observe that the step size yielded by PoNoS is generally small in the initial phase, it grows in the intermediate phase and reaches $\eta^{\max}$ towards the end.

This behavior can be also noticed in the amount of backtracks of PoNoS_reset0 in the rightmost column of Figure II. The number of backtracks is conspicuous at the beginning, rare in the intermediate phase and (almost) zero in the local phase. On the other hand, the resetting technique introduced in PoNoS maintains the amount of backtracks to be limited by 1 per iteration (on average), while still yielding the same step size. When PoNoS_reset0 starts to reduce the amount of backtracks, also PoNoS does the same.

The step size behavior described above and the corresponding amount of backtracks are in accordance with the intuition and the theory. Intuitively, a small step size allows the algorithm to proceed more cautiously in the global phase. A larger step size allows the algorithm to converge faster once it gets closer to the local phase. In fact, for achieving local Q-superlinear convergence, the theory predicts that the line search should always accept a new step size in the local phase [Nocedal and Wright, 2006].

### D.3    Time Comparison

See Figure III for the complete results relative to Figure 3 of the main paper. All the time measures are obtained by using the `time.time()` command from the `time python` package. The cumulative time in the $x$-axis of the first and second columns of Figure III has been computed as an average of 5 different runs. More precisely, the time of each epoch have been computed separately for the 5 runs and then averaged, so that the same average time has been assigned to each of the 5 runs. At this point, these averaged per-epoch times have been cumulated along the epochs. The shaded error bars in the plots correspond to the standard deviation from the mean of the loss/accuracy and not to the runtime. The runtime standard deviation is instead reported in the third column of Figure III.

### D.4    Experiments on Convex Losses

See Figure IV for the complete results relative to the convex experiments of Figure 4 of the main paper. Only the training sets available in the LIBSVM library were used for these datasets. The $80\%$ split of the data was used as a training set and $20\%$ split as the test set. For the RBF kernel bandwidth, we employed the parameters suggested in Vaswani et al. [2019], that is $\{0.5, 0.25, 0.05, 20\}$ respectively for mushrooms, rcv1, ijcnn and w8a. We did not use any bias parameter in these experiments. Furthermore, given the convexity of these problems, $\eta^{\text{max}}$ is not needed and it has been set to $\infty$.

In Figure IV, the three measures are reported by iterations and not by epochs. To avoid large fluctuations in the plots, train loss and step size have been smoothed using an exponential moving average ($\beta = 0.9$). On the other hand, the test accuracy is only computed at the beginning of each epoch and the same value is reported till the next epoch. In Figure IV, we can make additional observations which are not visible in Figure 4 of the main paper:

- PoNoS achieves the lowest loss value on `rcv1` and `ijcnn`. Regarding the test accuracy, PoNoS achieves always the highest score, apart from `rcv1` on which it loses  0.5 points w.r.t. SLS and SPS.

- SLS and SPS behave very similarly on all the datasets. They achieve the best accuracy on all the problems, but they are both very slow on `mushrooms`, `rcv1` and `ijcnn` in terms of loss. This behavior is due to (4), as it is clear from the third column of Figure IV. The step size yielded by (4) is often too small and it slowly grows exponentially through the whole optimization process. This choice is suboptimal if compared with the step size yielded by PoNoS.

### D.5    Experiments on Transformers

See Figure V for the complete results relative to experiments on transformers of Figure 4 of the main paper. In these experiments, we followed the setup by Kunstner et al. [2023]. The word-level language modeling has been addressed with sequences of 35 or 128 tokens respectively for PTB and Wikitext2. In case of PTB, we employed a simple transformer model whose architecture consists of an 200-dimensional embedding layer, 2 transformer layers (2-head self attention, layer normalization, linear($200 \times 200$)-ReLU-linear($200 \times 200$), layer normalization) followed by a linear layer. The data processing and the implementation of the Transformer-XL follow Dai et al. [2019]. In the case of Wikitext2, the hyperparameters are set as in the ENWIK8 base experiment of Dai et al. [2019], except with the modifications of Zhang et al. [2020], using 6 layers and a target length of 128.

Since Adam is commonly known to achieve better performances than SGD on these networks [Kunstner et al., 2023], we develop a preconditioned version of PoNoS, SLS and SPS respectively called PoNoS_prec, SLS_prec, and SPS_prec. These algorithms differ from the originals in three aspects. First, they all employ a direction which is Adam without momentum ($\beta_1 = 0$). More precisely, the mini-batch gradient $\nabla f_{i_k}(w_k)$ in Step 13 of Algorithm 1 is replaced with $d_k$ as computed below

$$g_k = \nabla f_{i_k}(w_k)$$
$$v_k = \beta_2 \cdot v_{k-1} + (1 - \beta_2) \cdot g_k^2$$
$$\hat{v}_k = v_k / (1 - \beta_2^k)$$
$$d_k = -g_k / (\sqrt{\hat{v}_k} + \epsilon),$$

where all the operations on vectors are to be considered component-wise, $\beta_2 \in (0, 1)$ and $\epsilon > 0$ is a small constant. We use the default values from Adam for $\beta_2$ and $\epsilon$, that is $\beta_2 = 0.9$ and $\epsilon = 10^{-8}$. The second difference concerns the line search, as PoNoS_prec and

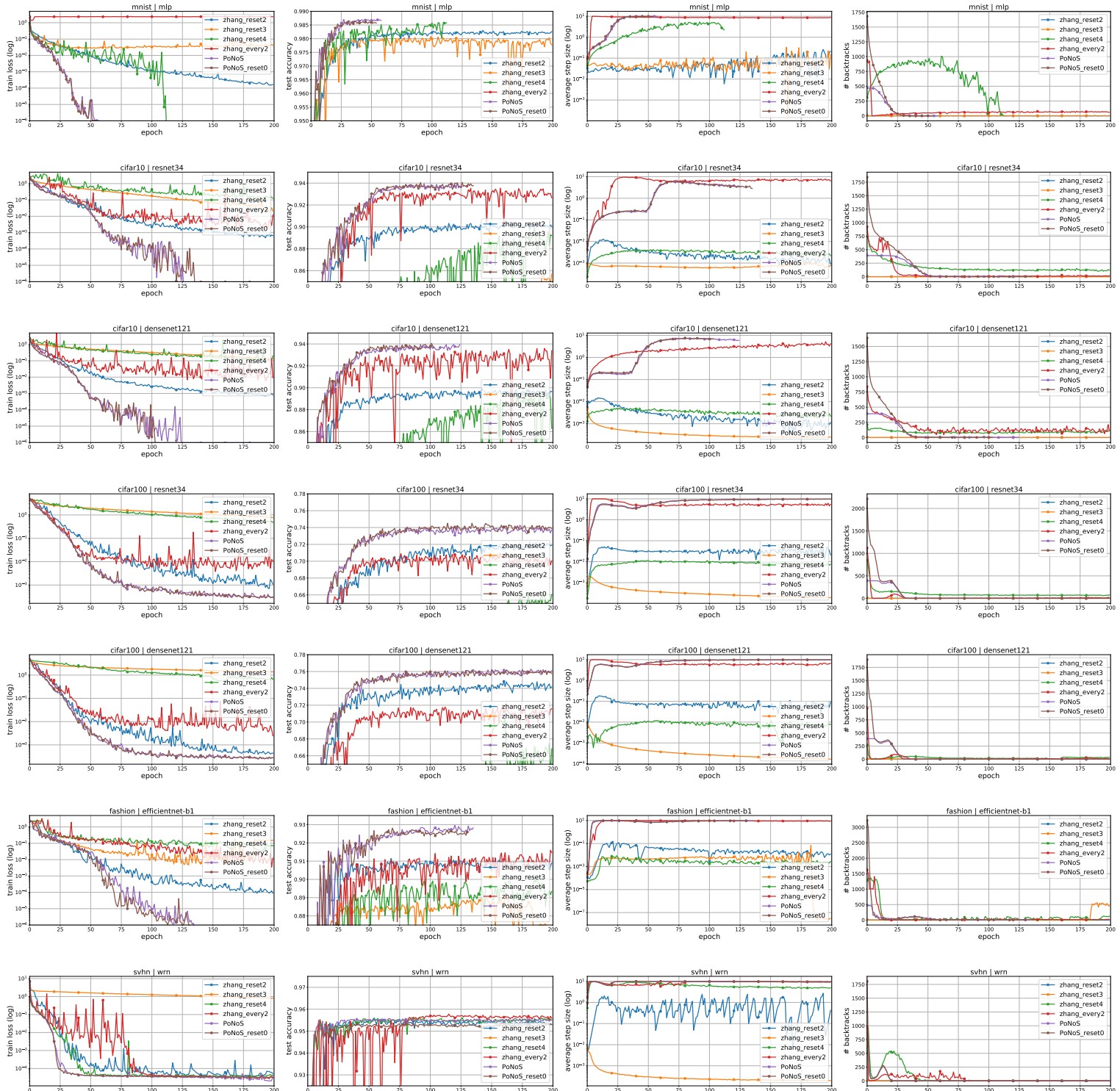

Figure II: Comparison between different initial step sizes and resetting techniques. Each row focus on a dataset/model combination. First column: train loss. Second column: test accuracy. Third column: average step size of the epoch. Fourth column: cumulative number of backtracks in the epoch.

SLS_prec exploit a condition that reflects the use of the above preconditioned direction, i.e.,

$$f_{i_k}(w_k + \eta_k d_k) \leq R_k + c \cdot \eta_k \langle d_k, \nabla f_{i_k}(w_k) \rangle = R_k - c \cdot \eta_k \sum_{j=0}^{n-1} \frac{(\nabla f_{i_k}(w_k))_j^2}{(\sqrt{\hat{v}_k} + \epsilon)_j},$$

where $(\cdot)_j$ is referring to the $j$-th component of the vector in brackets, and $R_k$ is either $C_k$ in case of PoNoS_prec or $f_{i_k}(w_k)$ in case of SLS_prec. The third difference concerns the Polyak step size, which is also computed taking into account the above directional derivative $\langle d_k, \nabla f_{i_k}(w_k) \rangle$. In particular, PoNoS_prec and SPS_prec replace (5) with the following

$$\tilde{\eta}_{k,0} := \frac{f_{i_k}(w_k) - f_{i_k}^*}{-c_p \langle d_k, \nabla f_{i_k}(w_k) \rangle}.$$

As in Kunstner et al. [2023] we focus on the training procedure, however, in the second column of Figure V we report the perplexity of the language model on the test. With respect to Figure 4 of the main paper, in Figure V we also show the gradient norm and the step size yielded by the different methods. From these plots, it is possible to notice that the range in which the step size varies is reduced if compared with that of Figure I. This behavior depends on the dynamics of the loss and of the norm of the gradient, which are also reduced if compared with those of Figure I. The reduced ranges of loss and gradient norm might be an issue for Polyak-based algorithms since they rely on these measures. Furthermore, the good results of SLS_prec on `wiki2|encoder` suggest that other initial step sizes might also be suited for training transformers. We leave such exploration to future works.

# E   Additional Plots

## E.1   Study on the Choice of $c$: Theory (0.5) vs Practice (0.1)

In these experiments, we consider the constant $c$ in (3) and (2). As described in Section 4 of the main paper, this value is required to be larger than $\frac{1}{2}$ in both Theorems 1 and 2 (and also in the corresponding monotone versions from Vaswani et al. [2019]). This value is often considered too large in practice and the default choice is $0.1$ (also for SLS [Vaswani et al., 2019]) or smaller [Nocedal and Wright, 2006]. The constant $c$ controls the weight of the sufficient decrease in the line search conditions and a smaller value of $c$ corresponds to a looser line search. In this subsection, we numerically try both $c = 0.5$ and $c = 0.1$. In Figure VI and VII we compare

- monotone|0.1: the monotone stochastic Armijo line search (2) with $c = 0.1$.
- monotone|0.5: the monotone stochastic Armijo line search (2) with $c = 0.5$.
- zhang|0.1: the nonmonotone Zhang and Hager line search (3) with $c = 0.1$.
- zhang|0.5: the nonmonotone Zhang and Hager line search (3) with $c = 0.5$. This setting corresponds to PoNoS.

For all the above algorithms the initial step size is (6). From Figure VI we can observe that:

- zhang|0.5 (PoNoS) achieves the best performances both in terms of train loss and test accuracy. It is the only algorithm able to reduce the train loss below the threshold of $10^{-6}$ on the problems `cifar10|resnet34` and `cifar100|resnet34`. Apart from the case of `svhn|wrn`(where it loses less than $0.5$ points w.r.t. zhang|0.1), it always achieves the best test accuracy.

- monotone|0.1 and zhang|0.1 are generally competitive in terms of training loss, but they achieve very poor generalization skills on `cifar100|resnet34` and `cifar100|densenet121`. In particular, their step size on these two problems is growing very rapidly. Already in the first epoch, the average step size is greater than 5, thanks to the fact that both algorithms are reducing the gradient norm rapidly below 1.

- The step size yielded by zhang|0.5 starts substantially lower than those of monotone|0.1 and zhang|0.1. Afterwards, the step size increases, then stabilizes, sometimes slightly decreases and finally increases again. This behavior is accomplished thanks to the combination between a larger $c$ and a nonmonotone line search. Also the gradient norm is affected by this choice, since it does not decrease as suddenly as for monotone|0.1 and zhang|0.1.

- monotone|0.5 never achieves the best test accuracy nor the best training loss. This line search is too strict and yields step sizes that are very small in comparison to those yielded by zhang|0.5.

In conclusion, PoNoS (zhang|0.5) employs a large constant $c$ (0.5), but it combines that with a nonmonotone line search to achieve the best middle way between strictness and tolerance.

In Figure VII we propose the same comparison as in Figure VI, but in the case of the convex experiments of Figure IV. From Figure VII we can observe that zhang|0.1 and monotone|0.1 do not obtain good performances. On both `rcv1` and `ijcnn`, these algorithms do not converge, suggesting that a large constant $c$ is sometimes required to achieve convergence.

## E.2   Study on the Line Search Choice: Various Nonmonotone Adaptations

In this subsection, we propose a comparison between various line search conditions. For the first time in this paper, we adapt different nonmonotone techniques to SGD. A straightforward adaptation of the nonmonotone line search from Grippo et al. [1986] is

$$f_{i_k}(w_k - \eta_k \nabla f_{i_k}(w_k)) \leq \max_{0 \leq j \leq W-1} f_{i_k}(w_{k-j}) - c\eta_k \| \nabla f_{i_k}(w_k) \|^2. \tag{25}$$

The nonmonotone term in (25) can be computed in two ways. Either by keeping in memory $W$ (usually 10 or 20) previous vector weights $\{w_{k-W}, \dots, w_k\}$ and computing the current mini-batch function $f_{i_k}$ on all of them. Or by computing all the following $W$

mini-batch functions $\{f_{i_k}(\cdot), \ldots, f_{i_{k+W}}(\cdot)\}$ on $w_k$. Both options are very expensive in the case of large neural networks and they should be avoided. The condition (25) is the same proposed in Hafshejani et al. [2023].

To reduce the cost of computing (25), we here propose two other stochastic adaptations of the nonmonotone line search from Grippo et al. [1986]. We name *cross-batch Grippo* the following line search condition

$$f_{i_k}(w_k - \eta_k \nabla f_{i_k}(w_k)) \leq f_{i_{k-\tilde{r}}}(w_{\tilde{r}}) - c\eta_k \|\nabla f_{i_k}(w_k)\|^2, \tag{26}$$

where $\tilde{r}$ is a short notation for $\tilde{r}(k, i_k)$, which is the (say, largest) iteration index such that $f_{i_{k-\tilde{r}(k,i_k)}}(w_{\tilde{r}(k,i_k)}) = \max_{0 \leq j \leq W-1} f_{i_{k-j}}(w_{k-j})$. The computation of (26) does not introduce overhead since it directly uses the function values computed in the previous iterations. However, the fact that (26) computes the maximum over different mini-batch functions $\{f_{i_{k-W}}(\cdot), \ldots, f_{i_k}(\cdot)\}$ complicates the convergence analysis. We conjecture that under the interpolation assumption alone, it is not possible to achieve convergence if (26) replaces (3), not even in the strongly convex case.

We call *single-batch Grippo* the following line search condition

$$f_{i_k}(w_k - \eta_k \nabla f_{i_k}(w_k)) \leq f_{i_k}(w_{r(k,i_k)}) - c\eta_k \|\nabla f_{i_k}(w_k)\|^2, \tag{27}$$

where $r(k, i_k)$ is the (say, largest) iteration index such that $f_{i_k}(w_{r(k,i_k)}) = \max_{0 \leq j \leq W-1} f_{i_k}(w_{k-j\frac{M}{b}})$. This line search is similar to (25) since it focuses on $f_{i_k}$. However, it also reduces the cost of computing the nonmonotone term. In fact, given a certain set of indexes $i_k$, it exploits the function values computed in the previous epochs, without re-computing $f_{i_k}$. This requires saving a matrix of $M \times W$ floating-point numbers. On the other hand, to use this computational trick, the nonmonotone term can only be computed starting from the second epoch. Moreover, (27) is not computationally as cheap as (26) or (3). In particular, at every mini-batch iteration (26) requires the extraction of the $b \times W$ values corresponding to the indexes in $i_k$ from the above-mentioned matrix and to compute the maximum over these values.

In Figure VIII we compare

- monotone: the monotone stochastic Armijo line search (2).
- cross_batch_grippo: the cross-batch nonmonotone Grippo's line search (26) with $W = \frac{M}{b}$.
- single_batch_grippo: the single-batch nonmonotone Grippo's line search (27) with $W = 10$.
- zhang: the nonmonotone Zhang and Hager [2004] line search adapted to the stochastic case (3). This setting corresponds to PoNoS.

For all the above algorithms the initial step size is (6). From Figure VIII we can observe that:

- zhang (PoNoS) achieves the best performances both in terms of train loss and test accuracy. Also in this case, it is the only algorithm able to reduce the train loss below the threshold of $10^{-6}$ on the problems `cifar10|resnet34`, `cifar10|densenet121` and `fashion|effb1`. Moreover, it always achieves the highest test accuracy. In `fashion|effb1` it is not as fast as cross_batch_grippo and single_batch_grippo.
- single_batch_grippo achieves similar performances as zhang. However, it does not always reach the same test accuracy (e.g., it loses ~1 point on `cifar100|resnet34`). On `fashion|effb1` it is the fastest algorithm in terms of train loss. On the other hand, it almost never reduces the amount of backtracks below 100 per epoch.
- The step sizes yielded by both Grippo's adaptations are often growing faster than those of zhang, especially in the initial phase of the optimization procedure. On the other extreme, the monotone line search is too strict and yields step sizes that are very small in comparison to those of zhang. In fact, monotone is achieving very poor results both in terms of train loss and test accuracy.
- cross_batch_grippo is never as fast as single_batch_grippo in terms of train loss and it loses ~5 points of accuracy on `cifar100|resnet34` and `cifar100|densenet121` w.r.t. zhang. In accordance with the conjecture above, cross_batch_grippo does not converge on `mnist|mlp`.

In conclusion, zhang (PoNoS) is the best-performing line search technique among those compared in Figure VIII. In terms of step size, zhang provides the right middle way between the very permissive Grippo's conditions and the very strict monotone one. Moreover, even if zhang and single_batch_grippo behave similarly, the second is computationally more expensive and it is never able to reduce the backtracks to (almost) always zero.

### E.3   Zoom in on the Amount of Backtracks

In this subsection, we zoom in on the amount of backtracks employed by PoNoS and PoNoS_reset0. Instead of showing the cumulative number of backtracks in each epoch (as in the fourth column of Figure II), Figure IX reports the amount of backtracks in each iteration. To help visualizing the whole optimization procedure, we average the number of backtracks over 10 consecutive iterations. Respectively on the leftmost and rightmost column of Figure IX we report the first 20000 iterations and 100 iterations (out of 78200-114600). From these columns we can observe:

- PoNoS_reset0 employs a stable amount of backtracks across iterations.

- PoNoS reduces the number of backtracks to 1 on average already after the first iteration. Afterwards, this value is first stable for the first 1000-10000 iterations (5-25 epochs) and it then reaches a median of 0 after a transition phase.

In the second column of Figure IX, we report the average difference between two consecutive amount of backtracks in the first 20000 iterations. In particular, this value is higher in the initial phase, while (almost) always 0 in a later stage. Focusing on PoNoS_reset0 this difference is almost always below 1. Regarding the newly proposed resetting technique (8), we can conclude that the value $l_{k-1}$ is a good estimate for $l_k$.

### E.4 Study on the Choice of $\eta^{\mathrm{max}}$

In this subsection, we report an ablation study on $\eta^{\mathrm{max}}$ for PoNoS, SLS and SPS. In Figure X, we compare

- SPS|10: SPS with $\eta^{\mathrm{max}} = 10$. This setting corresponds to SPS.
- SPS|100:SPS with $\eta^{\mathrm{max}} = 100$.
- SLS|10: SLS with $\eta^{\mathrm{max}} = 10$. This setting corresponds to SLS.
- SLS|100: SLS with $\eta^{\mathrm{max}} = 100$.
- PoNoS|10: PoNoS_reset0 with $\eta^{\mathrm{max}} = 10$. This setting corresponds to PoNoS_reset0.
- PoNoS|100: PoNoS_reset0 with $\eta^{\mathrm{max}} = 100$.

We report PoNoS_reset0 (without (8)) instead of PoNoS because we want to observe the difference in the amount of backtracks. To directly observe the effect of changing $\eta^{\mathrm{max}}$, we report the average initial step size within each epoch in the third column of Figure X. From Figure X we can observe:

- The value of $\eta^{\mathrm{max}} = 100$ is never reached by the step sizes of the three algorithms. The step size of PoNoS|100, SLS|100 and SPS|100 can be considered unbounded.

- The three algorithms are robust to the choice of $\eta^{\mathrm{max}}$. In fact, the train loss and test accuracy of PoNoS|100, SLS|100 and SPS|100 are similar to those of PoNoS|10, SLS|10 and SPS|10.

- In terms of train loss, there are some small differences on `cifar100|resnet34`, `cifar100|densenet121` and `fashion|effb1`. On these problems, PoNoS|100, SLS|100 and SPS|100 are sometimes slower during the intermediate phase of the optimization. However, the three unbounded algorithms are able to catch up towards the end.

- In terms of test accuracy, both SLS|100 and SPS|100 lose more than 2 points on `cifar100|densenet121` w.r.t. SLS|10 and SPS|10, while PoNoS|100 and SLS|100 lose more than 2 points on `cifar100|densenet121`.

- PoNoS|100 performs overall better than SLS|100 and SPS|100.

- The amount backtracks of PoNoS|100 is sometimes remarkably larger than for PoNoS|10. However, the use of (8) would reduce this number to (almost) always zero.

### E.5 Study on the Choice of $c_p$: Doubling the Legacy Value

In this subsection, we report an ablation study on $c_p$ in (5) for PoNoS and SPS. PoNoS utilize the Polyak step size as an initial guess for the line search method and not as a direct learning rate for SGD as in SPS [Loizou et al., 2021]. We suggest doubling the value employed in Loizou et al. [2021] by employing $c_p = 0.1$ in (5) instead of the default value of $c_p = 0.2$. Given the fact that $\delta = 0.5$, the line search will half the step size when needed, achieving only in this case the same effect of reducing $c_p$ back to $c_p = 0.2$. In Figure XI we compare

- SPS|0.1: SPS with $c_p = 0.1$.
- SPS|0.2: SPS with $c_p = 0.2$. This setting corresponds to SPS.
- PoNoS|0.1: SLS with $c_p = 0.1$. This setting corresponds to PoNoS.
- PoNoS|0.2: SLS with $c_p = 0.2$.

From Figure XI we can observe that the differences between $c_p = 0.1$ and $c_p = 0.2$ are not remarkable. However, PoNoS|0.1 always achieves slightly better performance than PoNoS|0.2 both in terms of train loss and test accuracy. On the other hand, the test accuracy of SPS|0.1 is not always as high as that achieved by SPS|0.2. To conclude, we can observe from the third column of Figure XI that the step size of PoNoS|0.1 is not always larger than that of PoNoS|0.2. In fact, this does not happen in the initial phase of the optimization, but in the intermediate phase and only on some problems. Given the improved performance of PoNoS|0.1 over PoNoS|0.2, one could argue that the line search finds the regions in which a larger step is beneficial.

## E.6 Profiling PoNoS

In this subsection, we profile a single mini-batch iteration of PoNoS. More precisely, in Figure XII we report the time (s) employed by the different operations required to perform an iteration of PoNoS. Averaging along the epoch (over $\frac{M}{b}$ values), in Figure XII we compare

- 1st_fwd: average time employed to perform the first forward pass;
- 2nd_fwd: average time employed to perform the second forward pass;
- extra_fwd: average time employed to perform all the extra forward passes beyond the second. Notice that the total amount of extra forward passes could be more or less than $\frac{M}{b}$. Despite the actual amount of extra forward passes, we still divide the sum by $\frac{M}{b}$;
- backward: average time employed to perform the backward pass;
- batch_load: average time employed to load a mini-batch of instances into the GPU memory;

From Figure XII, we can observe the following.

- 1st_fwd and 2nd_fwd employ roughly the same time. 2nd_fwd is always slightly faster than 1st_fwd, however, the difference is not substantial.
- A backward pass employs roughly twice the time of a forward pass.
- The time employed to load the mini-batch in the GPU memory is negligible on some networks (`cifar10|densenet121`, `cifar100|densenet121`and `fashion|effb1`). On the other networks, this time is between one-half and 4 times that of a forward pass.
- As previously shown in Figure IX, PoNoS transitions from a phase in which it employs 1 backtrack to 0 (on median). In Figure XII, this same behavior can be also observed in the time employed by all the forward passes beyond the second.

To conclude, the time employed by a single forward pass may reach up to one-third of the computation required for SGD (i.e., batch_load + 1st_fwd + backward). Following these calculations and referring to the two phases of Figure III and Figure XII, one iteration of PoNoS only costs $\frac{5}{3}$ that of SGD in the first phase and $\frac{4}{3}$ in the second.

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

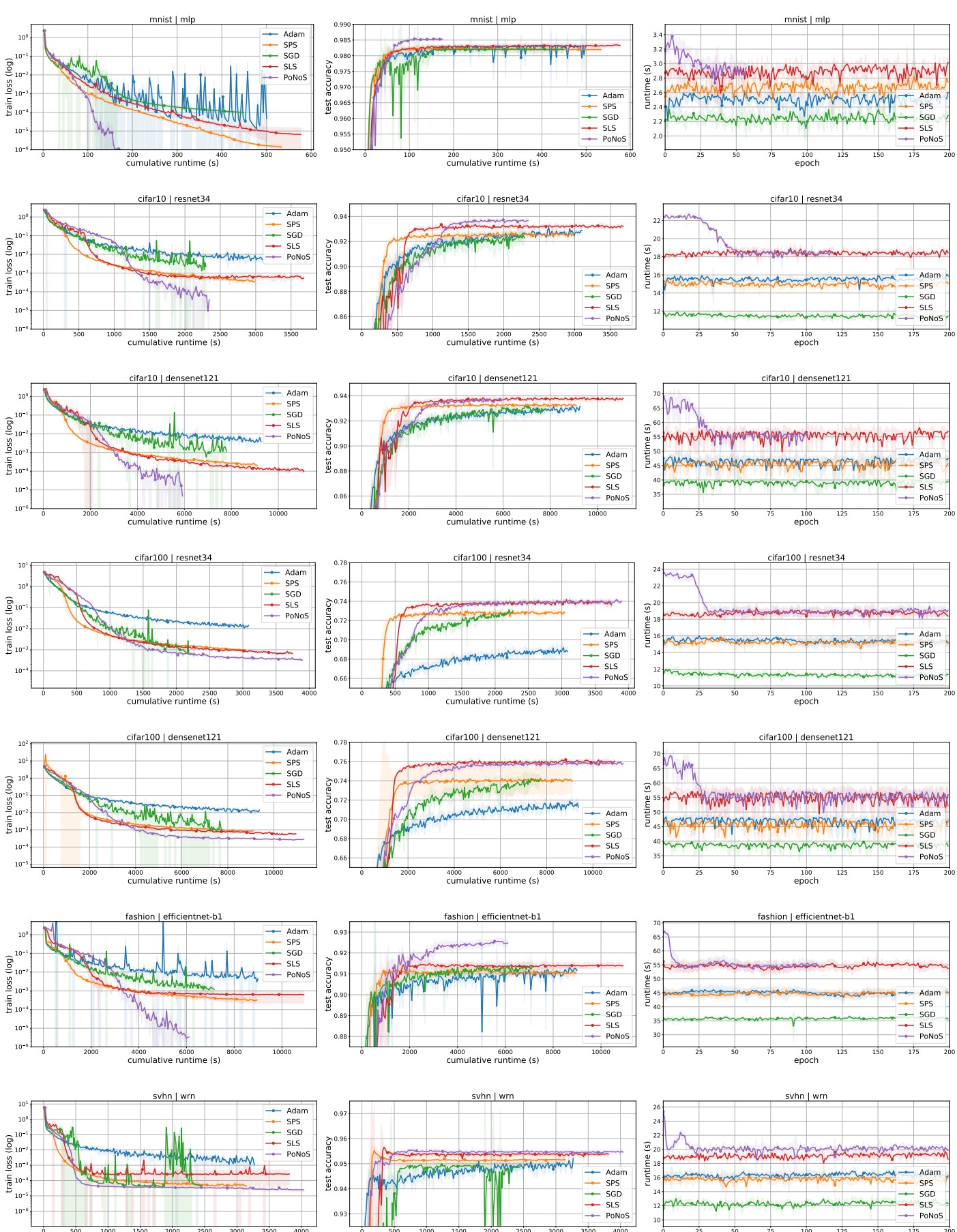

Figure III: Time comparison (s) between the proposed method (PoNoS) and the-state-of-the-art. Each row focus on a dataset/model combination. First column: train loss vs cumulative runtime. Second column: test accuracy vs cumulative runtime. Third column: per-epoch runtime.

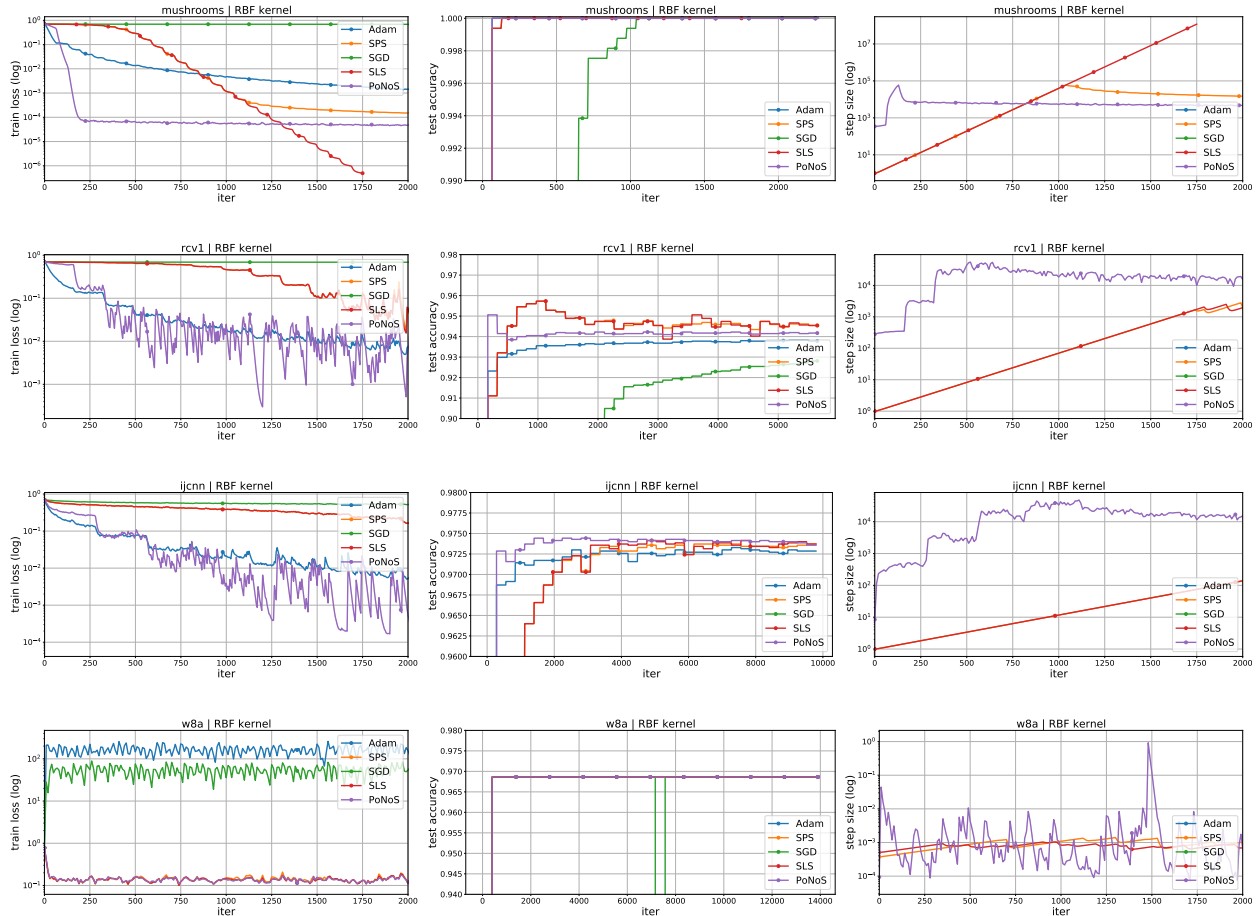

Figure IV: Comparison between the proposed method (PoNoS) and the state-of-the-art on convex kernel models for binary classification. Each row focus on a different dataset. First column: exponentially averaged train loss. Second column: test accuracy. Third column: exponentially averaged step size.

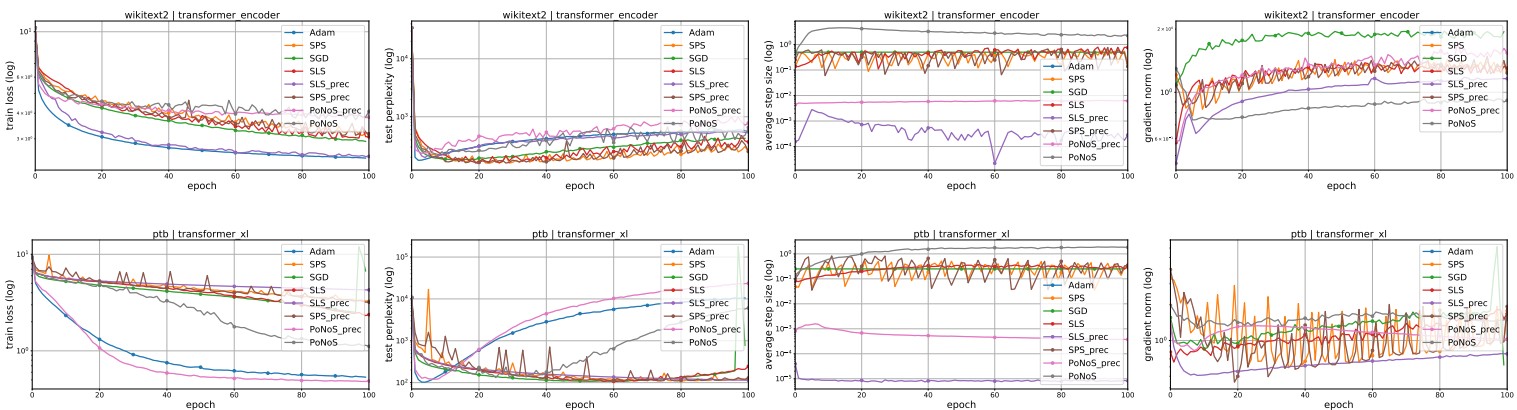

Figure V: Comparison between the proposed method (PoNoS) and the state-of-the-art on the training of transformers for language modeling tasks. Each row focus on a dataset/model combination. First column: train loss. Second column: train perplexity. Third column: average step size. Fourth column: average gradient norm.

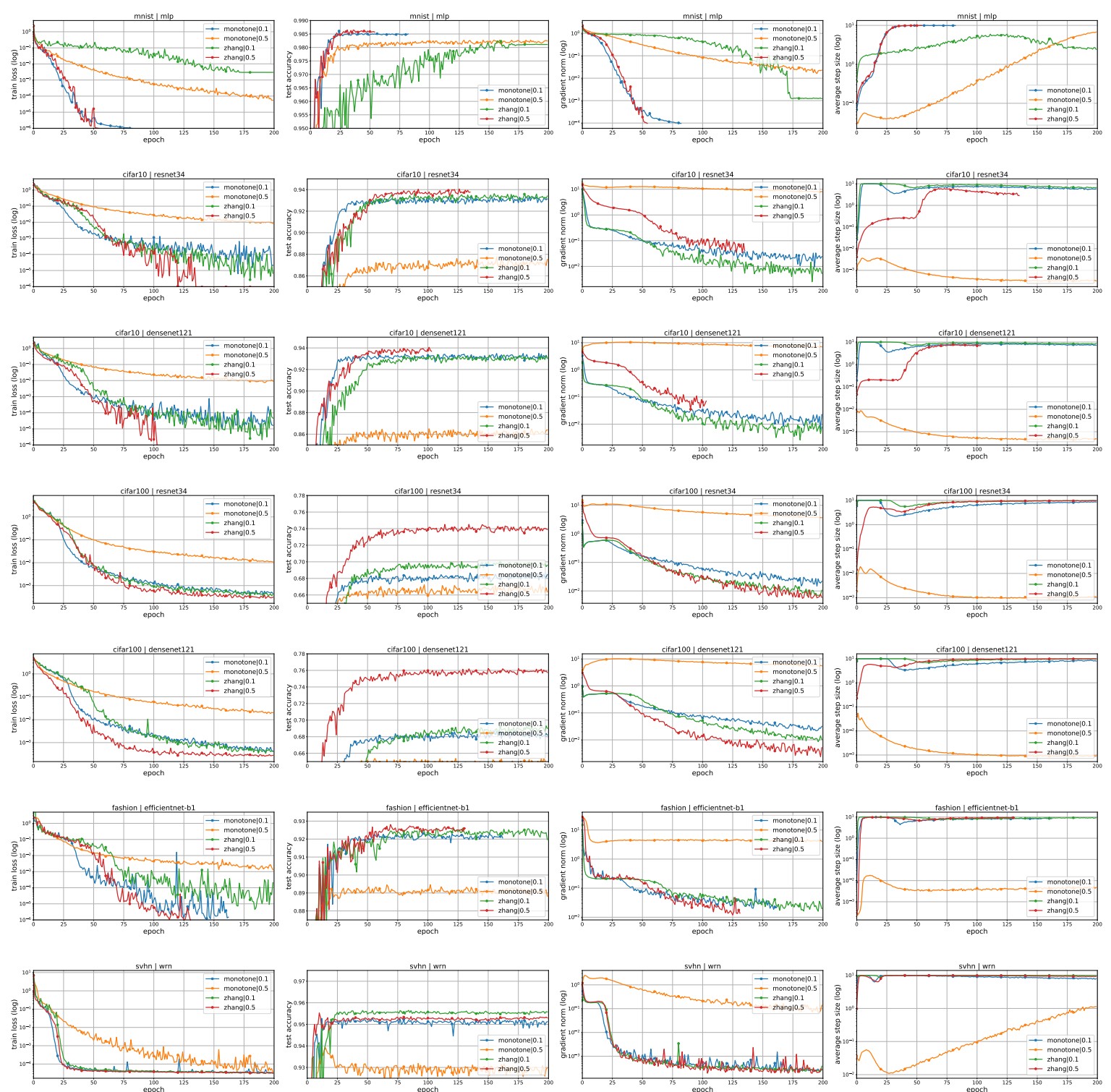

Figure VI: Comparison between the use of the constant $c = 0.1$ and $c = 0.5$ on both monotone and nonmonotone algorithms. Each row focus on a dataset/model combination. First column: train loss. Second column: test accuracy. Third column: average step size of the epoch. Fourth column: average gradient norm of the epoch.

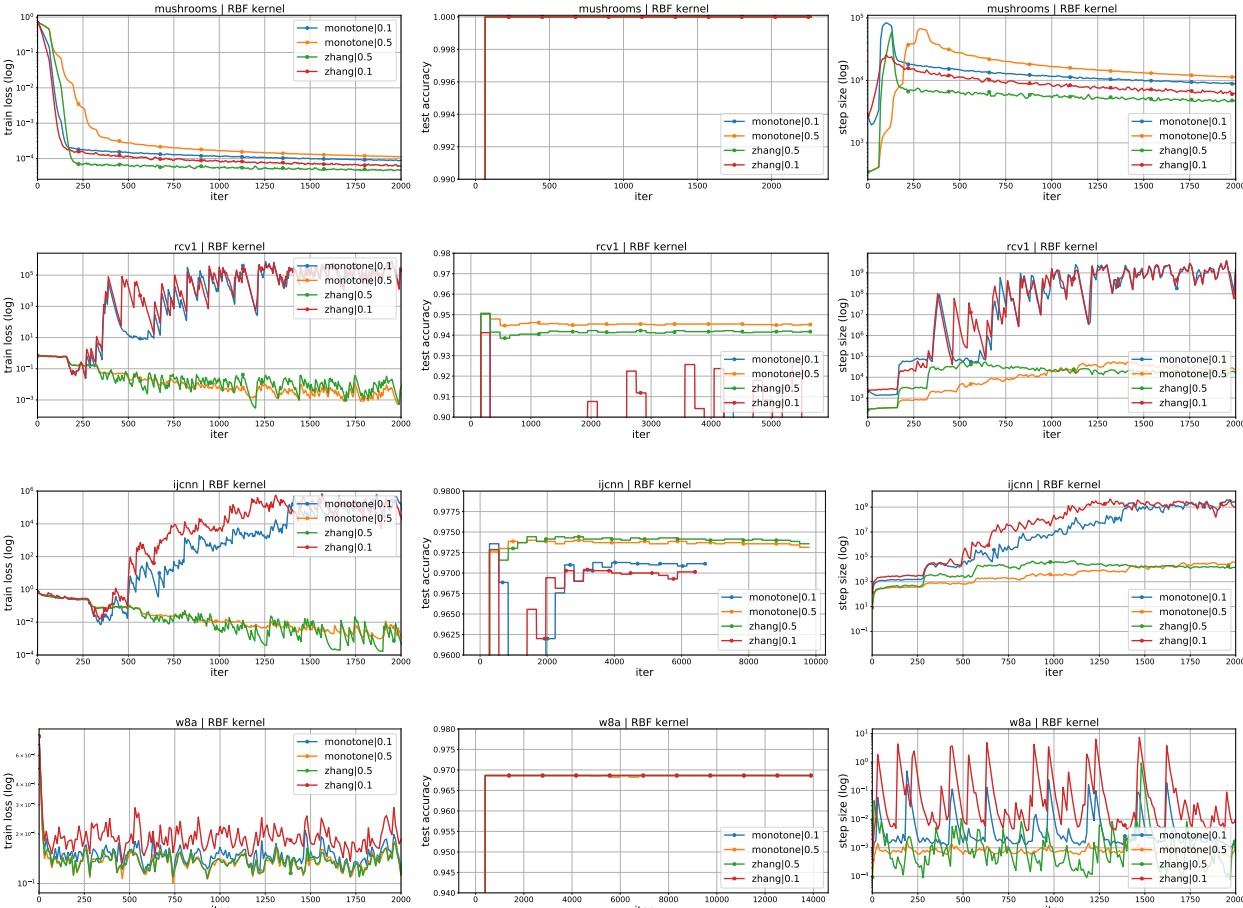

Figure VII: Comparison between the use of the constant $c = 0.1$ and $c = 0.5$ on both monotone and nonmonotone algorithms on convex binary classification problems. Each row focus on a different dataset. First column: exponentially averaged train loss. Second column: test accuracy. Third column: exponentially averaged step size.

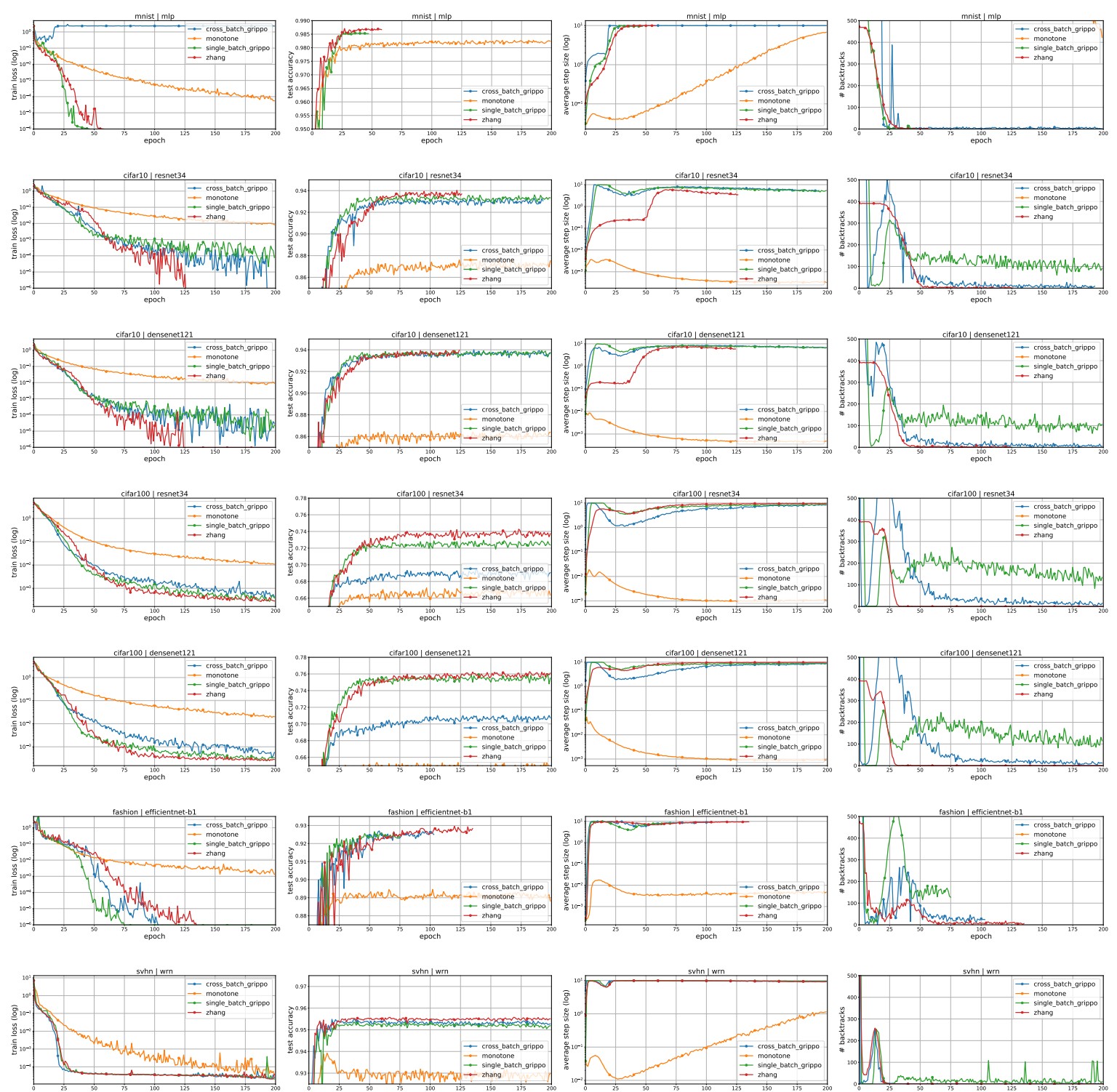

Figure VIII: Comparison between different monotone and nonmonotone line search conditions. Each row focus on a dataset/model combination. First column: train loss. Second column: test accuracy. Third column: average step size of the epoch. Fourth column: cumulative number of backtracks in the epoch.

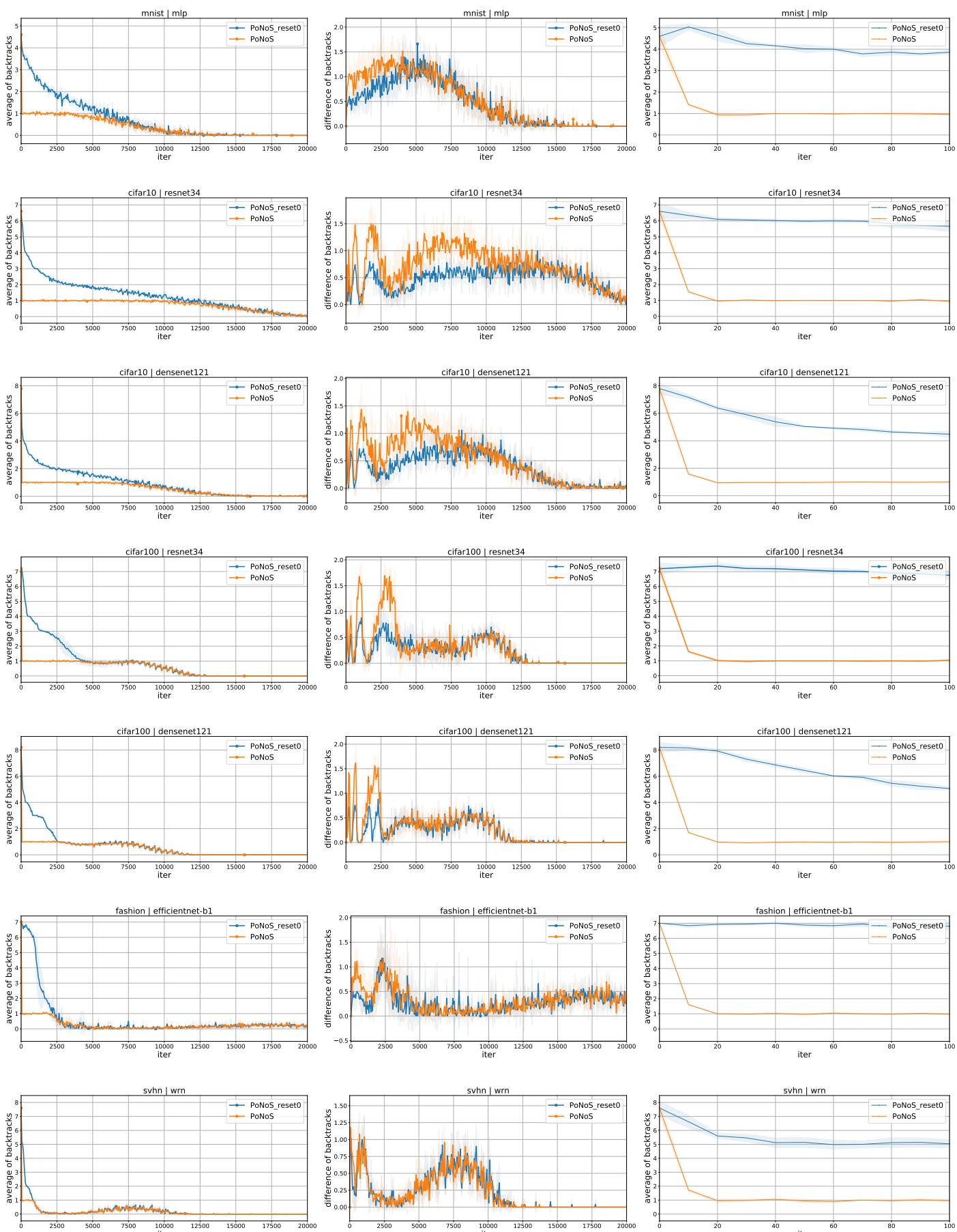

Figure IX: Comparison on the number of backtracks between the proposed method with (PoNoS) or without the new resetting technique (PoNoS_reset0). Each row focus on a dataset/model combination. First column: average number of backtracks in the first 200000 iterations. Second column: average difference between two consecutive amount of backtracks in the first 200000 iterations. Third column: average number of backtracks in the first 1000 iterations.

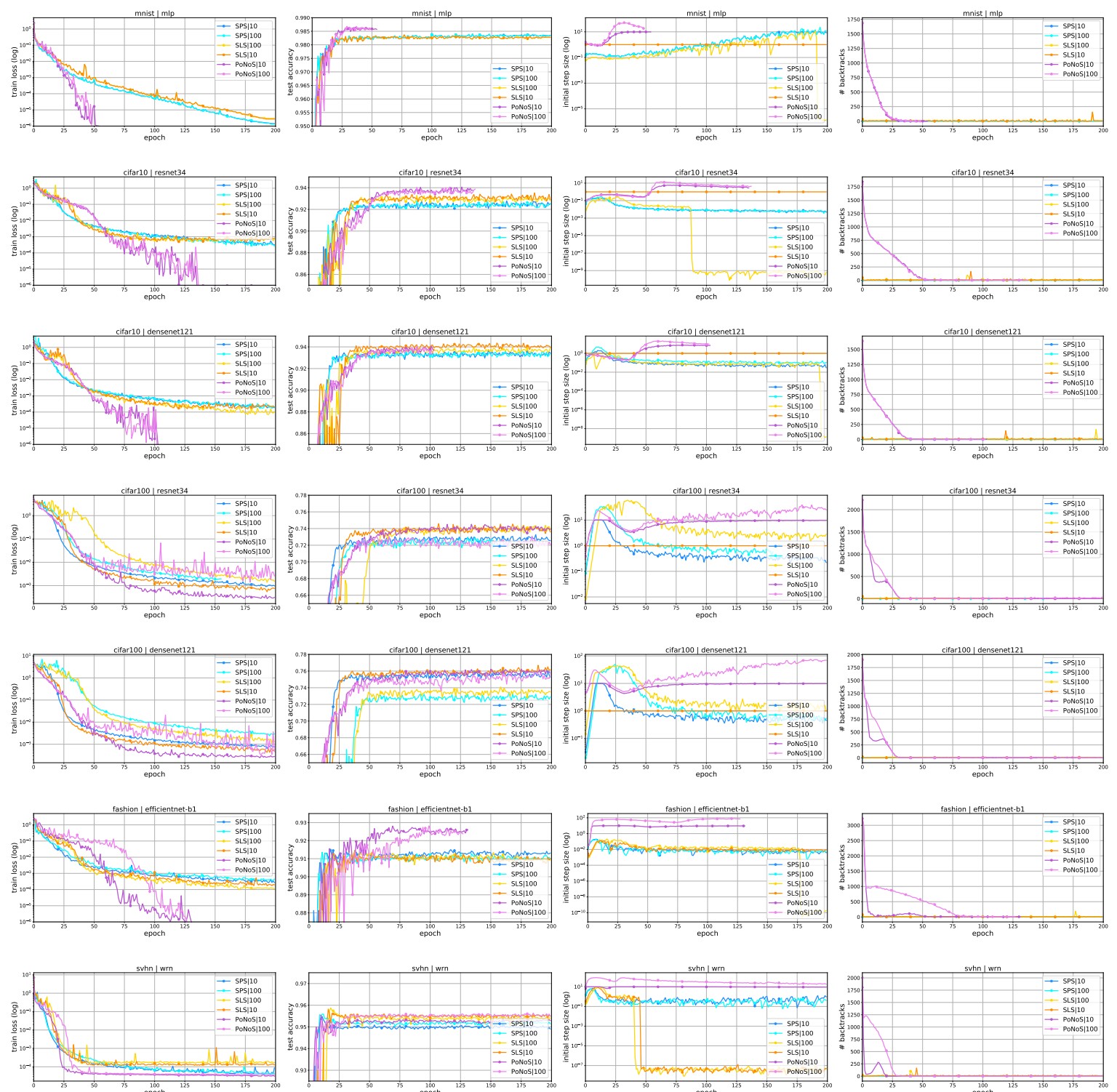

Figure X: Comparison between the use of the constant $\eta^{\mathrm{max}} = 10$ and $\eta^{\mathrm{max}} = 100$ on SPS, SLS and PoNoS. Each row focus on a dataset/model combination. First column: train loss. Second column: test accuracy. Third column: average initial step size of the epoch. Fourth column: cumulative number of backtracks in the epoch.

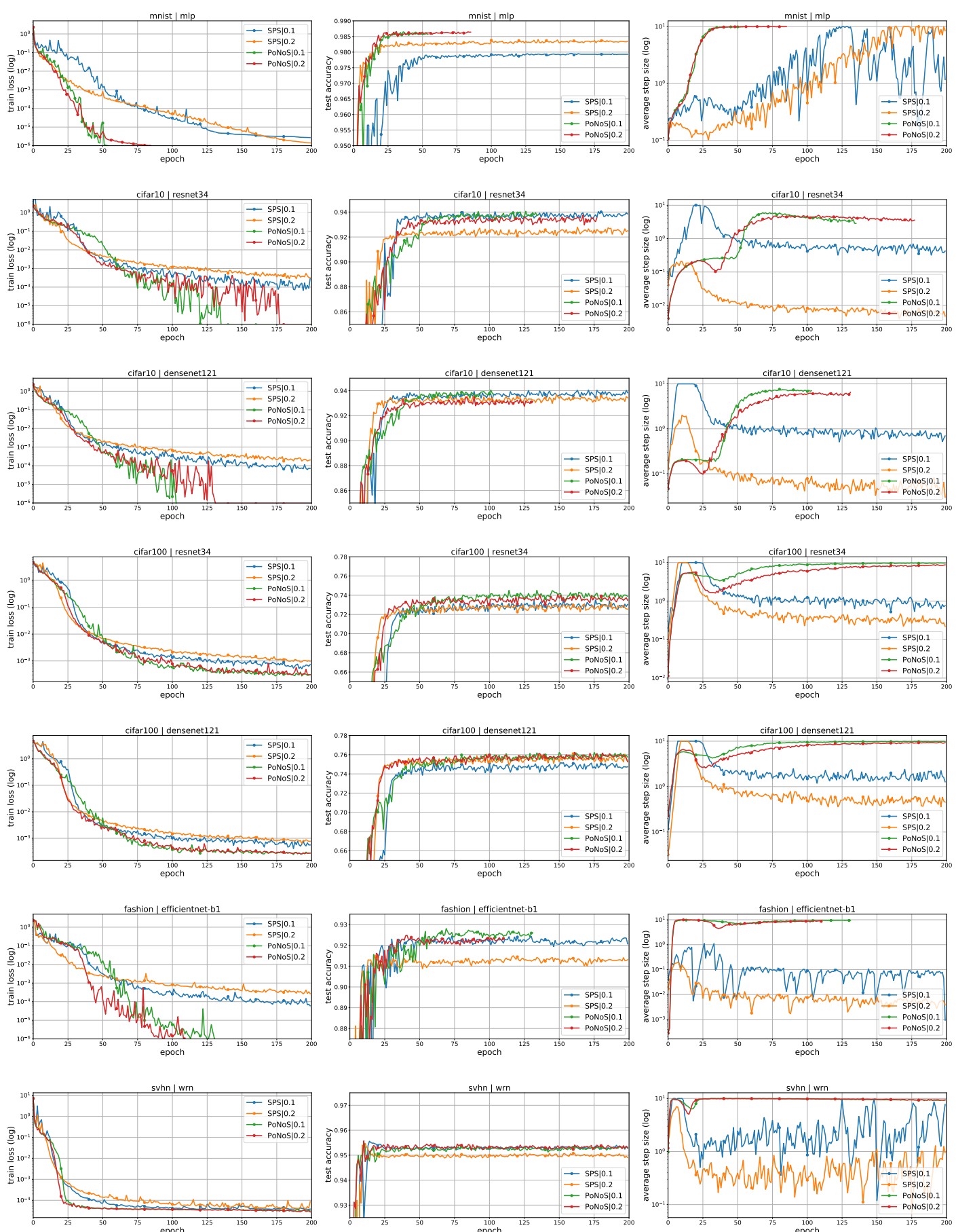

Figure XI: Comparison between the use of the constant $c_p = 0.1$ and $c_p = 0.2$ on PoNoS and SPS. Each row focus on a dataset/model combination. First column: train loss. Second column: test accuracy. Third column: average step size of the epoch.

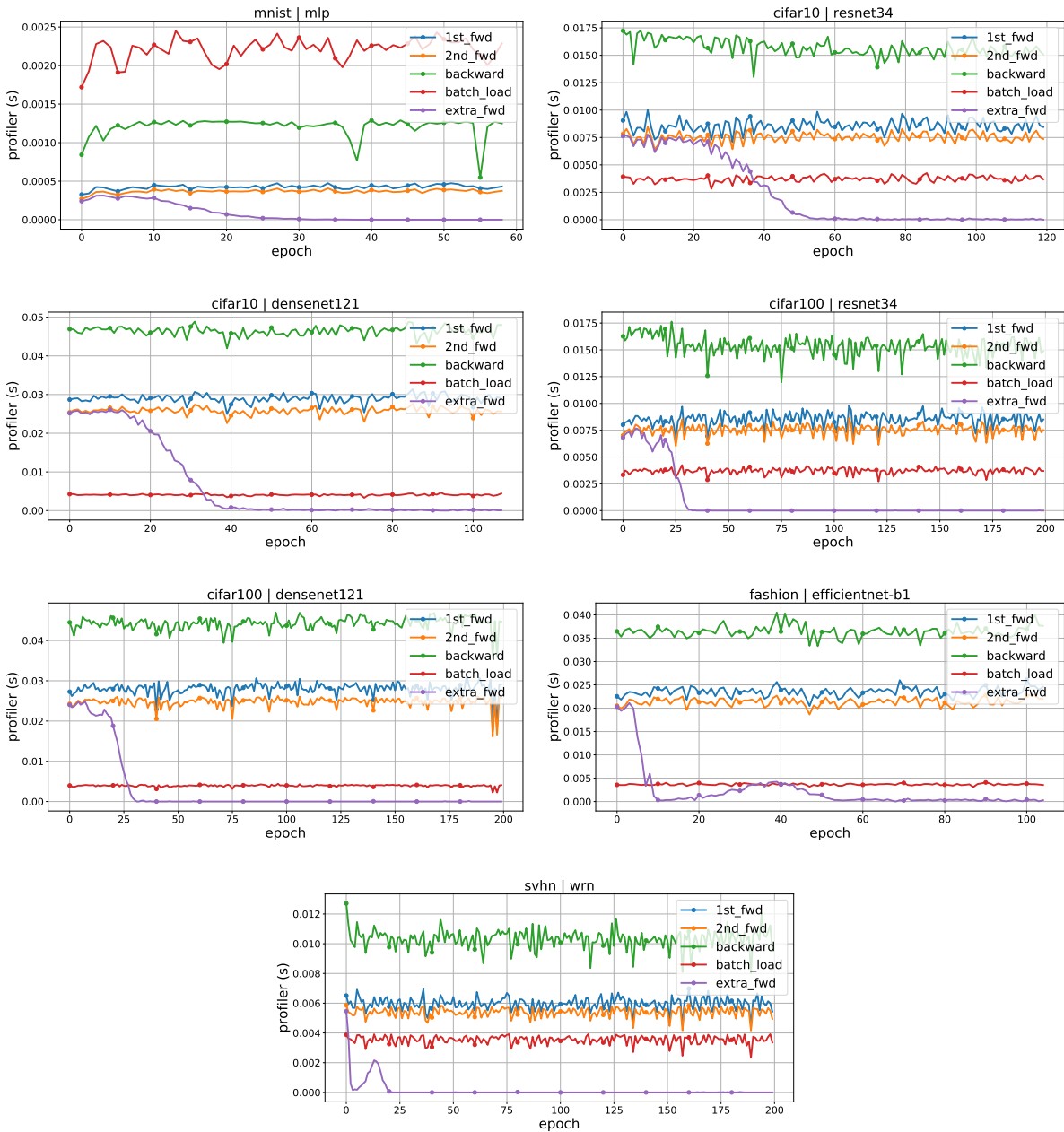

Figure XII: Profiling of a single iteration of PoNoS. We report the time of the 1st and second forward passes, the time of all the extra forward passes beyond the second, the time of the backward pass and the time to load the mini-batch in the GPU.