# OpenReview forum: "Don't be so Monotone: Relaxing Stochastic Line Search in Over-Parameterized Models"
_NeurIPS.cc/2023/Conference — NeurIPS 2023 poster_

### Official Review · Reviewer_9zy1 · 2023-06-30

**Soundness:** 3 good
**Presentation:** 3 good
**Contribution:** 3 good
**Rating:** 6
**Confidence:** 3

**Summary:**

This paper proposes the use of nonmonotone line search methods to speed up the optimization process of modern deep learning models, specifically Stochastic Gradient Descent (SGD) and Adam, in over-parameterized settings. The proposed method relaxes the condition of a monotonic decrease in the objective function and allows for larger step sizes. The authors introduce a new resetting technique that reduces the number of backtracks to zero while still maintaining a large initial step size. The proposed POlyak NOnmonotone Stochastic (PoNoS) method combines a nonmonotone line search with a Polyak initial step size. The paper proves the same rates of convergence as in the monotone case. The experiments show that nonmonotone methods outperform the rate of convergence and also generalization properties of SGD/Adam.

**Strengths:**

- **Originality:** The use of nonmonotone line search methods to relax the condition of a monotonic decrease in the objective function is a stochastic generalization of [Zhang and Hager 2004] which was proposed initially for deterministic optimization. The initial step size is chosen on the basis of previous work [Vaswani et al 2019]. The paper also introduces originally a new resetting technique that reduces the amount of backtracks to zero while still maintaining a large initial step size. Overall, the paper's originality is a significant strength.

- **Quality:** The paper provides rigorous proof that the proposed nonmonotone line search method has the same rate of convergence as in the monotone case despite the lack of a monotonic decrease. The experiments show that nonmonotone method has a larger speed of convergence and better generalization properties of SGD and Adam. Computational time comparison experiments also show the outperformance of the proposed method. The theory is solid and the experimental results are strong.

- **Clarity:** The paper is well-written and easy to understand, with clear explanations of technical terms and concepts. Qualitative explanations of the theorems are provided to help the readers understand the main messages. The authors provide detailed descriptions of the proposed method and the experiments conducted to evaluate its performance. Comparisons with other methods are presented clearly.

- **Significance:** The proposed method shows the outperformance of existing state-of-the-art algorithms in both computational time and generalization properties.

**Weaknesses:**

- The proposed method includes many parameters to be chosen artificially, such as $\eta_{\rm max}$, $c$, $c_p$, $\delta$, and $\xi$. Although the ranges of them are provided in the theorems, influences on the performance of the proposed method due to different choices of these parameters are not clear. Are the specific values used in a real experiment not so important? If so, to what extent?

**Questions:**

Same as stated in the **Weakness** part.

**Limitations:**

The limitations of the proposed method are stated not sufficiently. Only future perspectives are stated. For example, considering the local PL assumption. The claims that the proposed method outperforms many other the-state-of-the-art methods from several perspectives are quite strong. Are there any drawbacks, or points to be improved, of the proposed method?

---

> ### Author Rebuttal · Authors · 2023-08-08
>
> We would like to thank the reviewer for the accurate comments and the time spent reading the paper.
>
> Weaknesses:
>
>   The performance of PoNoS is actually not sensitive to hyperparameters. In fact, the same values work across experiments and there was no need to fine-tune them. Most of PoNoS's hyperparameters were set to very standard values, while others were either inherited by recent papers or fixed by the theory:
>    - $\delta = 0.5$, classical cut of the step [Nocedal and Wright, 2006]. SLS employs an unusual value of 0.9 and this choice is connected to the use of their resetting technique (3). We checked the results of SLS with $\delta=0.5$ and they indeed turned out to not be as good as with $\delta=0.9$.
>    - $\xi = 1$, fully nonmonotone version of Zhang and Hager [2004].
>    - $\eta^{\text{max}} = 10$, very classical value [Vaswani et al., 2019, Loizou et al., 2021]. We conducted an ablation study in Section E.4, which shows that larger values of $\eta^{\text{max}}$ do not have a remarkable impact on the results. These results show that PoNoS is more robust than SLS and SPS to these changes.
>    - $c = 0.5$, suggested by the theory. Both our theory and that of Vaswani et al. [2019] suggest employing 0.5 for $c$, rather than the classical 0.1 or lower [Nocedal and Wright, 2006]. The numerical results in Section E.1 of the supplementary support this choice. In particular, they show in Figure VII that $c=0.1$ might bring PoNoS and its monotone counterpart to diverge.
>    - $c_p = 0.1$, half of the inherited value [Loizou et al., 2021]. In Loizou et al. [2021], the value 0.2 was suggested for SPS, however in our case, the initial step size is not the final step since the backtracking procedure might reduce this value to its half (or less). For this reason, we decided to employ a step that is initially double that of SPS. The results show that PoNoS|0.1 is consistently better than PoNoS|0.2 (see Section E.5 of the supplementary). Thus, we also checked whether SPS|0.1 would be consistently better than the original SPS, but this is not the case.
>
>
> Limitations:
>
>   - In the case of transformers, PoNoS's advantage over Adam is not consistent with the one obtained for convolutional neural networks. The reason for this seems to be the reduced dynamics of the loss and of the norm of the gradient in the case of transformers. These two values directly determine the Polyak step, which consequently remains very flat along the training (see section D.5 of the supplementary). This observation suggests that (Po)NoS might need to be paired with a new initial step size and/or a different preconditioner for training transformers.
>
>   - Given the local PL result by Liu et al. [2022], our Theorem 3 can only be considered to hold locally and only for neural networks with some specific properties that are still not completely realistic. In fact, Theorem 8 by Liu et al. [2022] holds only for very wide networks with a squared loss function and whose tangent kernel at initialization is strictly positive deﬁnite. To the best of our knowledge, other works on neural networks use the same assumptions since they are still the closest available to applications.
>
>   - The $\xi$ permitted by our theorems is very small. On the other hand, the value employed in practice is large ($\xi = 1$). We conjecture that the bound on $\xi$ might be relaxed, even if the final shape of this bound might depend on the solution to the above-mentioned issue on Theorem 3.

---

> > ### Comment · Reviewer_9zy1 · 2023-08-19
> >
> > Your reply addresses my questions. Thank you very much. I will keep my score as "Weak Accept".

---

### Official Review · Reviewer_qQnN · 2023-07-04

**Soundness:** 3 good
**Presentation:** 4 excellent
**Contribution:** 2 fair
**Rating:** 5
**Confidence:** 4

**Summary:**

This paper proposes a non-monotonic line search method for choosing step sizes in stochastic optimization. Convergence rates are proved for strongly convex, convex, and PL functions, and the rates match those of previous work. Experimental results show that (1) for MLPs and CNNs, the proposed algorithm outperforms SGD, Adam, and previous line search methods, and (2) for kernel models and transformers, the proposed algorithm outperforms SGD and previous line search methods, and is competitive with Adam.

**Strengths:**

1. The question is significant. Given the observations of the "edge of stability" and non-monotonic decreases in loss when training deep networks, it seems natural that incorporating non-monotonicity into line search methods may yield significant performance improvements.
2. The presentation is clear and easy to follow.
3. The theoretical results (Theorems 1, 2, 3) can recover convergence rates from previous work.
4. The experimental evaluation is very broad, covering many datasets and neural network architectures.

**Weaknesses:**

1. The proposed algorithm appears to be a direct combination of existing techniques (non-monotonic line search with Polyak initial step size). While this isn't necessarily a problem in itself, as a result the technical novelty of the paper is not very high.
2. The theoretical results recover the previous convergence rate, but they do not exhibit any improvement over baselines. Recovering the previous convergence rates is natural and the proofs don't appear to contain any new techniques. Therefore, the theoretical contribution is not significant.
3. The main text contains no information about the tuning procedure for baselines or for the proposed algorithm, and the appendix contains very little information about hyperparameters. It's uncertain whether the experimental comparison is fair, and since the theoretical results do not exhibit improvement over baselines, the experimental performance is the only substantial contribution. Some previous baselines require additional hyperparameters (e.g. SPS with $\gamma$), but there is no assurance that this parameter was properly tuned. It is also difficult to see whether PoNoS was tuned more extensively than baselines, which would of course not be a fair comparison.
4. Evaluation for RBF kernel models and transformers does not compare by wall-clock time, and does not include results for test loss. Since PoNoS is competitive with Adam when measured by epochs, it is natural to assume that PoNoS lags behind Adam in terms of wall-clock time, which begs the question whether existing line search methods are useful for training Transformers. Line 344 says that only the training procedure is considered (following previous work), but this is not completely satisfying to me. If we compared test performance for MLPs, and CNNs, why not also for RBFs and Transformers?

**Questions:**

1. Why is interpolation important to achieve the theoretical results? The introduction discusses interpolation in detail, but Section 4 only mentions interpolation as a condition for the theorems, and does not discuss why interpolation is necessary. Do previous line search methods also require interpolation to recover the same convergence rates?
2. How were the hyperparameters chosen? In particular, were all algorithms fairly tuned?
2. Why does PoNoS outperform other line search methods empirically when the theoretical guarantees of PoNoS do not improve over baselines?
3. How much do the two individual components of PoNoS (non-monotonicity and choice of initial step size) affect the performance? In particular, how would PoNoS perform if we used non-monotonicity with a classical initial step size?

As a miscellaneous suggestion, please use lower resolution images for Figures 1-3, or if you're using a vector format please remove some data points. The provided PDF is unusually large and viewing the figures in Chrome's PDF viewer is quite slow.

**Limitations:**

The authors include some discussion of limitations and future work in the conclusion, though it would be nice to see some more discussion of the weaknesses of the proposed method instead of just directions for future work. Discussion of potential negative societal impact is, in my opinion, not necessary for this paper.

---

> ### Author Rebuttal · Authors · 2023-08-08
>
> We would like to thank the reviewer for the comments and the time spent reading the paper. As a general answer, it appears that many of his/her statements are influenced by the reviewer's belief regarding our theory not containing any novelty. In the reply to Weakness 2. below, we clarify that our proof does contain a new technique, which is a central contribution of our work and that seems already to have a candidate for being reused.
>
> Weaknesses:
>
>  1. Below, we provide an example of a valuable technical novelty of our work. Please also see the reply to Reviewer CWjK.
>
>  2. We understand the concern of the reviewer, however the derived theorems are not a straightforward consequence of Theorems 1-2 of Vaswani et al. [2019]. In particular, the proof structure is similar to theirs, on the other hand, the idea of summing the monotone and nonmonotone sequences is original and it is a non-trivial contribution of our study. This proof technique allows us to show for the first time in the stochastic setting that\
>  (*) the difference between the nonmonotone and the monotone terms is geometrically converging to 0.\
>  A few evidences of the non-triviality of our achievement are:
>   - all the existing convergence rates for stochastic nonmonotone methods assume (*), instead of proving it (since [Krejić and Krklec 2015]);
>   - in the preprint by Hafshejani et al. [2023], the converge proof seems incomplete because the authors did not show (*) but only an asymptotic version of it;
>   - one of the main contributions of Grippo et al. [1986] is the inductive proof showing (*) for the deterministic setting.
>
>  3. PoNoS's hyperparameters were not fine-tuned, they were either set to very standard values or fixed by theory (see reply to Reviewer 9zy1 for a detailed discussion). The reviewer refers to $\gamma$ of SPS. If the reference is to the maximum step size of Loizou et al., [2021], the robustness analysis for SPS, SLS and PoNoS can be found in Section E.4 of the supplementary. If the reference is to $\gamma$ in (3) (which is only affecting SLS and SPS), our experiments were not conclusive towards changing it, so we decided to keep the default value chosen by Loizou et al., [2021]. Notice that we performed a per-problem fine-tuning of hyperparameters only for the learning rates of SGD and Adam.
>
>  4. Concerning transformers, as the reviewer pointed out, PoNoS is competitive with Adam in terms of epochs, but slower than it in terms of wall-clock time. On the other hand, Adam's learning rate has been fine-tuned, while PoNoS has been used off-the-shelf. In fact, as soon as a hyperparameter selection is performed to select its learning rate, Adam is overall slower also in terms of cumulative-training time. It is true, however, that the improvement of PoNoS over Adam for training transformers is not as noticeable as that over SGD for CNN. In Section D.5 of the supplementary, we discuss the limitation of the Polyak step in this set of experiments. As in the case of SGD, it seems that transformers need to be treated differently also for (Po)NoS, especially in terms of its initial step size.
>     Regarding the wall-clock time of training RBF kernel models, these values are in the order of fractions of seconds, so we decided to not report them. The accuracy of RBF kernel models on the binary classification tasks can be found in Figure IV of the supplementary.
>
> Questions:
>
>  1. We would like to thank the reviewer for the question, we will clarify this point. The interpolation assumption is the property that allows stochastic methods to achieve linear rates [Ma et al. 2018] without needing to grow the batch size. Together with L-smoothness, this assumption allows the weak growth condition to be replaced by an alternative bound on the variance of the mini-batch gradients. A large chunk of the methods designed after Ma et al. [2018] assume interpolation because it is a realistic property of neural networks and because it allows us to avoid designing a hand-crafted sequence of step sizes. This includes the previous monotone line search [Vaswani et al., 2019] and the stochastic Polyak step size method [Loizou et al., 2021].
>  2. See above.
>  3. Our theory is consistent with the literature. In fact, none of the existing nonmonotone rates is faster than their monotone counterpart [Grippo et al. 1986, Raydan 1997, Dai 2002, Zhang and Hager 2004] not even in terms of constants. But nevertheless non-monotonicity has shown to improve practical performance in a variety of settings. Please also see the reply to all reviewers.
>  4. This question is replied half in Section D.2 and half in Section E.2 of the supplementary. In Section D.2, we fix the use of the proposed nonmonotone stochastic line search and study different initial step sizes. In particular, the Polyak step is by far the best initial step among those explored. In Section E.2, we fix the use of the Polyak intial step and analyze different line search methods. The results show that the adapted stochastic Zhang and Hager [2004] line search achieves the best performance, especially because of the dominance in terms of test accuracy and the heavy computational costs of single_batch_grippo.
>
> Miscellaneous:
>  Thanks for the suggestion, we will lower the resolution of all images.
>
> Limitations:
>  The limitations of PoNoS were partially discussed in the answer to the reviewer's comment on transformers, but a complete frame is given in the reply to Reviewer 9zy1.
>
>
> References not in the main paper:
>
>  [Ma et al., 2018] Siyuan Ma, Raef Bassily, and Mikhail Belkin. "The power of interpolation: Understanding the effectiveness of SGD in modern over-parametrized learning". In International Conference on Machine Learning, pages 3325–3334. PMLR, 2018.
>
>  [Dai, 2002] Dai, Yu-Hong. "On the nonmonotone line search." Journal of Optimization Theory and Applications 112 (2002): 315-330.

---

> > ### Comment · Reviewer_qQnN · 2023-08-18
> >
> > Thank you for your response, which cleared a few things up. There is slightly more theoretical contribution than I originally thought, and the discussion of the condition (*) clarified this contribution. However, I still believe that recovering the previous rate (as opposed to improving it) is a little unsatisfying. It is true that some papers, like Adam, only recover the previous rate in theory while improving the performance in practice. However, I don't think that the empirical results in this paper are enough to make up for the lack of theoretical improvement in the same way as e.g. Adam. The authors claim that the proposed algorithm is "the first stochastic nonmonotone line search method able to train neural networks efficiently while simultaneously achieving state-of-the-art generalization performance". I do not agree that PoNoS exhibits SOTA generalization performance, at least not for all tasks. The results in Figures 1-3 are good, but the main body contains no results on the test set for kernel models or transformers. The appendix does contain some test results for kernel models, but PoNoS does not outperform other methods. My biggest concern is that there are no test results for transformers anywhere, and without these results I don't believe the claim that PoNoS has SOTA generalization. With all of this in mind, the empirical improvement does not feel substantial enough to make up for the lack of theoretical improvement. I will keep my rating.

---

> > > ### Author Response · Authors · 2023-08-21
> > >
> > > "However, I still believe that recovering the previous rate (as opposed to improving it) is a little unsatisfying. It is true that some papers, like Adam, only recover the previous rate in theory while improving the performance in practice."
> > >
> > > It is true that in an ideal world we would be able to prove a faster rate. However, the (almost 4 decades of) impressive practical performance of non-monotone methods have thus far resisted analysis. On the other hand, it is reassuring that we can still match the rate of SOTA methods like SLS and SPS. Note that Adam did *not* recover the previous rates in theory (the proof in the original Adam paper is obviously wrong). The same is true for other important practical methods like AdamW and cosine annealing which have become the most popular optimization algorithms in the field. These practical contributions inspired later theoretical works exploring how to justify their empirical performance.
> > >
> > > "However, I don't think that the empirical results in this paper are enough to make up for the lack of theoretical improvement in the same way"
> > >
> > > We are confused by this comment. Our experiments indicate that PoNoS is the same or better than a variety of existing methods, across a range of tasks and datasets (the reviewer also acknowledged the breadth of our experiments within the Strengths of our paper). All our evidence points to PoNoS being a method that is useful in practice. And they indicate that PoNoS is particularly effective for CNNs, one of the most important model classes in computer vision. Our experiments are more extensive than most empirical works presenting new optimization algorithms, with most empirically-motivated methods only first showing success on a small-but-important class of problems (the AdamW paper with >10,000 citations only presented results on CIFAR-10 and downscaled ImageNet datasets).
> > >
> > > "The appendix does contain some test results for kernel models, but PoNoS does not outperform other methods"
> > >
> > > In these settings PoNoS outperformed the other methods on one dataset while matching SOTA performance on the other 3. This is in contrast to Adam, for example, which can perform poorly on convex problems [Reddi et al., 2018]. We should not expect any method to strictly outperform all other methods in every scenario.
> > >
> > > "My biggest concern is that there are no test results for transformers anywhere, and without these results I don't believe the claim that PoNoS has SOTA generalization."
> > >
> > > In the transformer experiments, all methods achieve the same test error while methods that decreased the training error faster achieve this error faster. We will update the paper to include these results.

---

### Official Review · Reviewer_CWjK · 2023-07-08

**Soundness:** 3 good
**Presentation:** 3 good
**Contribution:** 2 fair
**Rating:** 5
**Confidence:** 4

**Summary:**

This paper presents a non-monotone line search method for optimizing over-parameterized models. The method is equipped with some theoretical support for strongly convex, convex and the PL condition. Furthermore, experimentally, the method is shown to have favorable performance when optimizing various deep learning models of practical interest.

==> post rebuttal: increased score from 4 to 5.

**Strengths:**

Obtaining convergence results with non-monotone line search strategies appears to be a novel contribution - though, I don't know if some variant of this result has appeared in existing literature dealing with non-monotone line search.

**Weaknesses:**

- The proposed approach seems incremental compared to existing approaches.
- The theory doesn't adequately capture why the proposed method outperforms existing approaches. The bounds suggest identical convergence rates as ones that use monotone line search methods. This suggests all these bounds are fairly worst case (loose upper bounds) that do not help quantify why these methods work well in practice in the first place.

**Questions:**

- In the theorem, particularly for the PL case, the degree of (non-) monotonicity that can be tolerated seems to be rather small (the permitted \zeta appears to be super small)? Can the authors clarify this?

- The theorem for the strongly convex case doesn't appear to be particularly applicable in the over-parameterized case unless a regularized objective is being solved for, which doesn't reflect on how the behavior manifests on the loss function we truly care for.

- Can the authors clarify why the proposed stepsizes tend to increase as a function of iterations? Is this related to using normalization layers (e.g. batch norm/layer norm) within the architectures? How do these stepsizes look like for convex case in the experiments conducted in the paper?

**Limitations:**

Yes.

---

> ### Author Rebuttal · Authors · 2023-08-08
>
> We would like to thank the reviewer for the comments and the time spent reading the paper.
>
> Weaknesses:
>
>   · We understand the reviewer's position on this matter, however we would like to stress a few factors that make our contribution non-incremental. PoNoS is the first stochastic nonmonotone line search method able to train neural networks efficiently while simultaneously achieving state-of-the-art generalization performance. To the best of our knowledge, the following contributions were not existing in the literature before:\
>     &nbsp;&nbsp;1.1 the stochastic Zhang and Hager [2004] line search (2);\
>     &nbsp;&nbsp;1.2 the memory-based resetting technique (5);\
>     &nbsp;&nbsp;1.3 the combination of a Polyak initial step size and a Zhang and Hager [2004] line search (not even in the deterministic setting);\
>     &nbsp;&nbsp;1.4 Theorems 1-3 and the proof technique employed to show them.\
>     Of the above elements, 1.2 and 1.4 are non-incremental contributions of our work. In fact, (5) is very different from (3) and the idea of summing the monotone and nonmonotone sequences is original and it is a non-trivial contribution of our study.
>     Regarding 1.1 and 1.3, we agree with the reviewer on the fact that they seem more easily reachable from the existing methods. On the other hand, the process of designing them included creative and non-conventional elements that in the end were replaced by adaptations of more consolidated techniques. We believe that this is actually a stregth of our method and not a weakness. In fact, simple changes with big practical impact are often very appreciated contributions in the long term.
>
>   · Even though we do not improve on the worst-case theoretical rate, we match the worst-case theoretical rate while significantly improving practical performance. Please also see the reply to all reviewers.
>
>
> Questions:
>
>   · The reviewer is right, the $\xi$ permitted by Theorem 3 is very small. This bound is a consequence of the fact the nonmonotone term needs to be "squeezed" between the monotone sequence and the linear bound. In fact, also in the deterministic case, monotone and nonmonotone sequences need to be geometrically converging to the same value. We conjecture that this bound does not need to be this tight, but we leave this exploration to future works.
>
>   · We are not sure we understood the reviewer's comment correctly, which are the loss functions we truly care for? In deep learning, it is true that neural networks do not employ any regularization, on the other hand, one might be interested in applying our method to more classical machine learning models (e.g., linear Support Vector Machines (SVM)) on a dataset with many more features than instances. In this case, the over-parametrization property would be satisfied and also the strong convexity (SVM models are regularized models).
>
>   · The step size increases because the squared norm of the gradient decreases as we get closer to a stationary point (see fourth column of Figure I of the supplementary). This is the consequence of the Polyak step size, whose denominator is the squared norm of the gradient. The same happens in the case of the convex experiments (see the third column of Figure IV of the supplementary).

---

> > ### Comment · Reviewer_CWjK · 2023-08-18
> > **Re. author response**
> >
> > Thanks to the authors for their clarifications. I will increase my score to a 5.
> >
> > My comment about the loss function was that even if we solve the regularized loss, the eventual loss that we care about is the one without regularization, so the rates of convergence to the regularized loss don't really matter all that much. Nevertheless, thanks your response.

---

### Official Review · Reviewer_hZgC · 2023-07-11

**Soundness:** 2 fair
**Presentation:** 2 fair
**Contribution:** 3 good
**Rating:** 5
**Confidence:** 2

**Summary:**

This submission proposed a new linear search method to ensure convergence without the monotone decrease condition of the (mini-)batch objective function. The method is quite suitable for the modem DNN training, which prefers the larger training learning rate.

**Strengths:**

- The explanation of motivation is very clear.
- The related work has been extensively discussed.

**Weaknesses:**

- Discussing the difference between the proposed and the previous methods is inadequate, especially since the existing methods inspire some steps.
- The theoretical benefits of the proposed method are not shown/discussed in the convergence rate results. The proposed method seems to share a similar rate with the previous results. However, since PoNoS prefers a larger learning rate, its rate at least can demonstrate the advantage of a constant level.
- There is no numerical comparison in the main part, and just having the curve figures cannot fully display the results.

**Questions:**

- The experimental results are not entirely convincing. With the development of neural network training technology in recent years, the Adam optimizer can achieve better training losses than SGD in most cases. It is strange that SGD beat Adam constantly, especially on the transformer model for the NLP task.

**Limitations:**

See weakness.

---

> ### Author Rebuttal · Authors · 2023-08-08
>
> We would like to thank the reviewer for the comments and the time spent reading the paper.
>
> Weaknesses:
>
>   · We don't understand the reviewer's comment, in what sense is our discussion on the difference between the proposed and the previous methods inadequate?
>
>   · While it is true that nonmonotone methods take larger steps, they do not ensure that the function value will decrease at each iteration (but only every $W$ iterations). Even if practically this is a real advantage (i.e., it grants line searches more freedom of choice), this property it is not captured by the theory. In fact, none of the existing nonmonotone rates is faster than their monotone counterpart [Grippo et al. 1986, Raydan 1997, Dai 2002, Zhang and Hager 2004] not even in terms of constants. But nevertheless non-monotonicity has shown to improve practical performance in a variety of settings. For a discussion on the provable advantages of one method over another, see the general repose to all reviewers.
>
>   · We don't understand the reviewer's comment, what does the reviewer mean with numerical comparisons?
>
> Questions:
>
>   · Our results agree with the reviewer's comment: Adam is always faster than SGD for training transformers (see the last two columns of Figure 4 of the main paper and Figure V of the supplementary) and comparable with SGD for training convolutional neural networks (e.g., Figure I of the supplementary). This observation is consistent with other works [Kunstner et al., 2023]
>
> References not in the main paper:
>
>   [Dai, 2002] Dai, Yu-Hong. "On the nonmonotone line search." Journal of Optimization Theory and Applications 112 (2002): 315-330.

---

### Official Review · Reviewer_WR5e · 2023-07-18

**Soundness:** 3 good
**Presentation:** 3 good
**Contribution:** 3 good
**Rating:** 8
**Confidence:** 3

**Summary:**

This paper proposes a new line search method for determining the step size in SGD within the interpolation regime. In contrast to the previous approach called SLS, which relies on the monotonically decreasing Amijo condition, the proposed method adopts the non-monotone Zhang & Hager line search. The authors establish the convergence guarantees for the proposed Stochastic Zhang & Hager line search when an upper bound is placed on the initial step size, considering strongly convex, convex, and PL problems. Additionally, they introduce several enhancements to improve empirical performance: (1) utilizing the Stochastic Polyak Step (SPS) to set the initial step size of the line search and (2) introducing a new resetting technique to reduce the number of backtracking steps.








**Strengths:**

- Numerical experiments are extensive and the performance of PoNoS looks quite promising. It is great to see a provably convergent optimization algorithm (with other heuristics) working well in large-scale experiments while incurring only a minor computation overhead.
- The proof techniques look new and interesting.

**Weaknesses:**

- There are certain expressions within theorems that may lead to confusion. I have elaborated on these concerns in the "Questions" section of my review.
- The Polyak step and the resetting technique are only heuristics. It would be valuable to provide further analysis regarding how these heuristics influence the convergence properties of the algorithm.

**Questions:**

- While reading the main text, I was initially skeptical about Theorem 1 and 2 due to their requirement of Lipschitz continuity for function $f$. It has been shown that the bounded gradient assumption contradicts the strong convexity [1]. Upon reviewing the proof in the appendix, it seems that the Lipschitz continuity of $f$ may not be necessary for Theorem 1 and 2. Additionally, in Theorem 3, it is unclear why the smoothness of $f$ needs to be separately assumed given that $f_i$ is $L_i$-smooth. It would be helpful if the authors could provide clarification.
- Furthermore, the notation "LC" used in Theorem 2 and 3 lacks a definition in the main text. The corresponding definition can be found in Appendix B.
- In theory, how can you ensure that backtracking (Lines 10 - 12) halts in finite steps?

[1] Nguyen, Lam, et al. "SGD and Hogwild! convergence without the bounded gradients assumption." International Conference on Machine Learning. ICML, 2018.

---

> ### Author Rebuttal · Authors · 2023-08-08
>
> We would like to thank the reviewer for the accurate comments and the time spent reading the paper.
>
> Weaknesses:
>
>   Additional studies on the Polyak step alone show its connections to the "better model" of Asi and Duchi [2019] in Gower et al. [2021] and to the "passive aggressive" optimization framework [Gower et al. 2022]. However, we agree with the reviewer, it would be valuable to study what theoretical influence has the Polyak step when employed as an initial step for line search methods. Regarding the new resetting technique, we obtained some preliminary results linking $l_k$ and $l_{k-1}$, but they were not conclusive and they were left out of the paper. We will consider both these directions in future works.
>
> Questions:
>
>   · We would like to thank the reviewer for the very constructive feedback, we didn't realize that the assumption on the Lipschitz continuity of $f$ may not be necessary. We will study this extension in future works. Moreover, it is true that the smoothness of $f$ is not needed, once we assume that of the $f_i$. The reason for this redundancy was just that of defining $L$ ($L$ would be upper-bounded by the average of the $L_i$).
>
>   · We notice this issue and replaced the notation "LC" directly with "Lipschitz smoothness".
>
>   · The fact that backtracking halts in finite steps is a consequence of Lemma 1 from the supplementary. This lemma shows that the step size yielded by the backtracking algorithm has a lower bound. This, together with the fact that the initial step size has an upper bound, shows that the amount of backtracks is limited.

---

> > ### Comment · Reviewer_WR5e · 2023-08-12
> >
> > Thank you for the response. I am keeping my current rating to champion this paper.

---

### Official Review · Reviewer_TC8b · 2023-08-02

**Soundness:** 3 good
**Presentation:** 3 good
**Contribution:** 2 fair
**Rating:** 5
**Confidence:** 3

**Summary:**

The paper presents a proposed Polyak nonmonotone stochastic (PoNoS) method which combines a nonmonotone line search with a Polyak initial step size. It builds on the work of Vaswani et al. [2019] by modifying the monotone line search to incorporate a nonmonotone approach.

**Strengths:**

Originality:
The introduction of nonmonotone line search applied to Deep Learning is noteworthy.
Clarity:
The paper is generally well-written. However, the authors should emphasize their specific methodological innovation and contribution to the conclusion section to highlight the significance.

Quality: The overall quality is good. The paper demonstrates the outperformance of the proposed methods over other methods in regard to runtime.

Significance:
The experiments show the advantages of the methods in terms of convergence speed and runtime when applied to Deep Learning.

**Weaknesses:**

The authors admitted the adverse impact on convergence speed of the proposed method. This matter should be further investigated to have a more comprehensive intuition of the robustness of the method.

Additionally, the authors should conduct a further comparison with the state-of-the-art line search approaches and the based model of Vaswani et al. [2019]
?a Figurethis

**Questions:**

1.	Lines 12-23 are about a reduction of the number of backtracks to zero. Can you specify which parts of the experiments to support the argument.
2.	Did you have comparison with he based model of Vaswani et al. [2019]?
3.	Can you explain further or provide intuition about the observation in lines 317-319?
4.	In figure 4, data for SLS-prec, SPS-pre and PoNoS-pres seems to be missing. Do you have any reason for not to put the data?

**Limitations:**

The authors do address some of the limitations of their work.

---

> ### Author Rebuttal · Authors · 2023-08-08
>
> We would like to thank the reviewer for the comments and the time spent reading the paper.
>
> Strengths:
>   We will follow the reviewer's suggestion and highlight our contributions in Section 6.
>
> Weaknesses:
>
>   · We are not sure we understood the comment of the reviewer regarding the adverse effect on the convergence speed. We assume the reference is to our sentence "Our theory matches its monotone counterpart despite the use of a nonmonotone term" (line 367). What we meant there is that despite the convergence proof being more complicated, our method still achieves the same rate as its monotone counterpart. The additional complexity comes from the fact that nonmonotone methods only ensure the decrease of the (mini-batch) function value every $W$ iterations, instead of every iteration. In practice, this is a great advantage since it provides the line search with more freedom of choice. Despite the more complex analysis, nonmonotone rates have always been shown to be consistent with those of their monotone counterparts [Grippo et al. 1986, Raydan 1997, Dai 2002, Zhang and Hager 2004] and the same can be said for PoNoS. For a discussion on provable advantages of one method over another, see the general repose to all reviewers.
>
>   · The comparison with Vaswani et al. [2019] was reported in Figures 1, 3 and 4 of the main paper and in the corresponding Figures I, III and IV of the supplementary materials. The method is there denoted SLS. Additional experiments on other stochastic line search methods are reported in Figure VIII of the supplementary.
>
> Questions:
>
>   1. This argument is supported in Figure 2 and in the corresponding Figure II of the supplementary. Moreover, a zoom-in on the amount of backtracks is reported in Figure IX and Section E.3 of the supplementary.
>
>   2. See the reply above regarding the comparison with Vaswani et al. [2019].
>
>   3. We would like to thank the reviewer for this question, there is a mistake in the sentence, instead of "In fact, we can notice that the additional per-epoch runtime of PoNoS in the first 5-25 epochs is less than one half that of the later stage of the training." it should have been "In fact, ... is less than the double that of the later...". Also, what we meant is that the time of the last forward step (e.g., which in the case of "cifar10 | resnet34" is 4s/#batches-per-epoch) is less than the second forward step (6s/#batches-per-epoch). This observation supports the thesis that the first forward step is more expensive than the others, because of the loading of the data in the GPU memory. We will clarify this sentence and add a more rigorous timing comparison.
>
>   4. We understand the reviewer's confusion here, in Figure 4 we reported a convex experiment (first column) and two experiments on transformers (last two columns). SLS_prec, SPS_prec and PoNoS_prec were not tested on convex problems. We will clarify this point.
>
> References not in the main paper:
>
>   [Dai, 2002] Dai, Yu-Hong. "On the nonmonotone line search." Journal of Optimization Theory and Applications 112 (2002): 315-330.

---

> > ### Comment · Reviewer_TC8b · 2023-08-20
> >
> > Thank you for answering the questions.

---

### Author Rebuttal · Authors · 2023-08-08

We would like to thank all six reviewers for their feedback and the time spent reading the paper. Below we comment several reviewers' concerns regarding the lack of a theoretical result showing PoNoS's advantages over the existing methods.

We understand the reviewers' opinion, however we would like to clarify our point of view on this matter. For DL researchers and especially for the possible users of PoNoS, it is of great interest to know that it is numerically faster than state-of-the-art methods while still being backed up by the same convergence rates. This theoretical achievement is not trivial and it is more than what can be obtained for instance by Adam [Reddi et al., 2018]. Despite the lack of a theoretical justification showing why Adam is faster than SGD [Kunstner et al., 2023], Adam has a huge practical impact and it replaced SGD for training transformers. In fact, if we would always require a theoretical justification of the superiority of a method, Adam would probably not be published, while it is becoming one of the most cited works of all time. Another example that is very appropriate for our case, is the spectral projected gradient by Birgin et al. [2000]. In this paper, the authors introduced nonmonotonicity and clever step sizes in a different setting and they showed that the new method works much better in practice without showing a faster convergence rate. This paper was published in a top optimization journal and has more than one thousand citations.

References not in the main paper:

[Reddi et al., 2018] Reddi, S. J., Kale, S. & Kumar, S. On the convergence of Adam and beyond. 6th International Conference Learning Representation (ICLR), pp. 1–23 (2018).

---

### Decision · Program_Chairs · 2023-09-21

**Decision:**

Accept (poster)

**Comment:**

This paper proposes a new line search method for determining the step size in SGD within the interpolation regime. In contrast to SLS, which relies on the monotonically decreasing Amijo condition, the proposed method adopts a non-monotone line-search (following Zhang and Hager, 2004). The paper continues a line of work that focused on adaptive stepsize/line search methods for finite sum problems under interpolation. The paper highlights cases when the prior methos might be slow, and shows that the proposed method can address this in the mentioned cases.

The paper is well-written, and carefully discussing the placement of the contribution in light of prior work.

The opinion of the reviewers on this paper were divided, some were not convicted by the numerical evacuation (mostly conduced on small scale) and the many hyperparameters that might make the method difficult to tune on new classes of problems.

One reviewer was championing the paper, by arguing that this paper could impact further research on DL methods with 'non-traditional' stepsizes (e.g. larger what is dictated by conservative worst-case guarantees). Eventually, this lead to recommendation of the committee to propose acceptance (with none of the reviewers opposing this decision).

Please consider adding the clarifications and discussion points from below (especially also the clarifications on the hyperparameters) to the final version of the paper.